# A Unified Confidence Sequence for Generalized Linear Models, with Applications to Bandits

**Jungyhun Lee, Se-Young Yun**
Kim Jaechul Graduate School of AI
KAIST
Seoul, Republic of Korea
{jh_lee00, yunseyoung}@kaist.ac.kr

**Kwang-Sung Jun**
Department of Computer Science
University of Arizona
Tucson, AZ, USA
kjun@cs.arizona.edu

## Abstract

We present a unified likelihood ratio-based confidence sequence (CS) for *any* (self-concordant) generalized linear model (GLM) that is guaranteed to be convex and numerically tight. We show that this is on par or improves upon known CSs for various GLMs, including Gaussian, Bernoulli, and Poisson. In particular, for the first time, our CS for Bernoulli has a $\text{poly}(S)$-free radius where $S$ is the norm of the unknown parameter. Our first technical novelty is its derivation, which utilizes a time-uniform PAC-Bayesian bound with a uniform prior/posterior, despite the latter being a rather unpopular choice for deriving CSs. As a direct application of our new CS, we propose a simple and natural optimistic algorithm called `OFUGLB`, applicable to *any* generalized linear bandits (**GLB**; Filippi et al. (2010)). Our analysis shows that the celebrated optimistic approach simultaneously attains state-of-the-art regrets for various self-concordant (not necessarily bounded) **GLB**s, and even $\text{poly}(S)$-free for bounded **GLB**s, including logistic bandits. The regret analysis, our second technical novelty, follows from combining our new CS with a new proof technique that completely avoids the previously widely used self-concordant control lemma (Faury et al., 2020, Lemma 9). Numerically, `OFUGLB` outperforms or is at par with prior algorithms for logistic bandits.

## 1 Introduction

One paramount task in statistics and machine learning is to estimate the uncertainty of the underlying model from (possibly noisy) observations. For example, in interactive machine learning scenarios such as bandits (Lattimore and Szepesvári, 2020; Robbins, 1952; Thompson, 1933) and recently reinforcement learning with human feedback (RLHF; Christiano et al. (2017); Ouyang et al. (2022)), at each time step $t$, the learner chooses an action $\boldsymbol{x}_t$ from an available set of actions $\mathcal{X}_t$ and observes reward or outcome $r_t$ that is modeled as a distribution whose mean is an unknown function $f^*$ of $\boldsymbol{x}_t$; i.e., $r_t \sim p(\cdot|\boldsymbol{x}_t; f^*)$. One popular choice of such a model is the **generalized linear model** (GLM; McCullagh and Nelder (1989)) that extends exponential family distributions to have a linear structure in its natural parameter as $\langle \boldsymbol{x}, \boldsymbol{\theta}_\star \rangle$, where $\boldsymbol{\theta}_\star$ is an unknown parameter. In other words, the mean function is $f^*(\boldsymbol{x}) = \mu(\langle \boldsymbol{x}, \boldsymbol{\theta}_\star \rangle)$ for some inverse link function $\mu$. This encompasses a wide range of distributions, which in turn makes it ubiquitous in various real-world applications, such as news recommendations (Bernoulli; Li et al. (2010, 2012)), social network influence maximization (Poisson; Gisselbrecht et al. (2015); Lage et al. (2013)), and more. In such tasks, the learner must estimate the uncertainty about $\boldsymbol{\theta}_\star$ *at each time step* $t \geq 1$, given observations $\{(\boldsymbol{x}_s, r_s)\}_{s=1}^{t-1}$, to make wise decisions. One popular and useful way to capture the uncertainty is via a *time-uniform confidence sequence (CS)* $\{\mathcal{C}_t(\delta)\}_{t=1}^{\infty}$, which takes the form of $\mathbb{P}[\exists t \geq 1 : \boldsymbol{\theta}_\star \notin \mathcal{C}_t(\delta)] \leq \delta$. Recently, CS has been described as one of the key components for *safe anytime-valid inference (SAVI)* that can ensure the validity/safeness of sequentially adaptive statistical inference (Ramdas et al., 2023).

38th Conference on Neural Information Processing Systems (NeurIPS 2024).

Existing CSs for GLMs, however, are far from ideal. Much of the prior works focus on obtaining CS for specific instantiations of GLMs, such as Gaussian (Abbasi-Yadkori et al., 2011; Flynn et al., 2023) and Bernoulli (Abeille et al., 2021; Faury et al., 2020, 2022; Lee et al., 2024). Especially for Bernoulli, all the existing CSs suffer from $\mathrm{poly}(S)$ factor in the radius, where $S$ is the norm of the unknown parameter $\boldsymbol{\theta}_\star$. Emmenegger et al. (2023); Jun et al. (2017); Li et al. (2017) proposed generic CSs that work for any convex GLMs, but their radii all suffer from a globally worst-case curvature of $\mu$, which is detrimental in many cases (e.g., for Bernoulli, it scales as $e^S$).

**Contributions.** First, we propose a *unified* construction of likelihood ratio-based CS for any convex GLMs (Theorem 3.1) and then instantiate it as an ellipsoidal CS for self-concordant GLMs, including Bernoulli, Gaussian, and Poisson distributions (Theorem 3.2). *Notably, we keep track of all the constants so that any practitioner can directly implement it without trouble.* The proof uses ingredients from time-uniform PAC-Bayesian bounds (Chugg et al., 2023) – martingale + Donsker-Varadhan representation of KL + Ville's inequality. The main technical novelty lies in using *uniform* prior/posterior for the analysis, inspired by various literature on portfolios (Blum and Kalai, 1999) and fast rates in statistical/online learning (Foster et al., 2018; Grünwald and Mehta, 2020; Hazan et al., 2007; van Erven et al., 2015).

Secondly, we apply our novel CSs to contextual generalized linear bandits (**GLB**; Filippi et al. (2010)) with changing (and adversarial) arm-sets, and propose a new algorithm called **Optimism in the Face of Uncertainty for Generalized Linear Bandits** (OFUGLB). OFUGLB employs the simple and standard optimistic approach, choosing an arm that maximizes the upper confidence bound (UCB) computed by our CS (Abbasi-Yadkori et al., 2011; Auer, 2002). We show that OFUGLB achieves the state-of-the-art regret bounds for self-concordant (possibly *unbounded*) **GLB** (Theorem 4.1). This is the first time a computationally tractable, *purely* optimistic strategy attains such $\mathrm{poly}(S)$-free regret for logistic bandits in that OFUGLB does not involve an explicit warmup phase and only involves convex optimization subroutines. Our other significant main technical contribution is the analysis of OFUGLB, as naïvely applying existing analysis techniques for optimistic algorithms (Abeille et al., 2021; Lee et al., 2024) yields a regret bound whose leading term scales with $\mathrm{poly}(S)$. We identify the key reason for such additional dependency as the use of self-concordance control lemma (Faury et al., 2020, Lemma 9), and provide an alternate analysis that completely bypasses it, which may be of independent interest in the bandits community and beyond.

## 2 Problem Setting

We consider the realizable (online) regression with the **generalized linear model** (GLM; McCullagh and Nelder (1989)) whose conditional probability measure of $r$ is given as

$$dp(r|\boldsymbol{x};\boldsymbol{\theta}_\star) = \exp\left(\frac{r\langle\boldsymbol{x},\boldsymbol{\theta}_\star\rangle - m(\langle\boldsymbol{x},\boldsymbol{\theta}_\star\rangle)}{g(\tau)} + h(r,\tau)\right) d\nu, \qquad (1)$$

where $\tau$ is the dispersion parameter, and $\nu$ is some known base measure (e.g., Lebesgue, counting). We assume the following:

**Assumption 1.** $\boldsymbol{\theta}_\star \in \Theta \subseteq \mathcal{B}^d(S) := \{\boldsymbol{\theta} \in \mathbb{R}^d : \|\boldsymbol{\theta}\|_2 \leq S\}$ *for some known $S > 0$. Also, $\Theta$ is nonempty, compact, and convex with intrinsic dimension[1] $d$.*

**Assumption 2.** *The domain $X$ for arm (context) $\boldsymbol{x}$ satisfies $X \subseteq \mathcal{B}^d(1)$.*

**Assumption 3.** *$m$ is three times differentiable and convex, i.e., $m'''$ exists and $\dot{\mu} := m'' \geq 0$.*

In the **generalized linear bandit (GLB)** problem, at each time $t \in [T]$, the learner observes a time-varying, arbitrary (adversarial) arm-set $\mathcal{X}_t \subseteq X$, chooses a $\boldsymbol{x}_t \in \mathcal{X}_t$, and receives a reward $r_t \sim p(\cdot|\boldsymbol{x}_t, \boldsymbol{\theta}_\star)$. Let $\mathcal{X}_{[T]} := \cup_{t=1}^T \mathcal{X}_t$ and $\Sigma_{t+1} := \sigma(\Sigma_t, r_t, \boldsymbol{x}_{t+1})$ with $\Sigma_0 = \sigma(\boldsymbol{x}_1)$ be the filtration in the canonical bandit model (Lattimore and Szepesvári, 2020, Chapter 4.6). From well-known properties of GLMs (McCullagh and Nelder, 1989), we have that $\mathbb{E}[r_t|\Sigma_t] = m'(\langle\boldsymbol{x}_t, \boldsymbol{\theta}_\star\rangle) \triangleq \mu(\langle\boldsymbol{x}_t, \boldsymbol{\theta}_\star\rangle)$ and $\mathrm{Var}[r_t|\Sigma_t] = g(\tau)\dot{\mu}(\langle\boldsymbol{x}_t, \boldsymbol{\theta}_\star\rangle)$, where $\mu$ is the *inverse link function*. We also define the following quantity describing the maximum slope of $\mu$: $R_{\dot{\mu}} := \max_{\boldsymbol{x} \in \mathcal{X}_{[T]}, \boldsymbol{\theta} \in \Theta} \dot{\mu}(\langle\boldsymbol{x}, \boldsymbol{\theta}\rangle)$.

Note that many common distributions, such as Gaussian ($\mu(z) = z$, $R_{\dot{\mu}} = 1$), Poisson ($\mu(z) = e^z$, $R_{\dot{\mu}} = e^S$), and Bernoulli ($\mu(z) = (1 + e^{-z})^{-1}$, $R_{\dot{\mu}} = 1/4$), fall under the umbrella of GLM.

---

[1] the linear-algebraic dimension (minimum number of basis vectors spanning it) of the affine span of $\Theta$ in $\mathbb{R}^d$.

# 3 Unified Likelihood Ratio-based Confidence Sequence for GLMs

The learner's goal is to output a time-uniform confidence sequence (CS) for $\boldsymbol{\theta}_\star$, $\mathbb{P}[\exists t \geq 1 : \boldsymbol{\theta}_\star \notin \mathcal{C}_t(\delta)] \leq \delta$, where $\mathbb{P}$ is w.r.t. the randomness of the confidence sets $\mathcal{C}_t(\delta)$. In this work, we are particularly interested in the log-likelihood-based confidence set "centered" at the *norm-constrained*, batch maximum likelihood estimator (MLE):

$$\mathcal{C}_t(\delta) := \left\{ \boldsymbol{\theta} \in \Theta : \mathcal{L}_t(\boldsymbol{\theta}) - \mathcal{L}_t(\widehat{\boldsymbol{\theta}}_t) \leq \beta_t(\delta)^2 \right\}, \tag{2}$$

where $\beta_t(\delta)^2$ is the "radius" of the CS that we will define later, $\mathcal{L}_t(\boldsymbol{\theta})$ is the negative log-likelihood of $\boldsymbol{\theta}$ w.r.t. data collected up to $t - 1$, and $\widehat{\boldsymbol{\theta}}_t$ is the corresponding MLE:

$$\mathcal{L}_t(\boldsymbol{\theta}) := \sum_{s=1}^{t-1} \left\{ \ell_s(\boldsymbol{\theta}) \triangleq \frac{-r_s \langle \boldsymbol{x}_s, \boldsymbol{\theta} \rangle + m(\langle \boldsymbol{x}_s, \boldsymbol{\theta} \rangle)}{g(\tau)} \right\}, \quad \widehat{\boldsymbol{\theta}}_t := \arg\min_{\boldsymbol{\theta} \in \Theta} \mathcal{L}_t(\boldsymbol{\theta}). \tag{3}$$

Note that $h(r_s, \tau)$ is omitted as it plays no role in the confidence set nor the MLE.

The form of the confidence set is the same as Lee et al. (2024) and convex relaxation of Abeille et al. (2021), all of which utilizes a single, cumulative & *constrained* MLE $\widehat{\boldsymbol{\theta}}_t \in \Theta$ to compute the loss at time $t$. Other approaches include using a single regularized MLE $\widehat{\boldsymbol{\theta}}_t$ that may lie outside of $\Theta$ (Abbasi-Yadkori et al., 2011), using a sequence of MLEs $\{\widehat{\boldsymbol{\theta}}_s\}_{s=1}^t$ to compute the loss at time $t$ (Abbasi-Yadkori et al., 2012; Emmenegger et al., 2023; Faury et al., 2022; Jun et al., 2017; Wasserman et al., 2020), and computing the *expected* loss over some distribution (e.g., Gaussian) without committing to point estimators (Flynn et al., 2023). As one can see later, our derivation of the CS resembles the last approach: we also start from an expectation of loss over a prior distribution of $\boldsymbol{\theta}$ without committing to an estimator. Yet, we introduce a single estimator $\widehat{\boldsymbol{\theta}}_t$ to avoid the computational difficulty of evaluating the expectation.

Our first main contribution is the following unified confidence sequence for *any* GLMs, regardless of whether it is bounded or not, as long as the corresponding log-likelihood loss is Lipschitz:

---

**Theorem 3.1** (Unified CS for GLMs). *Let $L_t$ be the Lipschitz constant[a] of $\mathcal{L}_t(\cdot)$ that may depend on $\{(\boldsymbol{x}_s, r_s)\}_{s=1}^{t-1}$. Then, we have $\mathbb{P}[\exists t \geq 1 : \boldsymbol{\theta}_\star \notin \mathcal{C}_t(\delta)] \leq \delta$, where*

$$\beta_t(\delta)^2 = \log \frac{1}{\delta} + \inf_{c_t \in (0,1]} \left\{ d \log \frac{1}{c_t} + 2SL_t c_t \right\} \leq \log \frac{1}{\delta} + d \log \left( e \vee \frac{2eSL_t}{d} \right),$$

*where the last inequality follows from the choice $c_t = 1 \wedge \frac{d}{2SL_t}$.*

---
[a]If $\mathcal{L}_t$ is differentiable, one could apply the **Rademacher's theorem** (Federer, 1996, Theorem 3.1.6): $L_t := \inf \left\{ L \geq 0 : |\mathcal{L}_t(\boldsymbol{\theta}) - \mathcal{L}_t(\boldsymbol{\theta}')| \leq L \left\| \boldsymbol{\theta} - \boldsymbol{\theta}' \right\|_2, \forall \boldsymbol{\theta}, \boldsymbol{\theta}' \in \Theta \right\} = \max_{\boldsymbol{\theta} \in \Theta} \left\| \nabla \mathcal{L}_t(\boldsymbol{\theta}) \right\|_2$.

---

**Remark 1** (Generality of our Unfied CS). *The above holds for any distribution over any Polish space, although $\mathcal{C}_t(\delta)$ is convex if and only if $\mathcal{L}_t$ is convex. For GLMs, convexity is guaranteed.*

Practically, the computation of $L_t$ involves a potentially non-concave maximization over a convex set, which is NP-hard in general (Murty and Kabadi, 1987). In Table 1, we provide *closed-form* (up to absolute constants), high-probability upper bounds for $L_t$'s for various GLMs. Note that for the learner to implement the CS, she also needs to know $S$, or its upper bound.

**Comparisons to Prior Works.** Lai (1976) derived the first generic CS for the exponential family based on a generalized likelihood ratio, but it is only applicable for $\Theta \subset \mathbb{R}$ and is hard to instantiate. Recently, several works have provided CSs for either generic GLMs (Emmenegger et al., 2023; Jun et al., 2017; Li et al., 2017) or specific GLMs (linear: Abbasi-Yadkori et al. (2011); Flynn et al. (2023), logistic: Abeille et al. (2021); Faury et al. (2020); Lee et al. (2024)). The generic CSs are generally not tight as the "radius" often scales with $\kappa := \max_{\boldsymbol{x} \in X, \boldsymbol{\theta} \in \Theta} \dot{\mu}(\langle \boldsymbol{x}, \boldsymbol{\theta} \rangle)^{-1}$, which scales exponentially in $S$ for Bernoulli (Faury et al., 2020). For instance, Theorem 1 of Jun et al. (2017) and Theorem 1 of Li et al. (2017) propose ellipsoidal CSs that provably satisfy $\|\widehat{\boldsymbol{\theta}}_t - \boldsymbol{\theta}_\star\|_{V_t}^2 \leq \zeta_1(t, \delta)$, with $\zeta_1$ always scaling with $\kappa$. Emmenegger et al. (2023) proposed a weighted sequential likelihood

Table 1: Instantiations of $L_t$'s for various GLMs. "Bounded by $M$" means for any $\boldsymbol{x} \in X$ and $r \sim p(\cdot | \boldsymbol{x}, \boldsymbol{\theta}_\star)$, the following holds almost surely: $|r - \mu(\langle \boldsymbol{x}, \boldsymbol{\theta}_\star \rangle)| \leq M < \infty$.

| GLM | Upper bounds for $L_t$ | Proof |
|---|---|---|
| Bounded by $M$ | $(M + 2SR_{\dot{\mu}})(t-1)/g(\tau)$ | Appendix C.1 |
| Bernoulli | $(1 + S/2)(t-1)$ | $M = 1, R_{\dot{\mu}} = 1/4$ |
| $\sigma$-subGaussian* | $\sigma^{-2}\left(St + \sigma\sqrt{t \log \frac{d}{\delta}}\right)$ | Appendix C.2 |
| Poisson* | $e^S t + \log \frac{d}{\delta}$ | Appendix C.3 |

\* The omitted absolute constants can be found in the respective proofs.

testing-based CS $\mathcal{W}_t$ and showed its efficacy empirically. Theoretically, they showed that $\boldsymbol{\theta} \in \mathcal{W}_t$ satisfies $D(\boldsymbol{\theta}, \boldsymbol{\theta}_\star) \leq \zeta_2(t, \delta)$ for some Bregman divergence $D(\cdot, \cdot)$ and a $\zeta_2$ always scaling with $\kappa$ as well. We believe their relaxation is not tight enough to warrant a fair comparison and leave to future work on theoretically comparing our CS to theirs. Chowdhury et al. (2023) proposed Bregman divergence-based CSs for generic exponential families, which are quite closely related to our CS; see Appendix A for further discussions. On the other hand, the CSs for specific GLMs are inapplicable to GLM models beyond what they are designed for and may not even be sufficiently tight. The prior state-of-the-art (likelihood ratio-based) CS radius for Bernoulli is $\mathcal{O}\left(S \log(1/\delta) + d \log(St/d)\right)$ of Lee et al. (2024), while our theorem gives us $\mathcal{O}\left(\log(1/\delta) + d \log(St/d)\right)$. Note that we *completely* remove the $\text{poly}(S)$-dependency from the radius, resolving an open problem posited by Lee et al. (2024). Later in Section 4, we show this is significant, both theoretically *and* numerically.

### 3.1 Ellipsoidal Confidence Sequence for Self-Concordant GLMs

We now provide an *ellipsoidal* relaxation of Theorem 3.1 for the following class of GLMs:

**Assumption 4** (Russac et al. (2021)). *GLM is* **(generalized) self-concordant**, *i.e., the following quantity is well-defined (finite):* $R_s := \inf\left\{R \geq 0 : |\ddot{\mu}(\langle \boldsymbol{x}, \boldsymbol{\theta} \rangle)| \leq R\dot{\mu}(\langle \boldsymbol{x}, \boldsymbol{\theta} \rangle), \forall \boldsymbol{x} \in X, \boldsymbol{\theta} \in \Theta\right\}$.

For instance, Bernoulli satisfies this with $R_s = 1$, and more generally, GLM bounded by $R$ a.s. satisfy this assumption with $R_s = R$ (Sawarni et al., 2024, Lemma 2.1). Many unbounded GLMs also satisfy this assumption, such as Gaussian ($R_s = 0$), Poisson ($R_s = 1$), and Exponential ($R_s = 0$).

For such *self-concordant GLMs*, we have the following slightly relaxed ellipsoidal CS, whose proof is deferred to Appendix D:

> **Theorem 3.2** (Ellipsoidal CS for Self-Concordant GLMs). *With the same notations as Theorem 3.1, we have that for any $\lambda \geq 0$, $\mathbb{P}[\exists t \geq 1 : \boldsymbol{\theta}_\star \notin \mathcal{E}_t(\delta, \lambda)] \leq \delta$, where*
>
> $$\mathcal{E}_t(\delta, \lambda) := \left\{\boldsymbol{\theta} \in \mathbb{R}^d : \left\|\boldsymbol{\theta} - \widehat{\boldsymbol{\theta}}_t\right\|_{\nabla^2 \mathcal{L}_t(\widehat{\boldsymbol{\theta}}_t) + \lambda \boldsymbol{I}_d}^2 \leq \gamma_t(\delta) \triangleq 2(1 + SR_s)(4S^2\lambda + \beta_t(\delta)^2)\right\}.$$

Let us denote $A \lesssim B$ if $A \leq cB$ for some absolute constant $c > 0$. Note that the relaxation is order-wise strict only when $R_s > 0$. For instance, for Gaussian where $R_s = 0$, the ellipsoidal relaxation does not introduce additional $S$-dependency when we choose $\lambda = \Theta\left(\frac{1}{S^2}\right)$. We then have that $\nabla^2 \mathcal{L}_t(\widehat{\boldsymbol{\theta}}_t) = \frac{1}{\sigma^2}\sum_{s=1}^{t-1} \boldsymbol{x}_s \boldsymbol{x}_s^\top =: \frac{1}{\sigma^2}\boldsymbol{V}_t$, and $L_t \lesssim St$ with high probability (Proposition C.1). Combining everything, we have $\|\boldsymbol{\theta} - \widehat{\boldsymbol{\theta}}_t\|_{\boldsymbol{V}_t}^2 \lesssim \sigma^2\left(\log(t/\delta) + d\log(St/d)\right)$, which *completely* matches the prior state-of-the-art radius as in Lemma D.10 of Flynn et al. (2023) with $c = \sigma^2 S^2$.

In bandits, the ellipsoidal CS allows one to equivalently rewrite the optimistic optimization in the UCB algorithm (Auer et al., 2002) as a *closed form bonus-based optimization* over the arm-set $\mathcal{X}_t$:

$$\underset{\boldsymbol{x} \in \mathcal{X}_t, \boldsymbol{\theta} \in \mathcal{E}_t(\delta, \lambda)}{\arg\max} \langle \boldsymbol{x}, \boldsymbol{\theta} \rangle = \underset{\boldsymbol{x} \in \mathcal{X}_t}{\arg\max} \langle \boldsymbol{x}, \widehat{\boldsymbol{\theta}}_t \rangle + \sqrt{\gamma_t(\delta)}\|\boldsymbol{x}\|_{(\nabla^2 \mathcal{L}_t(\widehat{\boldsymbol{\theta}}_t) + \lambda \boldsymbol{I}_d)^{-1}}, \quad (4)$$

i.e., there is no need to solve a convex optimization for each arm. In the high-dimensional scenario where $t = o(d)$, one can compute $(\nabla^2 \mathcal{L}_t(\widehat{\boldsymbol{\theta}}_t) + \lambda \boldsymbol{I}_d)^{-1}$ with a time complexity of $\mathcal{O}(td^2)$ per round via the Sherman-Morrison formula (Sherman and Morrison, 1950), which is more efficient than the naïve matrix inversion that takes $\mathcal{O}(d^3)$ time complexity.

## 3.2 Proof of Theorem 3.1 – PAC-Bayes Approach with Uniform Prior

We consider $M_t(\boldsymbol{\theta}) := \exp\left(\mathcal{L}_t(\boldsymbol{\theta}_\star) - \mathcal{L}_t(\boldsymbol{\theta})\right)$, the likelihood ratio between the (estimated) distribution corresponding to $\boldsymbol{\theta}$ and the true distribution corresponding to $\boldsymbol{\theta}_\star$. This has been the subject of study for over 50 years (Darling and Robbins, 1967a,b; Lai, 1976; Robbins and Siegmund, 1972) and recently revisited by statistics and machine learning communities (Emmenegger et al., 2023; Flynn et al., 2023; Ramdas et al., 2023; Wasserman et al., 2020).

We follow the usual recipes for deriving time-uniform PAC-Bayesian bound (Alquier, 2024; Chugg et al., 2023). We start with the following time-uniform property:

**Lemma 3.1.** *Let $\delta \in (0,1)$. For any data-independent probability measure $\mathbb{Q}$ on $\Theta$, we have:*

$$\mathbb{P}\left(\exists t \geq 1 : \mathbb{E}_{\boldsymbol{\theta}\sim\mathbb{Q}}[M_t(\boldsymbol{\theta})] \geq \frac{1}{\delta}\right) \leq \delta, \tag{5}$$

*where $\mathbb{P}$ is over the randomness of the data (and thus randomness of $\mathcal{L}_t$'s).*

*Proof.* First, it is easy to see that $M_t(\boldsymbol{\theta}) = \prod_{s=1}^t \frac{dp(r_s|\boldsymbol{x}_s;\boldsymbol{\theta})}{dp(r_s|\boldsymbol{x}_s;\boldsymbol{\theta}_\star)}$ is a nonnegative martingale w.r.t. $\Sigma_t$:

$$\mathbb{E}[M_t(\boldsymbol{\theta})|\Sigma_{t-1}] = M_{t-1}(\boldsymbol{\theta})\mathbb{E}\left[\frac{dp(r_t|\boldsymbol{x}_t;\boldsymbol{\theta})}{dp(r_t|\boldsymbol{x}_t;\boldsymbol{\theta}_\star)}\Big|\Sigma_{t-1}\right] = M_{t-1}(\boldsymbol{\theta})\underbrace{\int \frac{dp(r|\boldsymbol{x}_t;\boldsymbol{\theta})}{dp(r|\boldsymbol{x}_t;\boldsymbol{\theta}_\star)}dp(r|\boldsymbol{x}_t;\boldsymbol{\theta}_\star)}_{=1}.$$

Now consider the random variable $\mathbb{E}_{\boldsymbol{\theta}\sim\mathbb{Q}}[M_t(\boldsymbol{\theta})]$, which is adapted to $\Sigma_t$. This is a martingale, as

$$\mathbb{E}[\mathbb{E}_{\boldsymbol{\theta}\sim\mathbb{Q}}[M_t(\boldsymbol{\theta})]|\Sigma_{t-1}] \overset{(*)}{=} \mathbb{E}_{\boldsymbol{\theta}\sim\mathbb{Q}}[\mathbb{E}[M_t(\boldsymbol{\theta})|\Sigma_{t-1}]] = \mathbb{E}_{\boldsymbol{\theta}\sim\mathbb{Q}}[M_{t-1}(\boldsymbol{\theta})]$$

where $(*)$ follows from Tonelli's theorem. We conclude by Ville's inequality (Ville, 1939). □

We recall the variational representation of the KL divergence:

**Lemma 3.2** (Theorem 2.1 of Donsker and Varadhan (1983)). *For two probability measures $\mathbb{P}, \mathbb{Q}$ over $\Theta$, we have the following: $D_{\mathrm{KL}}(\mathbb{P}||\mathbb{Q}) = \sup_{g:\Theta\to\mathbb{R}} \mathbb{E}_{\boldsymbol{\theta}\sim\mathbb{P}}[g(\boldsymbol{\theta})] - \log\mathbb{E}_{\boldsymbol{\theta}\sim\mathbb{Q}}[e^{g(\boldsymbol{\theta})}].$*

We then have the following:

**Lemma 3.3.** *For any data-independent prior $\mathbb{Q}$ and any sequence of adapted posterior distributions (possibly learned from the data) $\{\mathbb{P}_t\}$, the following holds: for any $\delta \in (0,1)$,*

$$\mathbb{P}\left(\exists t \geq 1 : \mathcal{L}_t(\boldsymbol{\theta}_\star) - \mathbb{E}_{\boldsymbol{\theta}\sim\mathbb{P}_t}[\mathcal{L}_t(\boldsymbol{\theta})] \geq \log\frac{1}{\delta} + D_{\mathrm{KL}}(\mathbb{P}_t||\mathbb{Q})\right) \leq \delta. \tag{6}$$

*Proof.* Note that

$$\log\mathbb{E}_{\boldsymbol{\theta}\sim\mathbb{Q}}[M_t(\boldsymbol{\theta})] - \mathcal{L}_t(\boldsymbol{\theta}_\star) = \log\mathbb{E}_{\boldsymbol{\theta}\sim\mathbb{Q}}[\exp\left(-\mathcal{L}_t(\boldsymbol{\theta})\right)] \overset{(*)}{\geq} \mathbb{E}_{\boldsymbol{\theta}\sim\mathbb{P}_t}[-\mathcal{L}_t(\boldsymbol{\theta})] - D_{\mathrm{KL}}(\mathbb{P}_t||\mathbb{Q}),$$

where $(*)$ follows from Lemma 3.2 with $g(\cdot) = -\mathcal{L}_t(\cdot)$. By Lemma 3.1, we have that $\mathbb{P}\left(\exists t \geq 1 : \log\frac{1}{\delta} \leq \log\mathbb{E}_{\boldsymbol{\theta}\sim\mathbb{Q}}[M_t(\boldsymbol{\theta})]\right) \leq \delta$. Rearranging gives the desired statement. □

**Remark 2** (Choice of KL). *One can replace KL with other divergences with similar variational formulations (Ohnishi and Honorio, 2021). As we will show later, KL suffices for our purpose.*

Up to now, it is well-known in the PAC-Bayes literature. Our main technical novelty lies in how to choose $\mathbb{Q}$ and $\mathbb{P}_t$, which is as follows: for $c_t \in (0,1]$ to be determined later, we set

$$\mathbb{Q} = \mathrm{Unif}(\Theta), \quad \mathbb{P}_t = \mathrm{Unif}(\widetilde{\Theta}_t \triangleq (1-c_t)\widehat{\boldsymbol{\theta}}_t + c_t\Theta), \tag{7}$$

where $\mathrm{Unif}(\cdot)$ is the uniform distribution and $\boldsymbol{a} + \Theta = \{\boldsymbol{a} + \boldsymbol{\theta} : \boldsymbol{\theta} \in \Theta\}$ for a vector $\boldsymbol{a} \in \mathbb{R}^d$.

Then, denoting $\mathrm{vol}(\cdot)$ as the (Lebesgue) volume in $\mathbb{R}^d$, we have

$$D_{\mathrm{KL}}(\mathbb{P}_t||\mathbb{Q}) = \log\frac{\mathrm{vol}(\Theta)}{\mathrm{vol}(\widetilde{\Theta})} = \log\frac{\mathrm{vol}(\Theta)}{\mathrm{vol}\left((1-c_t)\widehat{\boldsymbol{\theta}}_t + c_t\Theta\right)} = \log\frac{\mathrm{vol}(\Theta)}{\mathrm{vol}(c_t\Theta)} = \log\frac{\mathrm{vol}(\Theta)}{c_t^d\mathrm{vol}(\Theta)} = d\log\frac{1}{c_t}.$$

**Algorithm 1:** OFUGLB

---
**1** **for** $t = 1, 2, \cdots$ **do**
**2**     Compute the norm-constrained MLE: $\widehat{\boldsymbol{\theta}}_t \leftarrow \arg\max_{\boldsymbol{\theta} \in \Theta} \mathcal{L}_t(\boldsymbol{\theta})$;
**3**     Update the confidence set $\mathcal{C}_t$ as specified in Theorem 3.1;
**4**     UCB step: $(\boldsymbol{x}_t, \boldsymbol{\theta}_t) \leftarrow \arg\max_{\boldsymbol{x} \in \mathcal{X}_t, \boldsymbol{\theta} \in \mathcal{C}_t} \langle \boldsymbol{x}, \boldsymbol{\theta} \rangle$;
**5**     Pull the arm $\boldsymbol{x}_t$ and receive a reward $r_t$;

---

We also have that

$$\mathbb{E}_{\boldsymbol{\theta} \sim \mathbb{P}_t}[\mathcal{L}_t(\boldsymbol{\theta})] = \mathcal{L}_t(\widehat{\boldsymbol{\theta}}_t) + \mathbb{E}_{\boldsymbol{\theta} \sim \mathbb{P}_t}[\mathcal{L}_t(\boldsymbol{\theta}) - \mathcal{L}_t(\widehat{\boldsymbol{\theta}}_t)] \leq \mathcal{L}_t(\widehat{\boldsymbol{\theta}}_t) + 2SL_t c_t,$$

where follows from the Lipschitzness of $\mathcal{L}_t(\cdot)$ and the fact that for $\boldsymbol{\theta} = (1 - c_t)\widehat{\boldsymbol{\theta}}_t + c_t \tilde{\boldsymbol{\theta}} \in \widetilde{\Theta}_t$, $\left\| \boldsymbol{\theta} - \widehat{\boldsymbol{\theta}}_t \right\|_2 = c_t \left\| \tilde{\boldsymbol{\theta}} - \widehat{\boldsymbol{\theta}}_t \right\|_2 \leq 2Sc_t$. We conclude by minimizing over $c_t \in (0, 1]$. $\qquad\square$

### 3.3 Intuitions Behind the Proof of Theorem 3.1

**Constrained MLE and Uniform Prior/Poster.** As we consider **constrained MLE**, we know that $\widehat{\boldsymbol{\theta}}_t \in \Theta$, i.e., our "belief" on our MLE is precisely the prior $\mathbb{Q} = \mathrm{Unif}(\Theta)$. Then, as we want the true parameter $\boldsymbol{\theta}_\star$ to be close to $\widehat{\boldsymbol{\theta}}_t$, we want to show that a sufficiently large "posterior volume" is near $\widehat{\boldsymbol{\theta}}_t$, formalized as $\mathbb{P}_t = \mathrm{Unif}\left((1 - c_t)\widehat{\boldsymbol{\theta}}_t + c_t \Theta\right)$ for some time-dependent shrinkage factor $c_t \in (0, 1]$. We later appropriately choose $c_t$ to optimize the PAC-Bayesian bound.

We remark that the uniform prior/posterior has been previously considered in universal portfolios (Blum and Kalai, 1999, Theorem 1) and fast rates in online learning (Foster et al., 2018; Hazan et al., 2007); see Appendix A for discussions on relations to fast rates literature. To our knowledge, we are the first to use such uniform prior/posterior in the (time-uniform) PAC-Bayes context.

**Remark 3** (Use of Regularized MLE?). *When one uses regularized MLE instead of constrained, as it is not guaranteed to be in $\Theta$, one cannot directly use the same uniform prior/posterior. One approach may be to appropriately project the regularized MLE onto $\Theta$ (e.g., Eqn. (9) of Faury et al. (2020)). However, the previously considered projections that guarantee the tightness of the resulting CS involve a nonconvex optimization and are, thus, computationally intractable. One could also consider using high regularization, which may result in additional dependencies on $S$ in the final CS radius. We conjecture that similarly tight guarantees can be recovered with regularized MLE if one uses other appropriate prior/posterior whose supports are the entire $\mathbb{R}^d$ (e.g., Gaussian).*

**Relations to Theorem 3 of Foster et al. (2018).** Let us first briefly recall its proof. The authors first consider a distribution $P_t(\cdot)$ over the parameter $W \in \mathcal{W}$ (see their Algorithm 1) and use $\eta$-mixability of the logistic loss to obtain an inequality involving the negative-log-integral term $\int_{\mathcal{W}} \exp\left(-\eta \sum_t \ell(Wx_t, y_t)\right) dW$. They define $S = \theta W^\star + (1 - \theta)\mathcal{W} \subseteq \mathcal{W}$, where $W^\star$ is the ground-truth optimal parameter and $\theta \in [0, 1)$ is to be determined later. The proof concludes by chaining $\int_{\mathcal{W}} \geq \int_S$ with the $\ell_\infty$-Lipschitzness of the logistic loss and expanding the integral.

Our proof is inspired by the above, with some key differences. While the negative-log-integral also arises in our scenario, we adopt a more compact, streamlined PAC-Bayes approach. In our case, a similar quantity $\mathbb{E}_{\theta \sim \mathbb{Q}}[\exp(-\mathcal{L}_t(\theta))]$ arises from our Donsker-Varadhan representation (Lemma 3.2). We then apply Ville's inequality to obtain the time-uniform PAC-Bayes bound (Lemma 3.1), and our choices of prior/posterior resemble their choice of $S$. Our Lipschitzness argument at the end also resembles their $\ell_\infty$-Lipschitzness argument.

## 4 OFUGLB: A Generic UCB Algorithm for Self-Concordant GLBs

As a direct application of our CS, we consider self-concordant **GLB** (Filippi et al., 2010; Janz et al., 2024), where at each time $t$, the learner chooses a $\boldsymbol{x}_t \in \mathcal{X}_t$ dependent on the history $\{(\boldsymbol{x}_s, r_s)\}_{s=1}^{t-1}$ and receives $r_t \sim p(\cdot | \boldsymbol{x}_t, \boldsymbol{\theta}_\star)$. The learner's goal is to minimize the (pseudo-)regret, $\mathsf{Reg}(T) := \sum_{t=1}^{T} \left\{ \mu(\langle \boldsymbol{x}_{t,\star}, \boldsymbol{\theta}_\star \rangle) - \mu(\langle \boldsymbol{x}_t, \boldsymbol{\theta}_\star \rangle) \right\}$, where $\boldsymbol{x}_{t,\star} := \arg\max_{\boldsymbol{x} \in \mathcal{X}_t} \mu(\langle \boldsymbol{x}, \boldsymbol{\theta}_\star \rangle)$.

Inspired by the optimism principle (Abbasi-Yadkori et al., 2011; Auer, 2002), based on our new, improved confidence sequence (Theorem 3.1), we propose OFUGLB (Algorithm 1), a generic UCB-type algorithm that applies to *any* instantiations of **GLB**. Through a new proof technique that allows us to circumvent $\kappa$- and poly($S$)-dependencies in the leading term, our unified algorithm attains or improves the known state-of-the-art regret bound for the class of *self-concordant* **GLB**, which encompasses a zoo of well-studied stochastic bandits such as linear (Abbasi-Yadkori et al., 2011; Auer, 2002), Poisson (Gisselbrecht et al., 2015), logistic (Abeille et al., 2021; Faury et al., 2020), etc.

We define the following problem difficulty quantities: recalling that $\mathcal{X}_{[T]} = \bigcup_{t \in [T]} \mathcal{X}_t$,

$$\kappa_\star(T) := \frac{1}{\frac{1}{T} \sum_{t \in [T]} \dot{\mu}(\langle \boldsymbol{x}_{t,\star}, \boldsymbol{\theta}_\star \rangle)}, \quad \kappa(T) := \max_{\boldsymbol{x} \in \mathcal{X}_{[T]}, \boldsymbol{\theta} \in \Theta} \frac{1}{\dot{\mu}(\langle \boldsymbol{x}, \boldsymbol{\theta} \rangle)}. \tag{8}$$

These may scale exponentially in $S$, e.g., for logistic bandits (Faury et al., 2020; Filippi et al., 2010), but we will later show that through our new analysis, the leading term of the regret scales *inversely* with $\kappa_\star(T)$, and the transient term scales linearly with $\kappa(T)$.

We now present the *unified & state-of-the-art* regret guarantee for self-concordant **GLB**s:

---

**Theorem 4.1** (OFUGLB for Self-Concordant **GLB**). *OFUGLB attains the following regret bound for self-concordant* **GLB** *with probability at least* $1 - \delta$:

$$\mathsf{Reg}(T) \lesssim \underbrace{d \sqrt{\frac{g(\tau)T}{\kappa_\star(T)} \log \frac{SL_T}{d} \log \frac{R_{\dot{\mu}}ST}{d}}}_{leading\ term} + \underbrace{d^2 R_s R_{\dot{\mu}} \sqrt{g(\tau)} \kappa(T) \log \left( 1 + \frac{ST}{dg(\tau)\kappa(T)} \right)}_{transient\ term},$$

*where $L_T$ is as defined in Theorem 3.1 and we assume that* $\log \frac{1}{\delta} = \mathcal{O}\left( d \log \frac{SL_T}{d} \right)$.

---

## 4.1 Proof Sketch of Theorem 4.1 – Regret Analysis of OFUGLB

We first emphasize that even though we have a tight CS (Theorem 3.1), naïvely combining it with existing regret analyses of logistic bandits (Abeille et al., 2021; Lee et al., 2024) *still* results in an extra factor of $S$ in the leading term. The prior proof applies the Cauchy-Schwartz inequality w.r.t. the (regularized) Hessian $\boldsymbol{H}_t(\boldsymbol{\theta}_\star) = \lambda \boldsymbol{I} + \sum_{s=1}^{t-1} \dot{\mu}_s(\boldsymbol{\theta}_\star) \boldsymbol{x}_s \boldsymbol{x}_s^\top$ with $\dot{\mu}_s(\cdot) := \dot{\mu}(\langle \boldsymbol{x}_s, \cdot \rangle)$, which forces the use of self-concordant lemma (Abeille et al., 2021, Lemma 8). This results in a CS of the form $\|\boldsymbol{\theta}_\star - \widehat{\boldsymbol{\theta}}_t\|_{\boldsymbol{H}_t(\boldsymbol{\theta}_\star)} = \mathcal{O}(S\beta_t(\delta))$. Then, using the same regret decomposition of Abeille et al. (2021) and the optimism principle, the leading term of the regret is bounded as

$$\sum_t \dot{\mu}_t(\boldsymbol{\theta}_\star)\langle \boldsymbol{x}_{t,\star} - \boldsymbol{x}_t, \boldsymbol{\theta}_\star \rangle \lesssim S\beta_t(\delta) \underbrace{\sqrt{\sum_t \dot{\mu}_t(\boldsymbol{\theta}_\star)}}_{\leq \sqrt{T/\kappa_\star(T)}} \underbrace{\sqrt{\sum_t \left\| \sqrt{\dot{\mu}_t(\boldsymbol{\theta}_\star)} \boldsymbol{x}_t \right\|_{\boldsymbol{H}_t(\boldsymbol{\theta}_\star)^{-1}}^2}}_{\text{elliptical potential lemma (Abbasi-Yadkori et al., 2011)}} .$$

Our proof begins by applying Cauchy-Schwartz w.r.t. $\widetilde{\boldsymbol{G}}_t(\widehat{\boldsymbol{\theta}}_t)$, derived from the integral remainder in first-order Taylor expansion of $\mathcal{L}_t(\cdot)$ at $\widehat{\boldsymbol{\theta}}_t$. With this, we have that $\|\boldsymbol{\theta}_\star - \widehat{\boldsymbol{\theta}}_t\|_{\widetilde{\boldsymbol{G}}_t(\widehat{\boldsymbol{\theta}}_t)} = \mathcal{O}(\beta_t(\delta))$ (Lemma E.6), avoiding the extra $S$. However, as $\widetilde{\boldsymbol{G}}_t(\widehat{\boldsymbol{\theta}}_t) = \sum_{s=1}^{t-1} \xi(\boldsymbol{x}_s, \widehat{\boldsymbol{\theta}}_t) \boldsymbol{x}_s \boldsymbol{x}_s^\top$ for some well-defined scalar function $\xi_s$, the elliptical potential lemma (as done above) is *not* applicable due to the explicit dependency on $\widehat{\boldsymbol{\theta}}_t$! This difficulty is analogous to the analysis of Logistic-UCB-2 in Faury et al. (2020), where a similar difficulty arose because their improved bonus $\epsilon_{t,2}$ depends on the current estimate of the parameter as well (see their Lemma 4). They circumvent this issue by explicitly modifying the UCB algorithm to incorporate additional constraints on the "admissible log-odds," which leads to a computationally inefficient algorithm.

Notably, we show via a new proof technique that the vanilla UCB can *implicitly* handle those constraints by designating a "worst-case" parameter over all future iterations (Eqn. (21)). We develop many other intriguing results, such as a novel self-concordance lemma that bounds the difference of $\dot{\mu}$'s with that of $\mu$'s times $R_s$ (Lemma E.3). We provide the full proof in Appendix E. □

Table 2: Regret bounds of `OFUGLB` for various self-concordant **GLB**s. Logarithmic factors are omitted to avoid a cognitive overload. Let $\kappa_{\mathcal{X}}(T) := \max_{\boldsymbol{x} \in \cup_{t=1}^T \mathcal{X}_t} \frac{1}{\dot{\mu}(\langle \boldsymbol{x}, \boldsymbol{\theta}_\star \rangle)}$ and $g(\tau) = \mathcal{O}(1)$. Here, "$R$-Bounded" means $|r_t| \leq R$ $a.s.$.

| GLB | Our regret bound | Prior state-of-the-art |
|:---:|:---:|:---:|
| $R$-Bounded | $d\sqrt{\frac{T}{\kappa_\star(T)}} + d^2 R R_{\dot{\mu}} \kappa(T)$ | $d\sqrt{\frac{T}{\kappa_\star(T)}} + d^2 R^5 S^2 \kappa_{\mathcal{X}}(T)$ (Sawarni et al., 2024, Theorem 4.2) |
| Logistic | $d\sqrt{\frac{T}{\kappa_\star(T)}} + d^2 \kappa(T)$ | $d\sqrt{\frac{T}{\kappa_\star(T)}} + d^2 S^2 \kappa_{\mathcal{X}}(T)$ (Sawarni et al., 2024, Theorem 4.2) |
| Linear[a] | $\sigma d\sqrt{T}$ | $\sigma d\sqrt{T}$ (Flynn et al., 2023, Lemma D.10) |
| Poisson | $dS\sqrt{\frac{T}{\kappa_\star(T)}} + d^2 e^{2S} \kappa(T)$ | $d^{3/2}\sqrt{\frac{T}{\kappa_\star(T)}}$ (Janz et al., 2024, Theorem 1)[b] |

[a] We choose $c = \sigma^2 S^2$ in Lemma D.10 of Flynn et al. (2023).
[b] Here, we omit the dependencies on $S$ and the transient term from explicit warmup.

## 4.2 Instantiations and Discussions of Theorem 4.1

In Table 2, we instantiate Theorem 4.1 for various self-concordant **GLB**s. It can be seen that our `OFUGLB` attains state-of-the-art regret guarantees in all considered scenarios, either by achieving (linear) or improving upon (bounded, logistic) the known regret bounds! Note that the instantiation for (sub-)Gaussian linear bandits is meant to be a sanity check because tighter confidence sets are available in Flynn et al. (2023) and Chowdhury et al. (2023, Appendix F).

To our knowledge, only a few works deal with generic, (possibly unbounded) self-concordant **GLB**s. Jun et al. (2017) proposed UCB-style `GLOC` and its variants, which, however, incur regret bounds scaling with $\kappa_\star(T)$ in the leading term. Concurrent with our work, Liu et al. (2024) prove that all **GLB**s with light-tailed base distribution are self-concordant, and propose `OFU-GLB` with regret of $\widetilde{\mathcal{O}}(d\sqrt{T/\kappa_\star(T)} + d\kappa_\star(T))$. Another line of works (Abeille and Lazaric, 2017; Dong et al., 2019; Janz et al., 2024; Kim et al., 2023; Kveton et al., 2020) considers randomized exploration-based algorithms, including Thompson sampling, which we discuss further in Appendix A.

Below, we discuss our results for bounded **GLB**, logistic, and Poisson bandits in-depth.

**Bounded GLB.** The only prior work applicable to general bounded **GLB** is Sawarni et al. (2024), where the authors propose `RS-GLinCB` with regret as in Table 2. Compared to our regret, they are slightly better as their transient term scales as $\kappa_{\mathcal{X}}(T)$ while ours scales as $\kappa(T)$, but we have a much better dependency on $R$ ($R$ vs. $R^5$). Despite this seeming gap, as `RS-GLinCB` relies on an explicit warm-up scheme, our `OFUGLB` is expected to have superior numerical performance as it avoids excessive exploration in the early phase. We will elaborate more on this issue in Section 5. Also, it should be noted that Sawarni et al. (2024) requires a *nonconvex* optimization as a subroutine to obtain $\text{poly}(S)$-free regret (see their Appendix E). Still, `RS-GLinCB` has its advantages in that it only requires $\Omega(\log^2 T)$ switches while we require $\Omega(T)$ switches; it is an interesting open problem whether a lazy variant of `OFUGLB` with same (or better) regret guarantee is possible.

**Logistic Bandits.** Although the logistic bandit is a special case of the bounded **GLB**, the number of prior works and its practical applicability to recommender systems (Li et al., 2010, 2012) and recently RLHF (Das et al., 2024) makes it deserving of separate discussions. We first review the prior works on logistic bandits. Faury et al. (2020) was the first to obtain a regret bound of $\widetilde{\mathcal{O}}(d\sqrt{T} + d^2\kappa(T))$ (up to some dependencies on $S$) that is $\kappa$-free in the leading term. Subsequently, a local minimax regret lower bound of $\Omega((d/S)\sqrt{T/\kappa_\star(T)})$ was proven (Abeille et al., 2021, Theorem 2)[2], suggesting that more nonlinearity helps, and several works have focused on proposing and analyzing algorithms with matching upper bounds. One line of works (Abeille et al., 2021; Lee et al., 2024), including this work, focuses on getting a tight *convex* CS for logistic losses, which then directly gives an OFUL-type algorithm. Abeille et al. (2021) first proposed a likelihood ratio-based CS, albeit somewhat loose

---

[2]In their statement, dependency on $S$ is ignored. By tracking their lower bound proof, one can see that it leads to an extra factor of $1/S$.

in $S$. Lee et al. (2024) proposed a new framework for converting an achievable online learning algorithm to a tighter CS and proposed a UCB algorithm that attains the prior (to this work) state-of-the-art regret bound of $\widetilde{\mathcal{O}}(dS\sqrt{T/\kappa_\star(T)} + R_\mathcal{X}(T))$ with $R_\mathcal{X}(T)$ being arm-set geometry-dependent transient term[3] From a computational perspective, Faury et al. (2022) proposed an online Newton step-based algorithms that attain the regret bound of $\widetilde{\mathcal{O}}(dS\sqrt{T/\kappa_\star(T)} + d^2 S^6 \kappa(T))$ using only $\mathcal{O}(\log t)$ computational cost *and* $\mathcal{O}(1)$ storage per time step; the computational cost was later improved to $\mathcal{O}(1)$ in Zhang and Sugiyama (2023). Another line of works (Mason et al., 2022; Sawarni et al., 2024) proposed algorithms that perform an *explicit* warm-up in the early stages. Thanks to the explicit warmup, both attain regret with $\mathrm{poly}(S)$-free leading term, e.g., $\widetilde{\mathcal{O}}(d\sqrt{T/\kappa_\star(T)} + d^2 S^2 \kappa_\mathcal{X}(T))$ by Sawarni et al. (2024). However, the explicit warmup typically lasts for $\widetilde{\Omega}(\kappa(T))$ or $\widetilde{\Omega}(\kappa_\mathcal{X}(T))$ time steps, resulting in potentially very large initial regret. This is later verified in our logistic bandits experiments. Our OFUGLB is the first purely optimism-based UCB algorithm (no explicit warmup) that attains a $\mathrm{poly}(S)$-free leading term in the regret.

**Poisson Bandits.** Despite its potential to model various real-world problems involving count feedback, Poisson bandits have not been studied often in the literature. Gisselbrecht et al. (2015) was the first to consider contextual Poisson bandits and proposed UCB and optimistic Bayesian-based algorithms (May et al., 2012), but without any regret guarantees. To our knowledge, our Theorem 4.1 provides the first regret bound for the (finite-dimensional) contextual Poisson bandits without reward boundedness assumption. On a related note, Mutný and Krause (2021) consider Poisson bandits with the intensity function in an RKHS. Their linear RKHS formulation is, however, incompatible with our log-linear formulation; see their Appendix A.1 for further discussions.

## 5 Experiments

We perform experiments on logistic bandits to complement the theoretical improvement in our regret bounds and CS. The codes are available in our GitHub repository[4], forked from the previous repository[5] of Faury et al. (2022). Our GitHub provides the unified implementations of all considered algorithms, which we hope will be helpful in future research and benchmarking of logistic bandits. In Appendix G, we provide the missing implementation details and additional experimental results.

**Baselines and Setting.** We compare our OFUGLB (likelihood ratio-based CS; Theorem 3.1) and OFUGLB-e (ellipsoidal CS; Theorem 3.2) to the following five baselines: EMK (Emmenegger et al., 2023), EVILL (Janz et al., 2024), RS-GLinCB (Sawarni et al., 2024), OFULog+ (Lee et al., 2024), and ada-OFU-ECOLog (Faury et al., 2022). Note that the last two are specific to logistic bandits, and RS-GLinCB is specific to bounded **GLB**s. We emphasize that when implementing OFUGLB and OFUGLB-e, we use the precise theoretical hyperparameters as given in our theorem statements without further tuning. To highlight the practical effectiveness of our theoretical algorithms in comparison to randomized exploration, which is known to perform well in practice (Chapelle and Li, 2011; Russo et al., 2018), we use a single, untuned hyperparameter for EVILL[6]. For the experimental setup in this section, we set $T = 10000$, $d = 2$, and $\delta = 0.05$. We consider time-varying arm-set: at each $t \in [T]$, an arm-set $\mathcal{A}_t \subset \mathcal{B}^d(1)$ of size $|\mathcal{A}_t| = 20$ is uniformly sampled. We set $\boldsymbol{\theta}_\star = \frac{S-1}{\sqrt{d}}\mathbf{1}$ for $S \in \{4, 6, 8, 10\}$. Lastly, we consider 10 independent repeats per setting for statistical significance.

**Results and Discussions.** The results are shown in Figure 1. Note that in all considered settings, OFUGLB, EMK, and EVILL outperform every other baseline, both in terms of regret and numerical tightness of the CS. Moreover, for $S \in \{8, 10\}$, our OFUGLB seems to achieve the best performance, although more comprehensive numerical studies would shed more light on this matter. As for our OFUGLB-e, despite having worse performance than OFUGLB, EMK, and EVILL, it always attains better numerical performance than the remaining algorithms. Notably, OFUGLB and EMK achieve at par or better numerical regret compared to EVILL *with heuristic hyperparameter*. This highlights the effectiveness of our theoretical results even compared to heuristically tuned randomized exploration.

---

[3]One may wonder why our Theorem 4.1's transient term always scales as $\kappa(T)$. See Appendix B for further discussions on this interesting difference.

[4]https://github.com/nick-jhlee/logistic_bandit

[5]https://github.com/criteo-research/logistic_bandit

[6]We remark that in Appendix G, we provide additional results (for $K = 10$), where EVILL with its theoretical hyperparameters performs either worst or second-worst.

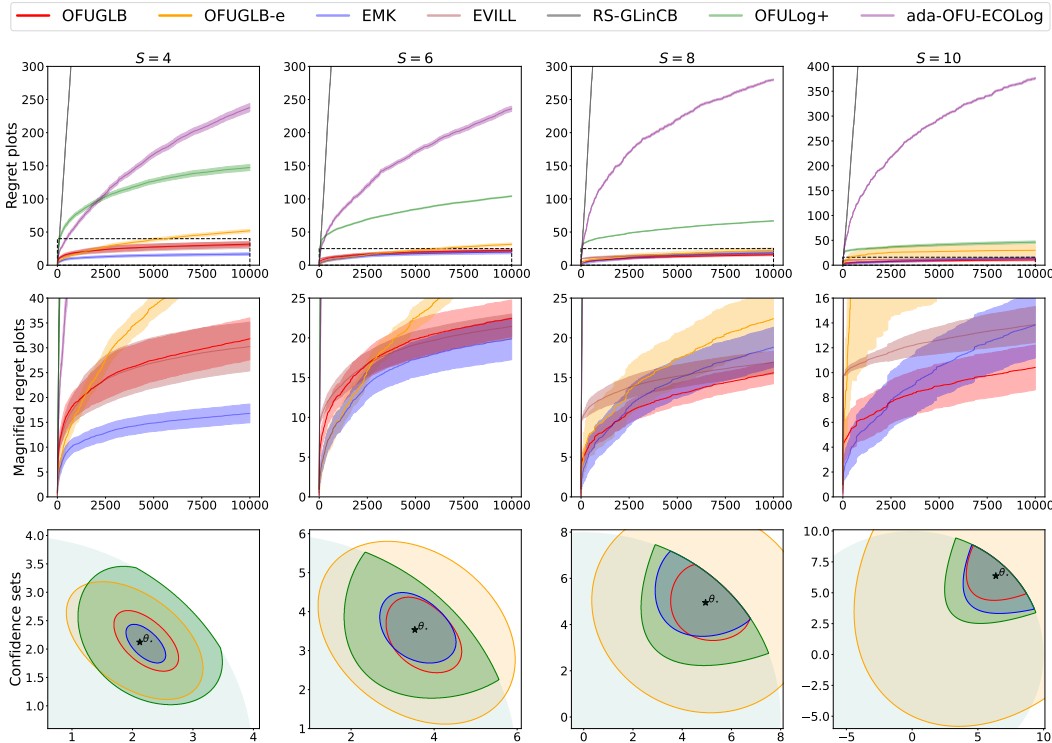

Figure 1: Time-varying arm-sets. (First row) Regret plots of all considered algorithms. (Second row) Magnified regret plots. (Third row) Confidence set plots at the final time $t = 10000$ when applicable. Each column represents a different logistic bandit instance for $S \in \{4, 6, 8, 10\}$.

One interesting observation is that even though `OFULog+` achieves a much tighter CS at the end, its regret is much worse than `OFUGLB-e`. We posit that this is related to the interplay between the CS and arm set geometries, and we leave further study of such discrepancy to future work. Another is that `RS-GLinCB` with the exact theoretical hyperparameters (Sawarni et al., 2024) performs the worst, even worse than `ada-OFU-ECOLog`. We believe this is because their theoretical hyperparameters are not numerically tight, forcing the algorithm to explore throughout the entire duration, probably as their Switching Criterion I (line 4 of their Algorithm 2) is always true. To use `RS-GLinCB` in practice, one must tune[7] the hyperparameters to *explicitly* control the degree of exploration, which is not the case for our `OFUGLB`, making ours a viable, practical algorithm with a *provable guarantee* as well. Lastly, note how the likelihood-based CS, $\{\boldsymbol{\theta} \in \Theta : \mathcal{L}_t(\boldsymbol{\theta}) \leq c_t\}$ for $c_t = \Theta(\log t)$, resembles an ellipsoid. This is because the "normalized" sublevel value $c_t/t$ (as there are $t$ summands in the LHS) gets smaller, making the second-order Taylor expansion more accurate.

## 6 Conclusion

This paper introduces a novel and *unified* likelihood ratio-based CS for generic (convex) GLMs, encompassing widely-used models such as Gaussian, Bernoulli, and Poisson. Our CS is equipped with exact constants for various scenarios, making it suitable for any practitioner. The proof involves leveraging key techniques from PAC-Bayes bounds with a uniform prior/posterior. We then propose `OFUGLB`, a generic UCB algorithm applicable to *any* **GLB**s, achieving state-of-the-art regret bounds across all self-concordant **GLB**s. The proof involves novel regret decomposition and maximally avoiding the self-concordance control lemma (Faury et al., 2020, Lemma 9), which may be of independent interest. Notably, for logistic bandits, `OFUGLB` is the first pure-optimism-based algorithm that achieves $\mathrm{poly}(S)$-free leading term in the theoretical regret, which is numerically verified to perform best. This work opens up various future directions, which we discuss in detail in Appendix B.

---

[7]This is the case in their GitHub implementation, where heuristic values were used for their experiments. Additionally, their algorithm requires the knowledge of $\kappa_\star$.

## Acknowledgments and Disclosure of Funding

J. Lee thanks Gergely Neu for hosting him at a wonderful mini-workshop after AISTATS '24 at UPF, during which many insightful discussions inspired the current PAC-Bayesian proof. J. Lee also thanks Branislav Kveton for suggesting trying a randomized algorithm for logistic bandits experiments, Tim van Erven for insightful discussions regarding the fast rates in statistical learning, and Aaditya Ramdas for insightful comments on the prior-posterior martingale, all during AISTATS '24. J. Lee also thanks Jaeyoung Cha for suggesting an elementary proof for Lemma C.1 that does not rely on Wolfram|Alpha. The authors thank the anonymous reviewers of the ICML '24 ARLET Workshop and NeurIPS '24 for insightful questions and suggestions that helped us significantly improve the paper.

J. Lee and S.-Y. Yun were supported in part by the Institute of Information & Communications Technology Planning & Evaluation (IITP) grant funded by the Korean government (MSIT) (No. RS-2022-II220311 Development of Goal-Oriented Reinforcement Learning Techniques for Contact-Rich Robotic Manipulation of Everyday Objects and No. RS-2019-II190075 Artificial Intelligence Graduate School Program (KAIST)) and the National Research Foundation of Korea(NRF) grant funded by the Korean government (MSIT) (No. RS-2019-NR040050 Stochastic Analysis and Application Research Center (SAARC)). K.-S. Jun was supported in part by the National Science Foundation under grant CCF-2327013.

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

# Contents

# A   Additional Related Works

**Relations to Chowdhury et al. (2023).**   Recently, Chowdhury et al. (2023) proposed two generic CSs for exponential family, one for i.i.d. samples and one for adaptively collected samples. Both CSs are expressed in the local Bregman geometry induced by the log-partition function. The proof relies on the method of mixtures (de la Peña et al., 2004; Kaufmann and Koolen, 2021), which resembles our PAC-Bayesian approach that utilizes a mixture of log-likelihood functions. One drawback is that their main result for i.i.d. samples (Chowdhury et al., 2023, Theorem 3) is instantiated for scalar parameters (e.g., $\mu \in [0, 1]$ for Bernoulli without observed feature vectors), and not for GLMs. While one can attempt to instantiate it to GLMs, we speculate that the resulting confidence set may not be convex since the prior itself is centered at the true parameter, unlike our choice of the prior. While we believe their second method for adaptively collected samples (Chowdhury et al., 2023, Theorem 7) results in a convex set when instantiated to GLMs, the authors do not provide any computationally efficient way to evaluate the integral over the unknown parameter except for the Gaussian GLM. We also mention that contrary to our work, they allow for $\Theta = \mathbb{R}^d$ by introducing a strictly convex regularizer $Z_0$ such that $\int_\Theta \exp(-Z_0(\boldsymbol{\theta}))d\boldsymbol{\theta} < \infty$. Still, theoretically, it is open on whose CS (ours or theirs) is tighter when $\Theta \subseteq \mathcal{B}^d(S)$. We mention in passing that their CS for Gaussian (Chowdhury et al., 2023, Appendix F) improves upon Abbasi-Yadkori et al. (2011) in the same manner (i.e., results in $\sqrt{a + b}$ instead of $\sqrt{a} + \sqrt{b}$) that Flynn et al. (2023) and ours do.

**Fast Rates in Statistical Learning.**   Our goal is to obtain a tight CS for $\boldsymbol{\theta}_\star$, which is quite different from that of statistical learning, which is to obtain the optimal decay rate of the ERM. Although it is not immediately clear, we believe they have a connection. To illustrate our suspicion, we recall Example 10 of Grünwald and Mehta (2020). By taking a uniform prior over a function space $\mathcal{F}$[8] and taking the posterior to be randomly sampling from $\epsilon$-ball centered at $\hat{f}$, the KL term becomes the metric entropy of $\mathcal{F}$, $\log \mathcal{N}(\mathcal{F}, \epsilon)$. Combining this with the Bernstein condition with exponent $\beta$, the ERM obtains the minimax rate of $\widetilde{\mathcal{O}}(n^{-1/(2-\beta)})$, which interpolates between the slow rate $\widetilde{\mathcal{O}}(n^{-1/2})$ and the fast rate $\widetilde{\mathcal{O}}(1/n)$, where $n$ is the number of samples. This is similar to what we obtain by considering discrete uniform prior in our proof; see Appendix F for more details. We also remark that our proof of taking a prior over $\mathcal{L}_t$ resembles improper learning and the $v$-central condition (Foster et al., 2018; van Erven et al., 2015), which also outputs a mixture of predictors to obtain fast rates.

**Randomized Exploration for GLBs.**   Somewhat orthogonal to UCB-based approaches (including ours), another line of works for **GLB**s (Abeille and Lazaric, 2017; Dong et al., 2019; Janz et al., 2024; Kim et al., 2023; Kveton et al., 2020) focuses on randomized exploration-based approaches. Kveton et al. (2020) proposed Thompson sampling and randomly perturbed history-based algorithms, both of which achieved a frequentist regret bound of $\widetilde{\mathcal{O}}(d\kappa\sqrt{T \log K})$ for finite arm-set of size $K$ (Kveton et al., 2020, Theorem 3 & 5). Recently, Janz et al. (2024) proposed EVILL, which *linearly* perturbs the (regularized) log-likelihood loss that achieves a frequentist regret bound[9] of $\widetilde{\mathcal{O}}(d^{3/2}\sqrt{T/\kappa_\star(T)})$ for infinite arm-set (Janz et al., 2024, Theorem 1). In all cases, the extra factor of $\sqrt{d}$ in the regret (due to posterior variance inflation) is shared across the randomized exploration-based approaches to **GLB**s (Abeille and Lazaric, 2017; Dong et al., 2019; Janz et al., 2024; Kim et al., 2023; Kveton et al., 2020) and is known to be unavoidable for linear Thompson sampling (Hamidi and Bayati, 2020). An interesting question is whether the intuitions from our PAC-Bayesian-derived CS can be combined with aggressive variants of Thompson sampling (e.g., `Feel-Good Thompson Sampling` of Li et al. (2024); Zhang (2022) or `TS-UCB` of Baek and Farias (2023)) to improve randomized exploration for **GLB**s.

---

[8]satisfying some regularity conditions including Lipschitzness and boundedness

[9]Although they considered fixed arm-set, their proof can be easily extended to time-varying arm-sets, the only difference being the length of their warm-up period $\tau$ now being dependent on the arm-set distribution.

# B    Future Works

**Extending our CS (Theorem 3.1) to RKHS.**    One may wonder if the framework in this paper can be extended to infinite dimensions, in which the covariate $\boldsymbol{x}$ and unknown $\boldsymbol{\theta}_\star$ are elements of some function space $\mathcal{F}$. Indeed, by exploiting the inner product structure of GLMs, in the case where $\mathcal{F} = \mathcal{H}_k$ is an RKHS with reproducing kernel $k$ (Paulsen and Raghupathi, 2016), the inner product $\langle \boldsymbol{x}, \boldsymbol{\theta}_\star \rangle$ can be replaced with $k(\boldsymbol{x}, \boldsymbol{\theta}_\star)$. In statistics literature, this is referred to as the kernelized or functional GLM (Cawley et al., 2007; Müller and Stadtmüller, 2005), and in bandits literature, this has been extensively studied under the name *kernelized bandits* (Chowdhury and Gopalan, 2017; Srinivas et al., 2010) as an infinite-dimensional generalization of linear bandits. Mussi et al. (2024) recently posited the kernelized logistic bandit (e.g., $r_t \sim \text{Bernoulli}(f(\boldsymbol{x}_t))$ with $f : \mathcal{X} \to [0, 1]$ satisfying $f \in \mathcal{H}_k$) as an important open problem in the bandits and online learning community.

Extending our CS to RKHS, however, raises several issues, all related to the fact that the usual properties of finite-dimensional spaces often fail in infinite dimensions (Bogachev, 1998; Da Prato and Zabczyk, 2014). For instance, it is well-known that there exists *no* translation-invariant, locally finite, non-trivial Borel measure on infinite-dimensional Banach space (Oxtoby, 1946, Theorem 1). Then, it is entirely unclear how to extend our current PAC-Bayesian proof of Theorem 3.1 to infinite dimensions, as there is no uniform distribution or likelihood.

One promising alternate approach based on the Gaussian Process has been recently proposed by Neiswanger and Ramdas (2021), where the authors proposed a CS to quantify the uncertainty of GPs. The important point is that the CS is statistically valid even when the prior is misspecified. Still, in our context, choosing the mean and covariance to obtain similar guarantees (e.g., better dependency on $S' := \|\boldsymbol{\theta}_\star\|_k$) is non-trivial.

**Optimality of CS Radius (Theorem 3.1).**    Another interesting question is whether our CS radius in Theorem 3.1 is optimal. For general time-uniform estimation with i.i.d. samples, Duchi and Haque (2024) recently showed an information-theoretic lower bound of $\Omega\left(\sqrt{\frac{\log \log t}{t}}\right)$ on the estimation error. It would be interesting to use a similar technique to show an information-theoretic lower bound on our CS radius for (self-concordant) GLMs, especially w.r.t. $S$.

**Regret Lower Bound for GLBs.**    One important open question here is the optimality of our obtained regret bound. As discussed in the **Logistic Bandits** paragraph of Section 4, the leading term of our regret bound for *logistic bandits* is (locally) minimax optimal in $d, T, \kappa_\star(T)$ relative to the lower bound of Abeille et al. (2021). A closer look into their proof shows that their lower bound additionally scales as $1/S$, indicating a gap of $S$ between the lower and upper bounds. To the best of our knowledge, there is no generic regret lower bound for self-concordant **GLB**s, and we suspect that a similar $d\sqrt{T/\kappa_\star(T)}$ lower bound holds. One could adapt the proof of Abeille et al. (2021) by modifying parts specific to Bernoulli (e.g., their relative entropy decomposition lemma (Lemma 6) relies on the fact that the reward distribution is Bernoulli), or come up with something new.

**Arm-Set Geometry-Dependent Regret Analyses of OFUGLB.**    For the prior OFUL-type algorithms (Abeille et al., 2021; Lee et al., 2024), the transient term is $R_\mathcal{X}(T) := \sum_{t=1}^T \mu(\langle \boldsymbol{x}_{t,\star}, \boldsymbol{\theta}_\star \rangle) \mathbb{1}[\boldsymbol{x}_t \in \mathcal{X}_-(t)]$, where $\mathcal{X}_-(t)$ is the set of *detrimental arms* with a large reward gap and little information (small conditional variance). $R_\mathcal{X}(T)$ is *adaptive to the arm-set geometry* and can be *completely independent* of $\kappa$ for certain arm geometries (Abeille et al., 2021, Proposition 2). For the warmup-based algorithms (Faury et al., 2022; Mason et al., 2022; Sawarni et al., 2024), the transient term *always* scale with $\kappa$, which is not adaptive to the arm-set geometry.

Abeille et al. (2021) showed that via an arm-set geometry-dependent analysis for UCB, such $\kappa$-scaling transient term can be potentially avoided. However, as our regret analysis utilizes "implicit warmup", our transient term scales with $\kappa(T)$, which *is not* adaptive to the arm-set geometry. Thus, the natural question is whether a similar, arm-set geometry adaptive transient term is attainable for logistic bandits, *while* keeping the optimal $\text{poly}(S)$-free leading term. Currently, it seems that the regret decomposition used in our analysis is *incompatible* with the arm-set geometry-dependent analysis, and we leave to future work for obtaining both characteristics ($\text{poly}(S)$-free leading term, arm-set geometry-dependent transient term) for logistic bandits and **GLB**s in general.

The reason for being *incompatible* is as follows. Even when using our new CS (Theorem 3.1 in the regret analysis of Lee et al. (2024)), one still obtains $\widetilde{\mathcal{O}}(dS\sqrt{T/\kappa_\star(T)} + R_{\mathcal{X}}(T))$, the same as Lee et al. (2024). This is because in their proof of Lemma 6, which involves deriving an ellipsoidal CS of the form $\|\boldsymbol{\theta} - \boldsymbol{\theta}_\star\|_{\boldsymbol{H}_t} \leq \gamma_t(\delta)$, the covering argument introduces a term in $\gamma_t(\delta)$ that *always* scales as $dS^2 \log \frac{St}{d}$, regardless of the likelihood-ratio CS radius. This is unavoidable due to the use of previous self-concordance control (Abeille et al., 2021, Lemma 8), which gives an extra factor of $S$, and the use of anytime Freedman's inequality (Lee et al., 2024, Lemma 3), which results in a multiplicative factor of $1/\eta$ for some $\eta \leq 1/2S$. In other words, attaining the best of both worlds may require thinking of an entirely new regret analysis technique.

**Remark 4** (Detrimental arms for **GLB**s.)**.** *In Abeille et al. (2021), one key component for allowing such transient term that is adaptive to arm-set geometry is that there exists a $\mathcal{Z}_\mu \subseteq \mathbb{R}$ such that $\sup_{z \in \mathcal{Z}_\mu} \ddot{\mu}(z) \leq 0$; e.g., for logistic bandits ($\mu(z) = (1 + e^{-z})^{-1}$), $\mathcal{Z} = (-\infty, 0]$. For general $\mu$, we can define the set of detrimental arms as $\mathcal{X}_-(t) := \{\boldsymbol{x} \in \mathcal{X}_t : \langle \boldsymbol{x}, \boldsymbol{\theta}_\star \rangle \in \mathcal{Z}_\mu\}$. Of course, the scaling of $R_{\mathcal{X}}(T)$ depends on various factors, whose precise characterization for $\mu$'s beyond the logistic function is left for future work.*

**Jointly Efficient and Optimal Algorithm for GLBs.** Despite the statistical superiority of our CS (Theorem 3.1) and our regret bound (Theorem 4.1), it is computationally heavy, especially the UCB maximization (line 6 of Algorithm 1). Our ellipsoidal CS is computationally efficient, but it incurs additional factors of $S$ in the final regret bound. Then the question remains whether one could achieve order-wise the same regret guarantee for **GLB**s (e.g., $\text{poly}(S)$-free for logistic bandits) while significantly improving the computational efficiency. One may, for instance, draw inspiration from recent progress in designing computationally efficient & statistically optimal algorithms for (multinomial) logistic bandits via online Newton steps (Faury et al., 2022; Zhang and Sugiyama, 2023).

**Other Applications.** It would be interesting to see if our new CS may lead to any improvements in algorithms for **GLB**s beyond OFU, e.g., information-directed sampling (Kirschner and Krause, 2018; Russo and Van Roy, 2018), best arm identification in **GLB**s (Azizi et al., 2022; Jun et al., 2021; Kazerouni and Wein, 2021), and even sample-efficient RLHF (Das et al., 2024; Shi et al., 2024).

# C    Missing Proofs from Table 1 – Bounding $L_t$'s

## C.1    GLMs that are Bounded by $M$

Recall that the GLM is bounded by $M$ if for any $\boldsymbol{x} \in X$ and $r \sim p(\cdot|\boldsymbol{x}, \boldsymbol{\theta}_\star)$, the following holds almost surely: $|r - \mu(\langle \boldsymbol{x}, \boldsymbol{\theta}_\star \rangle)| \leq M < \infty$.

Then, we have that for any $\boldsymbol{\theta} \in \Theta$,

$$
\begin{aligned}
\left\| \nabla \mathcal{L}(\boldsymbol{\theta}) \right\| &= \frac{1}{g(\tau)} \left\| \sum_{s=1}^{t-1} \left( -r_s + \mu(\langle \boldsymbol{x}_s, \boldsymbol{\theta} \rangle) \right) \boldsymbol{x}_s \right\|_2 \\
&\leq \frac{1}{g(\tau)} \sum_{s=1}^{t-1} \left| r_s - \mu(\langle \boldsymbol{x}_s, \boldsymbol{\theta} \rangle) \right| \left\| \boldsymbol{x}_s \right\|_2 \\
&\leq \frac{1}{g(\tau)} \sum_{s=1}^{t-1} \left( \left| r_s - \mu(\langle \boldsymbol{x}_s, \boldsymbol{\theta}_\star \rangle) \right| + \left| \mu(\langle \boldsymbol{x}_s, \boldsymbol{\theta}_\star \rangle) - \mu(\langle \boldsymbol{x}_s, \boldsymbol{\theta} \rangle) \right| \right) \\
&\leq \frac{1}{g(\tau)} \sum_{s=1}^{t-1} \left( M + R_{\dot{\mu}} |\langle \boldsymbol{x}_s, \boldsymbol{\theta}_\star - \boldsymbol{\theta} \rangle| \right) \\
&\leq \frac{(M + 2SR_{\dot{\mu}})(t-1)}{g(\tau)}.
\end{aligned}
$$

$\square$

## C.2    $\sigma$-subGaussian GLMs

We first recall some definitions:

**Definition C.1.** *A random variable $X \in \mathbb{R}$ is $\sigma$-subGaussian, if $\mathbb{P}(|X - \mathbb{E}[X]| \geq t) \leq 2 \exp\left( -\frac{t^2}{2\sigma^2} \right), \forall t \in \mathbb{R}$.*

**Definition C.2** (Definition 3 of Jin et al. (2019)). *A random vector $\boldsymbol{X} \in \mathbb{R}^d$ is $\sigma$-norm-subGaussian, if $\mathbb{P}(\left\| \boldsymbol{X} - \mathbb{E}[\boldsymbol{X}] \right\|_2 \geq t) \leq 2 \exp\left( -\frac{t^2}{2\sigma^2} \right), \forall t \in \mathbb{R}$.*

Here is the full statement:

**Proposition C.1.** *Suppose the GLM is $\sigma$-subGaussian. Then, for any $\delta \in (0,1)$,*

$$
\mathbb{P}\left( \exists t \geq 1 : L_t > \frac{2}{g(\tau)} \left( R_{\dot{\mu}} S(t-1) + 2\pi\sigma \sqrt{(t-1)\log \frac{\pi^2 d t^2}{3\delta}} \right) \right) \leq \delta. \tag{9}
$$

*Proof.* Here, as $\max_{\boldsymbol{x} \in \mathcal{X}, \boldsymbol{\theta} \in \Theta} |\dot{\mu}(\langle \boldsymbol{x}, \boldsymbol{\theta} \rangle)| \leq R_{\dot{\mu}}$, we have that

$$
\begin{aligned}
L_t &= \frac{1}{g(\tau)} \max_{\boldsymbol{\theta} \in \Theta} \left\| \sum_{s=1}^{t-1} (r_s - \mu(\langle \boldsymbol{x}_s, \boldsymbol{\theta} \rangle)) \boldsymbol{x}_s \right\|_2 \\
&\leq \frac{1}{g(\tau)} \max_{\boldsymbol{\theta} \in \Theta} \left\| \sum_{s=1}^{t-1} (\mu(\langle \boldsymbol{x}_s, \boldsymbol{\theta} \rangle) - \mu(\langle \boldsymbol{x}_s, \boldsymbol{\theta}_\star \rangle)) \boldsymbol{x}_s \right\|_2 + \frac{1}{g(\tau)} \left\| \sum_{s=1}^{t-1} \underbrace{(r_s - \mu(\langle \boldsymbol{x}_s, \boldsymbol{\theta}_\star \rangle)) \boldsymbol{x}_s}_{\triangleq \boldsymbol{y}_s} \right\|_2 \\
&\leq \frac{2R_{\dot{\mu}} S(t-1)}{g(\tau)} + \frac{1}{g(\tau)} \left\| \sum_{s=1}^{t-1} \boldsymbol{y}_s \right\|_2.
\end{aligned}
$$

We now utilize subGaussian concentrations from Jin et al. (2019). First note that $\boldsymbol{y}_s$ is a martingale difference sequence adapted to $\Sigma_s$ and is norm-subGaussian with (conditional) variance $\sigma^2$ be given.

Then, by Corollary 7 of Jin et al. (2019), we have that

$$\mathbb{P}\left( \left\| \sum_{s=1}^{t-1} \boldsymbol{y}_s \right\|_2 \leq 4\pi\sigma\sqrt{(t-1)\log\frac{2d}{\delta}} \right) \geq 1 - \delta, \quad \forall t \geq 1. \tag{10}$$

The exact constant $4\pi$ is not available in Jin et al. (2019), as all the constants are hidden under $c$. This is not useful, especially for practitioners wanting to use the concentration directly. Thus, we tracked the constant from their Corollary 7, the details of which we provide in Lemma C.1.

We then conclude by replacing $\delta$ with $\delta/t^2$ and applying union bound over $t \geq 1$, which yields the Basel sum. $\qquad\square$

**Lemma C.1** (Lemma 2 of Jin et al. (2019); originally Lemma 5.5 of Vershynin (2010)). *For any $\sigma$-norm-subGaussian random vector $\boldsymbol{X}$, we have that $\sup_{p\in\mathbb{N}} p^{-1/2}\left(\mathbb{E}[\|\boldsymbol{X}\|^p]\right)^{1/p} \leq \sqrt{\pi}\sigma$.*

*Proof.* This follows from brute-force computation. First, we have that

$$\mathbb{E}[\|\boldsymbol{X}\|^p] = \int_0^\infty \mathbb{P}[\|\boldsymbol{X}\|^p \geq t]dt = p\int_0^\infty \mathbb{P}[\|\boldsymbol{X}\| \geq t]t^{p-1}dt \leq 2p\int_0^\infty t^{p-1}\exp\left(-\frac{t^2}{2\sigma^2}\right)dt$$

$$= 2^{\frac{p-1}{2}}\sigma^p p\Gamma\left(\frac{p}{2}\right).$$

Let us denote $f(p) := p^{-1/2}\left(\mathbb{E}[\|\boldsymbol{X}\|^p]\right)^{1/p}$ for $p \in \mathbb{N}$.

Then, using well-known properties of the Gamma function, we have that

$$f(2p) = \sigma 2^{\frac{2p-1}{4p}}(2p)^{\frac{1}{2p}-\frac{1}{2}}\left((p-1)!\right)^{\frac{1}{2p}} = \sigma\sqrt{p^{-1}\left(\sqrt{2}p!\right)^{\frac{1}{p}}}$$

and

$$f(2p-1) = \sigma 2^{\frac{2p-2}{2(2p-1)}}(2p-1)^{\frac{1}{2p-1}-\frac{1}{2}}\left(\sqrt{\pi}\frac{(2p-3)!!}{2^{p-1}}\right)^{\frac{1}{2p-1}} = \sigma(2p-1)^{\frac{1}{2p-1}-\frac{1}{2}}\left(\sqrt{\pi}(2p-3)!!\right)^{\frac{1}{2p-1}},$$

where we define $(-1)!! := 1$.

Then, we have that

$$f(2p) \overset{(i)}{<} \sigma\sqrt{p^{-1}(\sqrt{2}p^p)^{\frac{1}{p}}} = \sigma 2^{\frac{1}{4p}} \leq \sigma 2^{\frac{1}{4}},$$

where $(i)$ follows from $p! < p^p$. We also have that

$$f(2p-1) \overset{(i)}{<} \sigma(2p-1)^{\frac{1}{2p-1}-\frac{1}{2}}\left(\sqrt{\pi}(2p-1)^p\right)^{\frac{1}{2p-1}} \overset{(ii)}{<} \sigma\left(\sqrt{\pi}(2p-1)\right)^{\frac{1}{2p-1}} \overset{(iii)}{<} \sigma\sqrt{\pi},$$

where $(i)$ follows from $(2p-3)!! < (2p-1)^p$, $(ii)$ follows from $\frac{p}{2p-1} > \frac{1}{2}$, and $(iii)$ follows from the observations that for $z \geq e$, $f(z) = (\sqrt{\pi}z)^{1/z}$ is decreasing[10], and $f(1) = \sqrt{\pi} > f(3) = (3\sqrt{\pi})^{1/3}$. Finally, as $2^{1/4} < \sqrt{\pi}$, we have that $\sup_{p\in\mathbb{N}} f(p) \leq \sqrt{\pi}\sigma$.

$\qquad\square$

### C.3 Poisson Distribution

We have the following result for Poisson, which may be of independent interest (to our knowledge, this is the first explicit martingale concentration for Poisson in the GLM form):

**Proposition C.2.** *For the Poisson distribution, we have that for any $\delta \in (0, 1)$: when $S > 1$,*

$$\mathbb{P}\left( L_t \leq C(S)(t-1) + \frac{2}{1-2e^{-S}}\log\frac{\pi^2(d+1)t^2}{3\delta} \right) \geq 1 - \delta, \quad \forall t \geq 1, \tag{11}$$

---

[10] $\frac{d}{dz}\log f(z) = \sqrt{\pi}\frac{1-\log z}{z^2} \leq 0, \ \forall z \geq e.$

where $C(S) := \frac{1}{4}(1 - 2e^{-S})(e^S + 2S + 2\log\frac{2(1-2e^{-S})}{e}) + 2Se^S$. When $S \le 1$,

$$\mathbb{P}\left(L_t \le \tilde{C}(S)(t-1) + 4\log\frac{\pi^2(d+1)t^2}{3\delta}\right) \ge 1 - \delta, \quad \forall t \ge 1, \tag{12}$$

where $\tilde{C}(S) := \frac{1}{16}\left(e^S + 4S + 4\log(8 + 2e^S)\right) + 2Se^S$.

*Proof.* Proceeding similarly as in the previous subsection, we first have that

$$L_t \le 2Se^S(t-1) + \left\|\sum_{s=1}^{t-1} \boldsymbol{y}_s\right\|_2, \tag{13}$$

where $\boldsymbol{y}_s = (r_s - e^{\langle \boldsymbol{x}_s, \boldsymbol{\theta}_\star\rangle})\boldsymbol{x}_s$ is the martingale difference sequence satisfying $\mathbb{E}[\boldsymbol{y}_s|\Sigma_s] = \boldsymbol{0}$ as $r_s|\Sigma_s \sim \text{Poi}(\langle \boldsymbol{x}_s, \boldsymbol{\theta}_\star\rangle)$.

We now modify the proof of Corollary 7 of Jin et al. (2019) (which is based upon the celebrated Chernoff-Cramér method) for the Poisson martingale vectors, details of which we provide here for completeness.

First, we consider the following MGF bound of the Poisson distribution whose proof is deferred to the end of this subsection:

**Lemma C.2.** *Suppose that the random vector $\boldsymbol{y}$ is of the form $\boldsymbol{y} = (r - \lambda)\boldsymbol{x}$ for some fixed $\boldsymbol{x} \in \mathcal{B}^d(1)$, $r \sim \text{Poi}(\lambda)$, and $\lambda > 0$. Then, for the Hermitian dilation (Tropp, 2015, Definition 2.1.5) of $\boldsymbol{y}$, $\boldsymbol{Y} := \begin{bmatrix} 0 & \boldsymbol{y}^\top \\ \boldsymbol{y} & \boldsymbol{0} \end{bmatrix}$, we have that $\mathbb{E}e^{\theta\boldsymbol{Y}} \preceq \exp\left(F(\theta, \lambda)\right)\boldsymbol{I}_{d+1}$ for $|\theta| < \frac{1}{2}$, where $F(\theta, \lambda) := \lambda|\theta| + \log(2|\theta|) + \log\left(\frac{e^{-\frac{\lambda}{2}}}{\frac{1}{2} - |\theta|} + \lambda\right)$.*

We also recall the Lieb's trace inequality:

**Theorem C.3** (Theorem 6 of Lieb (1973)). *Let $\boldsymbol{A}$ be a fixed symmetric matrix, and let $\boldsymbol{Y}$ be a random symmetric matrix. Then,*

$$\mathbb{E}\,\text{tr}(\exp(\boldsymbol{A} + \boldsymbol{Y})) \le \text{tr}\exp(\boldsymbol{A} + \log\mathbb{E}e^{\boldsymbol{Y}}) \tag{14}$$

Now let $0 < \theta < \frac{1}{2}$ be fixed, and let us denote $\lambda_s := e^{\langle \boldsymbol{x}_s, \boldsymbol{\theta}_\star\rangle}$ and $\mathbb{E}_s[\cdot] := \mathbb{E}[\cdot|\Sigma_s]$ for $s \le t - 1$. We start by noting that

$$\mathbb{E}\,\text{tr}\exp\left(-\theta^2\boldsymbol{I}_{d+1}\sum_{s=1}^{t-1}F(\theta, \lambda_s) + \theta\sum_{s=1}^{t-1}\boldsymbol{Y}_s\right)$$

$$= \mathbb{E}\left[\mathbb{E}_{t-1}\left[\text{tr}\exp\left(-\theta^2\boldsymbol{I}_{d+1}\sum_{s=1}^{t-1}F(\theta, \lambda_s) + \theta\sum_{s=1}^{t-1}\boldsymbol{Y}_s\right)\right]\right]$$

$$\le \mathbb{E}\left[\text{tr}\exp\left(-\theta^2\boldsymbol{I}_{d+1}\sum_{s=1}^{t-1}F(\theta, \lambda_s) + \theta\sum_{s=1}^{t-2}\boldsymbol{Y}_s + \log\mathbb{E}_{t-1}\left[e^{\theta\boldsymbol{Y}_{t-1}}\right]\right)\right] \quad \text{(Theorem C.3)}$$

$$\le \mathbb{E}\left[\text{tr}\exp\left(-\theta^2\boldsymbol{I}_{d+1}\sum_{s=1}^{t-1}F(\theta, \lambda_s) + \theta\sum_{s=1}^{t-2}\boldsymbol{Y}_s + F(\theta, \lambda_{t-1})\boldsymbol{I}_{d+1}\right)\right]$$

$$\text{(Lemma C.2, } \boldsymbol{A} \preceq \boldsymbol{B} \Rightarrow e^{\boldsymbol{C}+\boldsymbol{A}} \preceq e^{\boldsymbol{C}+\boldsymbol{B}})$$

$$\le \mathbb{E}\left[\text{tr}\exp\left(-\theta^2\boldsymbol{I}_{d+1}\sum_{s=1}^{t-2}F(\theta, \lambda_s) + \theta\sum_{s=1}^{t-2}\boldsymbol{Y}_s\right)\right]$$

$$\leq \cdots \leq \operatorname{tr} \exp(0 \boldsymbol{I}_{d+1}) = d + 1.$$

Thus, for any $\rho \geq 0$,

$$\mathbb{P}\left(\left\|\sum_{s=1}^{t-1} \boldsymbol{y}_s\right\| \geq \theta \sum_{s=1}^{t-1} F(\theta, \lambda_s) + \frac{\rho}{\theta}\right)$$

$$= \mathbb{P}\left(\left\|\sum_{s=1}^{t-1} \boldsymbol{Y}_s\right\| \geq \theta \sum_{s=1}^{t-1} F(\theta, \lambda_s) + \frac{\rho}{\theta}\right)$$

$$\qquad\qquad\qquad (\textstyle\sum_s \boldsymbol{Y}_s \text{ is a rank-2 matrix with eigenvalues } \pm \left\|\sum_s \boldsymbol{y}_s\right\|_2)$$

$$= 2\mathbb{P}\left(\lambda_{\max}\left(\sum_{s=1}^{t-1} \boldsymbol{Y}_s\right) \geq \theta \sum_{s=1}^{t-1} F(\theta, \lambda_s) + \frac{\rho}{\theta}\right) \qquad (\boldsymbol{Y}_s\text{'s are symmetric})$$

$$= 2\mathbb{P}\left(\lambda_{\max}\left(\exp\left(\theta \sum_{s=1}^{t-1} \boldsymbol{Y}_s\right)\right) \geq \exp\left(\theta^2 \sum_{s=1}^{t-1} F(\theta, \lambda_s) + \rho\right)\right)$$

$$\leq 2\mathbb{P}\left(\operatorname{tr}\exp\left(\theta \sum_{s=1}^{t-1} \boldsymbol{Y}_s\right) \geq \exp\left(\theta^2 \sum_{s=1}^{t-1} F(\theta, \lambda_s) + \rho\right)\right)$$

$$\leq 2e^{-\rho}\mathbb{E}\operatorname{tr}\exp\left(-\theta^2 \sum_{s=1}^{t-1} F(\theta, \lambda_s) + \theta \sum_{s=1}^{t-1} \boldsymbol{Y}_s\right) \qquad (\text{Markov's inequality})$$

$$\leq 2(d+1)e^{-\rho}. \qquad\qquad\qquad\qquad\qquad\qquad (\text{Lemma C.2})$$

Finally, by reparametrizing, we have that for any $\delta \in (0, 1)$,

$$\mathbb{P}\left(\left\|\sum_{s=1}^{t-1} \boldsymbol{y}_s\right\| \geq \inf_{\theta \in (0,1/2)}\left\{\theta \sum_{s=1}^{t-1} F(\theta, \lambda_s) + \frac{1}{\theta} \log \frac{2d}{\delta}\right\}\right) \leq \delta, \qquad (15)$$

where we recall that $F(\theta, \lambda) = \lambda\theta + \log(2\theta) + \log\left(\frac{e^{-\frac{\lambda}{2}}}{\frac{1}{2} - \theta} + \lambda\right)$ for $\theta > 0$.

First, when $S > 1$, let us choose $\theta = \frac{1}{2} - e^{-S}$, which is guaranteed to be positive. Noting that $\lambda_s = e^{\langle \boldsymbol{x}_s, \boldsymbol{\theta}_\star \rangle} \leq e^S$, we have

$$F\left(\frac{1}{2} - e^{-S}, \lambda_s\right) \leq e^S\left(\frac{1}{2} - e^{-S}\right) + \log(1 - 2e^{-S}) + \log(2e^S) = \frac{1}{2}e^S + S + \log\frac{2(1 - 2e^{-S})}{e}.$$

Thus, the RHS of Eqn. (15)

$$\frac{(1 - 2e^{-S})(e^S + 2S + 2\log\frac{2(1-2e^{-S})}{e})}{4}(t - 1) + \frac{2}{1 - 2e^{-S}} \log \frac{2(d+1)}{\delta}. \qquad (16)$$

For the case $S \leq 1$, choosing $\theta = \frac{1}{4}$, the RHS becomes

$$\frac{e^S + 4S + 4\log(8 + 2e^S)}{16}(t - 1) + 4\log \frac{2(d+1)}{\delta}. \qquad (17)$$

Finally, we conclude by parametrizing $\delta$ as $\delta/t^2$, applying union bound over $t \geq 1$, and using the Basel sum. $\qquad\square$

*Proof of Lemma C.2.* We first have that

$$\mathbb{E}e^{\theta \boldsymbol{Y}} \overset{(*)}{=} \boldsymbol{I}_{d+1} + \sum_{p=1}^{\infty} \frac{\theta^p \mathbb{E}\boldsymbol{Y}^{2p}}{(2p)!} \preceq \boldsymbol{I}_{d+1} + \sum_{p=1}^{\infty} \frac{\theta^{2p}\mathbb{E}\left\|\boldsymbol{y}\right\|^{2p}}{(2p)!}\boldsymbol{I}_{d+1} = \mathbb{E}\left[\frac{e^{\theta\|\boldsymbol{y}\|} + e^{-\theta\|\boldsymbol{y}\|}}{2}\right]\boldsymbol{I}_{d+1}$$

$$\preceq \mathbb{E}\left[e^{|\theta||r-\lambda|}\right]\boldsymbol{I}_{d+1},$$

where $(*)$ follows from the observation that $\mathbb{E}\boldsymbol{Y}^{2p+1} = \boldsymbol{0}$. We now recall a concentration result for Poisson distribution:

**Lemma C.3** (Theorem 1 of the note by C. Canonne). $\mathbb{P}(|r - y| \geq x) \leq 2e^{-\frac{x^2}{2(\lambda+x)}}$.

Then, we have that

$$\mathbb{E}[e^{|\theta||r-\lambda|}] = \int_0^{\infty} \mathbb{P}(e^{|\theta||r-\lambda|} > k)dk \qquad\qquad (dk \text{ is the Lebesgue measure})$$

$$\leq 1 + \int_1^{\infty} \mathbb{P}(e^{|\theta||r-\lambda|} \geq k)dk$$

$$\leq 2\int_1^{\infty} e^{-\frac{(\log k/|\theta|)^2}{2(\lambda+\log k/|\theta|)}} dk \qquad\qquad (\text{Lemma C.3})$$

$$= 2|\theta| \int_0^{\infty} e^{-\frac{u^2}{2(\lambda+u)}+|\theta|u} du$$

$$= 2|\theta|\left\{\int_{\lambda}^{\infty} e^{-\frac{u^2}{2(\lambda+u)}+|\theta|u}du + \int_0^{\lambda} e^{-\frac{u^2}{2(\lambda+u)}+|\theta|u}du\right\}$$

$$\leq 2|\theta|\left\{\int_{\lambda}^{\infty} e^{-(\frac{1}{2}-|\theta|)u}du + \lambda e^{|\theta|\lambda}\right\} \qquad (\frac{u^2}{2(\lambda+u)} \geq \frac{1}{2}u \text{ for } u \geq \lambda)$$

$$\leq 2|\theta|\left(\frac{1}{\frac{1}{2}-|\theta|}e^{-(\frac{1}{2}-|\theta|)\lambda} + \lambda e^{|\theta|\lambda}\right)$$

$$= \exp\left(F(\theta,\lambda) \triangleq \lambda|\theta| + \log(2|\theta|) + \log\left(\frac{e^{-\frac{\lambda}{2}}}{\frac{1}{2}-|\theta|} + \lambda\right)\right).$$

$\square$

# D  Proof of Theorem 3.2 – Ellipsoidal Confidence Sequence

First, similarly to prior works on logistic bandits (Abeille et al., 2021; Lee et al., 2024), let us define the following quantities:

$$\widetilde{\boldsymbol{G}}_t(\boldsymbol{\theta}, \boldsymbol{\nu}) := \frac{1}{g(\tau)} \sum_{s=1}^{t-1} \tilde{\alpha}_s(\boldsymbol{\theta}, \boldsymbol{\nu}) \boldsymbol{x}_s \boldsymbol{x}_s^\top, \ \ \tilde{\alpha}_s(\boldsymbol{\theta}, \boldsymbol{\nu}) := \int_0^1 (1-v)\dot{\mu}\left(\langle \boldsymbol{x}_s, \boldsymbol{\theta} + v(\boldsymbol{\nu} - \boldsymbol{\theta})\rangle\right) dv.$$

(We will later come back to these quantities in the regret analysis.)

Then, by Taylor's theorem with integral remainder and first-order optimality condition for convex constrained optimization[11], we have that for any $\lambda \geq 0$,

$$\beta_t(\delta)^2 \geq \mathcal{L}_t(\boldsymbol{\theta}) - \mathcal{L}_t(\widehat{\boldsymbol{\theta}}_t) = \underbrace{\langle \nabla \mathcal{L}_t(\widehat{\boldsymbol{\theta}}_t), \boldsymbol{\theta} - \widehat{\boldsymbol{\theta}}_t\rangle}_{\geq 0} + \left\|\boldsymbol{\theta} - \widehat{\boldsymbol{\theta}}_t\right\|_{\widetilde{\boldsymbol{G}}_t(\widehat{\boldsymbol{\theta}}_t, \boldsymbol{\theta})}^2$$

$$\geq \left\|\boldsymbol{\theta} - \widehat{\boldsymbol{\theta}}_t\right\|_{\widetilde{\boldsymbol{G}}_t(\widehat{\boldsymbol{\theta}}_t, \boldsymbol{\theta}) + \lambda \boldsymbol{I}_d}^2 - \lambda \left\|\boldsymbol{\theta} - \widehat{\boldsymbol{\theta}}_t\right\|_2^2$$

$$\geq \left\|\boldsymbol{\theta} - \widehat{\boldsymbol{\theta}}_t\right\|_{\widetilde{\boldsymbol{G}}_t(\widehat{\boldsymbol{\theta}}_t, \boldsymbol{\theta}) + \lambda \boldsymbol{I}_d}^2 - 4S^2\lambda.$$

We conclude by using the self-concordance control for $\widetilde{\boldsymbol{G}}$ (Abeille et al., 2021, Lemma 8), which we recall here:

**Lemma D.1** (A slight extension of Lemma 8 of Abeille et al. (2021)). *Let $\mu$ be increasing ($\dot{\mu} \geq 0$, which is basically Assumption 3) and self-concordant with constant $R_s$ (as in Assumption 4). Let $\mathcal{Z} \subset \mathbb{R}$ be bounded. Then, the following holds for any $z_1, z_2 \in \mathcal{Z}$:*

$$\int_0^1 (1-v)\dot{\mu}(z_1 + v(z_2 - z_1))dv \geq \frac{\dot{\mu}(z_1)}{2 + R_s|z_1 - z_2|}.$$

*This then implies that $\widetilde{\boldsymbol{G}}_t(\boldsymbol{\theta}, \boldsymbol{\nu}) \succeq \frac{1}{2+2SR_s}\nabla^2\mathcal{L}_t(\boldsymbol{\theta})$.*

---

[11]Let $\Theta$ be convex and $f : \Theta \to \mathbb{R}$ be convex and differentiable. Then, $\theta^* \in \arg\min_{\theta \in \Theta} f(\theta)$ if and only if $\langle \nabla f(\theta^*), \nu - \theta^*\rangle \geq 0, \ \ \forall \nu \in \Theta$ (Boyd and Vandenberghe, 2004, Section 4.2.3).

# E    Proof of Theorem 4.1 – Regret Bound of `OFUGLB`

Let us denote $\mu_t(\cdot) := \mu(\langle \boldsymbol{x}_t, \cdot \rangle)$ and $[a, b] := \{a, a+1, \cdots, b\}$ for two integers $a \leq b$. We recall the following quantities:

$$R_{\mu,\star} := \max_{\boldsymbol{x} \in X} |\mu(\langle \boldsymbol{x}, \boldsymbol{\theta}_\star \rangle)|, \quad R_{\dot{\mu}} := \max_{\boldsymbol{x} \in X, \boldsymbol{\theta} \in \Theta} \dot{\mu}(\langle \boldsymbol{x}, \boldsymbol{\theta} \rangle). \tag{18}$$

## E.1    Key Ideas of the Proof

We will first expand upon the proof sketch provided in Section 4.1 of the main text. Recall the UCB strategy: $(\boldsymbol{x}_t, \boldsymbol{\theta}_t) \leftarrow \arg\max_{\boldsymbol{x} \in \mathcal{X}_t, \boldsymbol{\theta} \in \mathcal{C}_t} \langle \boldsymbol{x}, \boldsymbol{\theta} \rangle$.

**Why Prior Proof Technique Fails.**    We first show that even though we have a tight CS (Theorem 3.1), naïvely combining it with existing regret analyses of logistic bandits (Abeille et al., 2021; Lee et al., 2024) *still* results in an extra factor of $S$ in the leading term. To see this, let us first recall the existing analyses.

The prior proof starts by bounding the regret by $\sum_{t=1}^{T} \langle \dot{\mu}_t(\boldsymbol{\theta}_\star) \boldsymbol{x}_t, \boldsymbol{\theta}_t - \boldsymbol{\theta}_\star \rangle$ (which follows from optimism and first-order Taylor expansion), plus a lower-order term that is easy to control. The first term becomes the leading regret we will now focus on. Using the Cauchy-Schwartz inequality w.r.t. the (regularized) Hessian $\boldsymbol{H}_t(\boldsymbol{\theta}_\star) = \lambda \boldsymbol{I} + \nabla^2 \mathcal{L}_t(\boldsymbol{\theta}_\star) = \lambda \boldsymbol{I} + \sum_{s=1}^{t-1} \dot{\mu}_s(\boldsymbol{\theta}_\star) \boldsymbol{x}_s \boldsymbol{x}_s^\top$, each summand is bounded as

$$\dot{\mu}_t(\boldsymbol{\theta}_\star) \langle \boldsymbol{x}_t, \boldsymbol{\theta}_t - \boldsymbol{\theta}_\star \rangle \leq \dot{\mu}_t(\boldsymbol{\theta}_\star) \|\boldsymbol{x}_t\|_{H_t(\boldsymbol{\theta}_\star)^{-1}} \left( \|\boldsymbol{\theta}_t - \widehat{\boldsymbol{\theta}}_t\|_{H_t(\boldsymbol{\theta}_\star)} + \|\boldsymbol{\theta}_\star - \widehat{\boldsymbol{\theta}}_t\|_{H_t(\boldsymbol{\theta}_\star)} \right).$$

The prior proof then uses Taylor expansion (again) and self-concordant control (Abeille et al., 2021, Lemma 8) to obtain $\|\boldsymbol{\theta}_t - \widehat{\boldsymbol{\theta}}_t\|_{H_t(\boldsymbol{\theta}_\star)} = \mathcal{O}\left(S\beta_T(\delta)^2\right)$ from the likelihood-based confidence set $\mathcal{L}_t(\boldsymbol{\theta}_t) - \mathcal{L}_t(\widehat{\boldsymbol{\theta}}_t) \leq \beta_T(\delta)^2$, which introduces a factor of $S$. This then leads to

$$\sum_t \dot{\mu}_t(\boldsymbol{\theta}_\star) \langle \boldsymbol{x}_{t,\star} - \boldsymbol{x}_t, \boldsymbol{\theta}_\star \rangle \lesssim S\beta_T(\delta)^2 \underbrace{\sqrt{\sum_t \dot{\mu}_t(\boldsymbol{\theta}_\star)}}_{\leq \sqrt{T/\kappa_\star(T)}} \underbrace{\sqrt{\sum_t \left\| \sqrt{\dot{\mu}_t(\boldsymbol{\theta}_\star)} \boldsymbol{x}_t \right\|^2_{\boldsymbol{H}_t(\boldsymbol{\theta}_\star)^{-1}}}}_{\text{elliptical potential lemma (Abbasi-Yadkori et al., 2011)}},$$

resulting in a regret whose leading term is *not* $\text{poly}(S)$-free.

**Towards Our Approach.**    To obtain a $\text{poly}(S)$-free leading term in the regret, we maximally avoid the self-concordance lemma (Abeille et al., 2021, Lemma 8). To do this, our proof begins by obtaining an elliptical CS w.r.t. $\widetilde{\boldsymbol{G}}_t(\widehat{\boldsymbol{\theta}}_t)$, derived from the first-order Taylor expansion of $\mathcal{L}_t(\cdot)$ at $\widehat{\boldsymbol{\theta}}_t$. With this, we have that $\|\boldsymbol{\theta}_\star - \widehat{\boldsymbol{\theta}}_t\|_{\widetilde{\boldsymbol{G}}_t(\widehat{\boldsymbol{\theta}}_t)} = \mathcal{O}(\beta_T(\delta)^2)$ (Lemma E.6), avoiding the extra $S$ compared to the prior proof that derives an elliptical CS w.r.t. $\boldsymbol{H}_t(\boldsymbol{\theta}_\star)$.

However, the main difficulty of the proof is that $\widetilde{\boldsymbol{G}}_t(\widehat{\boldsymbol{\theta}}_t)$ is *not* in a suitable form for elliptical potential arguments. To see this clearly, consider the following natural optimistic upper-bound of the instantaneous regret:

$$\mu(\langle \boldsymbol{x}_{t,\star}, \boldsymbol{\theta}_\star \rangle) - \mu_t(\boldsymbol{\theta}_\star) \leq \mu_t(\boldsymbol{\theta}_t) - \mu_t(\boldsymbol{\theta}_\star) \qquad \text{(Optimism)}$$

$$= \mu_t(\boldsymbol{\theta}_t) - \mu_t(\widehat{\boldsymbol{\theta}}_t) - \mu_t(\widehat{\boldsymbol{\theta}}_t) + \mu_t(\boldsymbol{\theta}_\star)$$

$$\leq 2|\mu_t(\boldsymbol{\theta}'_t) - \mu_t(\widehat{\boldsymbol{\theta}}_t)|$$

$$\lesssim \dot{\mu}_t(\widehat{\boldsymbol{\theta}}_t) |\langle \boldsymbol{x}_t, \boldsymbol{\theta}'_t - \widehat{\boldsymbol{\theta}}_t \rangle| + \text{lower order terms}, \qquad \text{(Taylor's theorem)}$$

where $\boldsymbol{\theta}'_t = \arg\max_{\boldsymbol{\theta} \in \mathcal{C}_t} |\mu_t(\boldsymbol{\theta}) - \mu_t(\widehat{\boldsymbol{\theta}}_t)|$. One can then apply the aforementioned Cauchy-Schwarz w.r.t. $\widetilde{\boldsymbol{G}}_t(\widehat{\boldsymbol{\theta}}_t)$ to obtain

$$\dot{\mu}_t(\widehat{\boldsymbol{\theta}}_t) |\langle \boldsymbol{x}_t, \boldsymbol{\theta}'_t - \widehat{\boldsymbol{\theta}}_t \rangle| \leq \dot{\mu}_t(\widehat{\boldsymbol{\theta}}_t) \|\boldsymbol{x}_t\|_{\widetilde{\boldsymbol{G}}_t(\widehat{\boldsymbol{\theta}}_t)^{-1}} \|\boldsymbol{\theta}_t - \widehat{\boldsymbol{\theta}}_t\|_{\widetilde{\boldsymbol{G}}_t(\widehat{\boldsymbol{\theta}}_t)} \lesssim \dot{\mu}_t(\widehat{\boldsymbol{\theta}}_t) \beta_T(\delta) \|\boldsymbol{x}_t\|_{\widetilde{\boldsymbol{G}}_t^{-1}(\widehat{\boldsymbol{\theta}}_t)}.$$

This successfully avoids using previous self-concordant control (Abeille et al., 2021, Lemma 8), and thus seemingly getting closer to obtaining a $\text{poly}(S)$-free regret. Omitting details, the final step is

to sum the above over $t \in [T]$ and apply the elliptical potential lemma (EPL; Abbasi-Yadkori et al. (2011)). EPL is applicable *only* when $\widetilde{\boldsymbol{G}}_t(\widehat{\boldsymbol{\theta}}_t)$ can be written as $\lambda \boldsymbol{I} + \sum_{s=1}^{t-1} \dot{\mu}_s(\widehat{\boldsymbol{\theta}}_s) \boldsymbol{x}_s \boldsymbol{x}_s^\top$ for some $\lambda > 0$. However, as $\widetilde{\boldsymbol{G}}_t(\widehat{\boldsymbol{\theta}}_t) = \sum_{s=1}^{t-1} \xi(\boldsymbol{x}_s, \widehat{\boldsymbol{\theta}}_t) \boldsymbol{x}_s \boldsymbol{x}_s^\top$ for some scalar function $\xi_s$, the EPL is *not* applicable due to the explicit dependency on $\widehat{\boldsymbol{\theta}}_t$, not on $\{\widehat{\boldsymbol{\theta}}_s\}_{s \in [t-1]}$. The most challenging part of our proof development is making EPL applicable to the summation resulting from some decomposition of the (instantaneous) regret while avoiding extra $S$-dependencies.

**Our Approach.** The key insight is that if we could designate a "worst-case" $\overline{\boldsymbol{\theta}}_s$ for each time step $s$ such that $\widetilde{\boldsymbol{G}}_t(\widehat{\boldsymbol{\theta}}_t) \succeq \lambda I + \sum_{s=1}^{t-1} \dot{\mu}(\overline{\boldsymbol{\theta}}_s) x_s x_s^\top =: \boldsymbol{Q}_t$, then we can perform the following:

$$\dot{\mu}_t(\widehat{\boldsymbol{\theta}}_t)|\langle \boldsymbol{x}_t, \boldsymbol{\theta}_t' - \widehat{\boldsymbol{\theta}}_t\rangle| \le \dot{\mu}_t(\overline{\boldsymbol{\theta}}_t)|\langle \boldsymbol{x}_t, \boldsymbol{\theta}_t' - \widehat{\boldsymbol{\theta}}_t\rangle| + |\dot{\mu}_t(\widehat{\boldsymbol{\theta}}_t) - \dot{\mu}_t(\overline{\boldsymbol{\theta}}_t)||\langle \boldsymbol{x}_t, \boldsymbol{\theta}_t' - \widehat{\boldsymbol{\theta}}_t\rangle|$$

where the first term is now be bounded by $\mathcal{O}\left(\dot{\mu}_t(\overline{\boldsymbol{\theta}}_t)\beta_T(\delta)\|\boldsymbol{x}_t\|_{\boldsymbol{Q}_t^{-1}}\right)$. We can now apply the EPL when summing over $t \in [T]$, thanks to the form of $\boldsymbol{Q}_t$. The second term turns out to be a lower order term via our new self-concordant control that doesn't give additional $S$-dependency (Lemma E.3).

We note that this is analogous to the analysis of `Logistic-UCB-2` in Faury et al. (2020), where a similar difficulty arose because their improved bonus $\epsilon_{t,2}$ depends on the current estimate of the parameter as well (see their Lemma 4). They circumvent this issue by explicitly modifying the UCB algorithm to incorporate additional constraints on the "admissible log-odds," which leads to a computationally inefficient algorithm. Indeed, initially, we took a similar approach by either using a confidence set defined as an intersection over all the confidence sets used so far, or by using an additional constraint set $\mathcal{W}_t$ as defined in `Logistic-UCB-2` of Faury et al. (2020). However, either approach significantly increases the computational complexity.

We later discovered that we could resolve the issue without changing the confidence set through an alternate analysis, which is the current proof. Specifically, we consider the following decomposition of the instantaneous regret:

$$\mu(\langle \boldsymbol{x}_{t,\star}, \boldsymbol{\theta}_\star\rangle) - \mu_t(\boldsymbol{\theta}_\star) \le \mu_t(\boldsymbol{\theta}_t) - \mu_t(\boldsymbol{\theta}_\star) \le 2|\mu_t(\boldsymbol{\nu}_t) - \mu_t(\widehat{\boldsymbol{\theta}}_{b(t)})|,$$

where we define $(b(t), \boldsymbol{\nu}_t) := \arg\max_{b \in [t,T], \boldsymbol{\theta} \in \mathcal{C}_b} |\mu_t(\boldsymbol{\theta}) - \mu_t(\widehat{\boldsymbol{\theta}}_b)|$. That is, we are bounding the instantaneous regret by how large the difference can be from the current confidence set *and* how large the difference can be from the future confidence sets. With this, we can then define $\overline{\boldsymbol{\theta}}_t := \arg\min_{\boldsymbol{\theta} \in \bigcup_{b \in [t,T]} \mathcal{C}_b} \dot{\mu}_t(\boldsymbol{\theta})$, which satisfies the aforementioned desired property. This is our main technical novelty that allows for us to bypass all the aforementioned difficulties.

Among the omitted details, we consider a slightly more intricate regret decomposition by considering timesteps in which the "warmup conditions" are satisfied and the remaining term, and we derive a novel self-concordance lemma that bounds the difference of $\dot{\mu}$'s with that of $\mu$'s times $R_s$ (Lemma E.3) that does not incur additional $S$-dependencies. We then utilize the elliptical potential *count* lemma (EPCL; Gales et al. (2022)) for the terms that do not satisfy such conditions, and the remaining terms follow the reasoning as detailed above.

## E.2 Main Proof

We defer the statements and proofs of the supporting lemmas to Appendix E.3, although we will provide relevant context when using those lemmas for the proof's duration. We first define the following crucial quantities that we have discussed in the proof sketch: for $\lambda > 0$ to be chosen later,

$$\overline{\boldsymbol{\theta}}_t := \arg\min_{\boldsymbol{\theta} \in \bigcup_{b \in [t,T]} \mathcal{C}_b} \dot{\mu}_t(\boldsymbol{\theta}), \quad (b(t), \boldsymbol{\nu}_t) := \arg\max_{b \in [t,T], \boldsymbol{\theta} \in \mathcal{C}_b} \left|\mu_t(\boldsymbol{\theta}) - \mu_t(\widehat{\boldsymbol{\theta}}_b)\right|, \tag{19}$$

$$\bar{\boldsymbol{H}}_t := 2g(\tau)\lambda \boldsymbol{I} + \sum_{s=1}^{t-1} \dot{\mu}_s(\overline{\boldsymbol{\theta}}_s) \boldsymbol{x}_s \boldsymbol{x}_s^\top, \quad \boldsymbol{V}_t := 2g(\tau)\kappa(T)\lambda \boldsymbol{I} + \sum_{s=1}^{t-1} \boldsymbol{x}_s \boldsymbol{x}_s^\top, \tag{20}$$

and

$$\tilde{\alpha}_t(\boldsymbol{\theta}, \boldsymbol{\nu}) := \int_0^1 (1-v)\dot{\mu}_t\left(\boldsymbol{\theta} + v(\boldsymbol{\nu} - \boldsymbol{\theta})\right) dv, \quad \widetilde{\boldsymbol{G}}_t(\boldsymbol{\theta}, \boldsymbol{\nu}) := \lambda \boldsymbol{I} + \frac{1}{g(\tau)} \sum_{s=1}^{t-1} \tilde{\alpha}_s(\boldsymbol{\theta}, \boldsymbol{\nu}) \boldsymbol{x}_s \boldsymbol{x}_s^\top. \tag{21}$$

$\bar{\boldsymbol{\theta}}_t$ in the union of future confidence sets, combined with the "warmup conditions" allows for the elliptical potential lemma (EPL; Lemma E.2) and elliptical potential count lemma (EPCL; Lemma E.1) to be directly applicable, avoiding dependencies on $\mathrm{poly}(S)$ and $\kappa$ in the leading term; refer to the expanded proof sketch above for a more detailed explanation of the intuition.

Throughout, let us assume that the event $\{\forall t \geq 1, \; \boldsymbol{\theta}_\star \in \mathcal{C}_t\}$ holds, which is with probability at least $1 - \delta$ by Theorem 3.1.

**Regret Decomposition.** Define the set of timesteps satisfying the "warmup conditions":

$$\mathcal{I}_T := \left\{ t \in [T] : \left( \left\| \sqrt{\dot{\mu}_t(\bar{\boldsymbol{\theta}}_t)} \boldsymbol{x}_t \right\|_{\bar{\boldsymbol{H}}_t^{-1}} \geq 1 \right) \vee \left( \|\boldsymbol{x}_t\|_{\boldsymbol{V}_t^{-1}} \geq 1 \right) \right\}. \tag{22}$$

Then we have the following regret decomposition:

$$\mathrm{Reg}(T) = \sum_{t \in \mathcal{I}_T} \left\{ \mu(\langle \boldsymbol{x}_{t,\star}, \boldsymbol{\theta}_\star \rangle) - \mu(\langle \boldsymbol{x}_t, \boldsymbol{\theta}_\star \rangle) \right\} + \underbrace{\sum_{t \notin \mathcal{I}_T} \left\{ \mu(\langle \boldsymbol{x}_{t,\star}, \boldsymbol{\theta}_\star \rangle) - \mu(\langle \boldsymbol{x}_t, \boldsymbol{\theta}_\star \rangle) \right\}}_{\triangleq \mathrm{Reg}_{\mathcal{I}}(T)}$$

$$\leq 2R_{\mu,\star}|\mathcal{I}_T| + \mathrm{Reg}_{\mathcal{I}}(T)$$

$$\leq 2R_{\mu,\star} \sum_{t \in [T]} \mathbb{1} \left[ \left\| \sqrt{\dot{\mu}_t(\bar{\boldsymbol{\theta}}_t)} \boldsymbol{x}_t \right\|_{\bar{\boldsymbol{H}}_t^{-1}} \geq 1 \right] + 2R_{\mu,\star} \sum_{t \in [T]} \mathbb{1} \left[ \|\boldsymbol{x}_t\|_{\boldsymbol{V}_t^{-1}} \geq 1 \right] + \mathrm{Reg}_{\mathcal{I}}(T)$$

$$\text{(Definition of } \mathcal{I}_T)$$

$$\leq \frac{4dR_{\mu,\star}}{\log 2} \left\{ \log \left( 1 + \frac{R_{\dot{\mu}}}{2\lambda g(\tau)\log 2} \right) + \log \left( 1 + \frac{1}{2\kappa(T)\lambda g(\tau)\log 2} \right) \right\} + \mathrm{Reg}_{\mathcal{I}}(T).$$

$$\text{(EPCL (Lemma E.1))}$$

We now focus on bounding the last term:

$$\mathrm{Reg}_{\mathcal{I}}(T) = \sum_{t \notin \mathcal{I}_T} \left\{ \mu_{t,\star}(\boldsymbol{\theta}_\star) - \mu_t(\widehat{\boldsymbol{\theta}}_t) \right\} + \sum_{t \notin \mathcal{I}_T} \left\{ \mu_t(\widehat{\boldsymbol{\theta}}_t) - \mu_t(\boldsymbol{\theta}_\star) \right\}$$

$$(\mu_t(\cdot) := \mu(\langle \boldsymbol{x}_t, \cdot \rangle), \mu_{t,\star}(\cdot) := \mu(\langle \boldsymbol{x}_{t,\star}, \cdot \rangle))$$

$$\leq \sum_{t \notin \mathcal{I}_T} \left\{ \mu_t(\boldsymbol{\theta}_t) - \mu_t(\widehat{\boldsymbol{\theta}}_t) \right\} + \sum_{t \notin \mathcal{I}_T} \left\{ \mu_t(\widehat{\boldsymbol{\theta}}_t) - \mu_t(\boldsymbol{\theta}_\star) \right\}$$

$$\text{(optimism – line 7 of Algorithm 1)}$$

$$\leq 2 \sum_{t \notin \mathcal{I}_T} \max_{b \in [t,T]} \max_{\boldsymbol{\theta} \in \mathcal{C}_b} \left| \mu_t(\boldsymbol{\theta}) - \mu_t(\widehat{\boldsymbol{\theta}}_b) \right|$$

$$= 2 \sum_{t \notin \mathcal{I}_T} \left| \mu_t(\boldsymbol{\nu}_t) - \mu_t(\widehat{\boldsymbol{\theta}}_{b(t)}) \right|. \tag{Eqn. (19)}$$

Using Taylor's theorem with integral remainder form, we have that for $t \notin \mathcal{I}_T$,

$$\left| \mu_t(\boldsymbol{\nu}_t) - \mu_t(\widehat{\boldsymbol{\theta}}_{b(t)}) \right|$$

$$= \left| \dot{\mu}_t(\widehat{\boldsymbol{\theta}}_{b(t)}) \langle \boldsymbol{x}_t, \boldsymbol{\nu}_t - \widehat{\boldsymbol{\theta}}_{b(t)} \rangle + \int_{\mu_t(\widehat{\boldsymbol{\theta}}_{b(t)})}^{\mu_t(\boldsymbol{\nu}_t)} (\mu_t(\boldsymbol{\nu}_t) - z)\ddot{\mu}_t(z)dz \right|$$

$$\leq \dot{\mu}_t(\widehat{\boldsymbol{\theta}}_{b(t)}) \left| \langle \boldsymbol{x}_t, \boldsymbol{\nu}_t - \widehat{\boldsymbol{\theta}}_{b(t)} \rangle \right| + \langle \boldsymbol{x}_t, \boldsymbol{\nu}_t - \widehat{\boldsymbol{\theta}}_{b(t)} \rangle^2 \int_0^1 (1-v) \left| \ddot{\mu}_t \left( \widehat{\boldsymbol{\theta}}_{b(t)} + v(\boldsymbol{\nu}_t - \widehat{\boldsymbol{\theta}}_{b(t)}) \right) \right| dv$$

$$\text{(triangle inequality, reparametrization)}$$

$$\leq \dot{\mu}_t(\widehat{\boldsymbol{\theta}}_{b(t)}) \left| \langle \boldsymbol{x}_t, \boldsymbol{\nu}_t - \widehat{\boldsymbol{\theta}}_{b(t)} \rangle \right| + R_s \langle \boldsymbol{x}_t, \boldsymbol{\nu}_t - \widehat{\boldsymbol{\theta}}_{b(t)} \rangle^2 \underbrace{\int_0^1 (1-v)\dot{\mu}_t \left( \widehat{\boldsymbol{\theta}}_{b(t)} + v(\boldsymbol{\nu}_t - \widehat{\boldsymbol{\theta}}_{b(t)}) \right) dv}_{=\tilde{\alpha}_{b(t)}(\widehat{\boldsymbol{\theta}}_{b(t)}, \boldsymbol{\nu}_t)}$$

$$\text{(Assumption 4)}$$

$$\leq \dot{\mu}_t(\bar{\boldsymbol{\theta}}_t)\left|\langle \boldsymbol{x}_t, \boldsymbol{\nu}_t - \widehat{\boldsymbol{\theta}}_{b(t)}\rangle\right| + \left|\dot{\mu}_t(\bar{\boldsymbol{\theta}}_t) - \dot{\mu}_t(\widehat{\boldsymbol{\theta}}_{b(t)})\right|\left|\langle \boldsymbol{x}_t, \boldsymbol{\nu}_t - \widehat{\boldsymbol{\theta}}_{b(t)}\rangle\right|$$
$$+ R_s\langle \boldsymbol{x}_t, \boldsymbol{\nu}_t - \widehat{\boldsymbol{\theta}}_{b(t)}\rangle^2\tilde{\alpha}_{b(t)}(\widehat{\boldsymbol{\theta}}_{b(t)}, \boldsymbol{\nu}_t)$$
$$\leq \underbrace{\dot{\mu}_t(\bar{\boldsymbol{\theta}}_t)\left\|\boldsymbol{x}_t\right\|_{\widetilde{\boldsymbol{G}}_{b(t)}(\widehat{\boldsymbol{\theta}}_{b(t)}, \boldsymbol{\nu}_t)^{-1}}\left\|\boldsymbol{\nu}_t - \widehat{\boldsymbol{\theta}}_{b(t)}\right\|_{\widetilde{\boldsymbol{G}}_{b(t)}(\widehat{\boldsymbol{\theta}}_{b(t)}, \boldsymbol{\nu}_t)}}_{\triangleq A_t}$$
$$+ \underbrace{\left|\dot{\mu}_t(\bar{\boldsymbol{\theta}}_t) - \dot{\mu}_t(\widehat{\boldsymbol{\theta}}_{b(t)})\right|\left\|\boldsymbol{x}_t\right\|_{\widetilde{\boldsymbol{G}}_{b(t)}(\widehat{\boldsymbol{\theta}}_{b(t)}, \boldsymbol{\nu}_t)^{-1}}\left\|\boldsymbol{\nu}_t - \widehat{\boldsymbol{\theta}}_{b(t)}\right\|_{\widetilde{\boldsymbol{G}}_{b(t)}(\widehat{\boldsymbol{\theta}}_{b(t)}, \boldsymbol{\nu}_t)}}_{\triangleq B_t}$$
$$+ \underbrace{R_s\left\|\boldsymbol{\nu}_t - \widehat{\boldsymbol{\theta}}_{b(t)}\right\|^2_{\widetilde{\boldsymbol{G}}_{b(t)}(\widehat{\boldsymbol{\theta}}_{b(t)}, \boldsymbol{\nu}_t)}\tilde{\alpha}_{b(t)}(\widehat{\boldsymbol{\theta}}_{b(t)}, \boldsymbol{\nu}_t)\left\|\boldsymbol{x}_t\right\|^2_{\widetilde{\boldsymbol{G}}_{b(t)}(\widehat{\boldsymbol{\theta}}_{b(t)}, \boldsymbol{\nu}_t)^{-1}}}_{\triangleq C_t},$$

(Cauchy-Schwartz inequality)

where $\widetilde{\boldsymbol{G}}$ is as defined in Eqn. (21). As one will see soon, $\sum_t A_t$ is the *leading term*, and $\sum_t (B_t + C_t)$ is the *transient term*.

We bound each sum separately:

**Bounding $\sum_t A_t$.**

$$\sum_{t\notin\mathcal{I}_T} A_t = \sum_{t\notin\mathcal{I}_T} \dot{\mu}_t(\bar{\boldsymbol{\theta}}_t)\left\|\boldsymbol{x}_t\right\|_{\widetilde{\boldsymbol{G}}_{b(t)}(\widehat{\boldsymbol{\theta}}_{b(t)}, \boldsymbol{\nu}_t)^{-1}}\left\|\boldsymbol{\nu}_t - \widehat{\boldsymbol{\theta}}_{b(t)}\right\|_{\widetilde{\boldsymbol{G}}_{b(t)}(\widehat{\boldsymbol{\theta}}_{b(t)}, \boldsymbol{\nu}_t)}$$

$$\leq \sqrt{4\lambda S^2 + \beta_T(\delta)^2}\sum_{t\notin\mathcal{I}_T}\dot{\mu}_t(\bar{\boldsymbol{\theta}}_t)\left\|\boldsymbol{x}_t\right\|_{\widetilde{\boldsymbol{G}}_{b(t)}(\widehat{\boldsymbol{\theta}}_{b(t)}, \boldsymbol{\nu}_t)^{-1}} \qquad (\boldsymbol{\nu}_t \in \mathcal{C}_{b(t)}, \text{Lemma E.6 }(i))$$

$$\leq \sqrt{4\lambda S^2 + \beta_T(\delta)^2}\sqrt{\sum_{t\notin\mathcal{I}_T}\dot{\mu}_t(\bar{\boldsymbol{\theta}}_t)}\sqrt{\sum_{t\notin\mathcal{I}_T}\dot{\mu}_t(\bar{\boldsymbol{\theta}}_t)\left\|\boldsymbol{x}_t\right\|^2_{\widetilde{\boldsymbol{G}}_{b(t)}(\widehat{\boldsymbol{\theta}}_{b(t)}, \boldsymbol{\nu}_t)^{-1}}}$$

(Cauchy-Schwartz inequality)

$$\leq \sqrt{4\lambda S^2 + \beta_T(\delta)^2}\sqrt{\sum_{t\notin\mathcal{I}_T}\dot{\mu}_t(\bar{\boldsymbol{\theta}}_t)}\sqrt{2g(\tau)\sum_{t\notin\mathcal{I}_T}\dot{\mu}_t(\bar{\boldsymbol{\theta}}_t)\left\|\boldsymbol{x}_t\right\|^2_{\bar{\boldsymbol{H}}_t^{-1}}} \qquad (\text{Lemma E.5})$$

$$\leq \sqrt{2g(\tau)\left(4\lambda S^2 + \beta_T(\delta)^2\right)}\sqrt{\sum_{t\notin\mathcal{I}_T}\dot{\mu}_t(\bar{\boldsymbol{\theta}}_t)}\sqrt{\sum_{t\in[T]}\min\left\{1, \dot{\mu}_t(\bar{\boldsymbol{\theta}}_t)\left\|\boldsymbol{x}_t\right\|^2_{\bar{\boldsymbol{H}}_t^{-1}}\right\}}.$$

(Definition of $\mathcal{I}_T$)

Note that now, $\bar{\boldsymbol{H}}_t$ is of the form such that we can use the EPL with $\sqrt{\dot{\mu}_t(\bar{\boldsymbol{\theta}}_t)}\boldsymbol{x}_t$:

$$\sum_{t\notin\mathcal{I}_T} A_t \leq 2\sqrt{dg(\tau)(4\lambda S^2 + \beta_T(\delta)^2)\log\left(1 + \frac{R_{\dot{\mu}}T}{d\lambda}\right)}\sqrt{\sum_{t\notin\mathcal{I}_T}\dot{\mu}_t(\bar{\boldsymbol{\theta}}_t)}. \qquad (\text{EPL (Lemma E.2)})$$

$$\leq 2\sqrt{dg(\tau)(4\lambda S^2 + \beta_T(\delta)^2)\log\left(1 + \frac{R_{\dot{\mu}}T}{d\lambda}\right)}\sqrt{\sum_{t\in[T]}\dot{\mu}_{t,\star}(\boldsymbol{\theta}_\star) + \sum_{t\notin\mathcal{I}_T}\left\{\dot{\mu}_t(\bar{\boldsymbol{\theta}}_t) - \dot{\mu}_{t,\star}(\boldsymbol{\theta}_\star)\right\}}$$

$(\mu_{t,\star}(\cdot) := \mu(\langle \boldsymbol{x}_{t,\star}, \cdot\rangle))$

$$= 2\sqrt{dg(\tau)(4\lambda S^2 + \beta_T(\delta)^2)\log\left(1 + \frac{R_{\dot{\mu}}T}{d\lambda}\right)}\sqrt{\frac{T}{\kappa_\star(T)} + \sum_{t\notin\mathcal{I}_T}\left\{\dot{\mu}_t(\bar{\boldsymbol{\theta}}_t) - \dot{\mu}_{t,\star}(\boldsymbol{\theta}_\star)\right\}}.$$

The last term in the square root is bounded as follows:

$$\sum_{t\notin\mathcal{I}_T}\left\{\dot{\mu}_t(\bar{\boldsymbol{\theta}}_t) - \dot{\mu}_{t,\star}(\boldsymbol{\theta}_\star)\right\} = \sum_{t\notin\mathcal{I}_T}\left\{\dot{\mu}_t(\bar{\boldsymbol{\theta}}_t) - \dot{\mu}_t(\boldsymbol{\theta}_\star)\right\} + \sum_{t\notin\mathcal{I}_T}\left\{\dot{\mu}_t(\boldsymbol{\theta}_\star) - \dot{\mu}_{t,\star}(\boldsymbol{\theta}_\star)\right\}$$

$$\leq R_s \left\{ \sum_{t \notin \mathcal{I}_T} \left| \mu_t(\bar{\boldsymbol{\theta}}_t) - \mu_t(\boldsymbol{\theta}_\star) \right| + \sum_{t \notin \mathcal{I}_T} \left| \mu_t(\boldsymbol{\theta}_\star) - \mu_{t,\star}(\boldsymbol{\theta}_\star) \right| \right\}$$
$$\text{(Lemma E.3)}$$

$$\leq R_s \left\{ \sum_{t \notin \mathcal{I}_T} \left| \mu_t(\boldsymbol{\nu}_t) - \mu_t(\boldsymbol{\theta}_\star) \right| + \textcolor{red}{\sum_{t \notin \mathcal{I}_T} \left\{ \mu_{t,\star}(\boldsymbol{\theta}_\star) - \mu_t(\boldsymbol{\theta}_\star) \right\}} \right\}$$

$$\leq R_s \left\{ \sum_{t \notin \mathcal{I}_T} \left| \mu_t(\boldsymbol{\nu}_t) - \mu_t(\widehat{\boldsymbol{\theta}}_t) \right| + \sum_{t \notin \mathcal{I}_T} \left| \mu_t(\boldsymbol{\theta}_\star) - \mu_t(\widehat{\boldsymbol{\theta}}_t) \right| + \textcolor{red}{\mathsf{Reg}_{\mathcal{I}}(T)} \right\}$$

$$\leq 4 R_s \sum_{t \notin \mathcal{I}_T} \left| \mu_t(\boldsymbol{\nu}_t) - \mu_t(\widehat{\boldsymbol{\theta}}_{b(t)}) \right|. \qquad \text{(Definition of } (\boldsymbol{\nu}_t, b(t))\text{)}$$

**Bounding $\sum_t B_t$.**

$$\sum_{t \notin \mathcal{I}_T} B_t = \sum_{t \notin \mathcal{I}_T} \left| \dot{\mu}_t(\bar{\boldsymbol{\theta}}_t) - \dot{\mu}_t(\widehat{\boldsymbol{\theta}}_{b(t)}) \right| \left\| \boldsymbol{x}_t \right\|_{\widetilde{\boldsymbol{G}}_{b(t)}(\widehat{\boldsymbol{\theta}}_{b(t)}, \boldsymbol{\nu}_t)^{-1}} \left\| \boldsymbol{\nu}_t - \widehat{\boldsymbol{\theta}}_{b(t)} \right\|_{\widetilde{\boldsymbol{G}}_{b(t)}(\widehat{\boldsymbol{\theta}}_{b(t)}, \boldsymbol{\nu}_t)}$$

$$\leq \sqrt{4\lambda S^2 + \textcolor{blue}{\beta_T(\delta)^2}} \sum_{t \notin \mathcal{I}_T} \left| \dot{\mu}_t(\bar{\boldsymbol{\theta}}_t) - \dot{\mu}_t(\widehat{\boldsymbol{\theta}}_{b(t)}) \right| \left\| \boldsymbol{x}_t \right\|_{\widetilde{\boldsymbol{G}}_{b(t)}^{-1}(\widehat{\boldsymbol{\theta}}_{b(t)}, \boldsymbol{\nu}_t)}$$
$$(\boldsymbol{\nu}_t \in \mathcal{C}_{b(t)}, \text{ Lemma E.6 } (i))$$

$$\leq R_s \sqrt{4\lambda S^2 + \textcolor{blue}{\beta_T(\delta)^2}} \sum_{t \notin \mathcal{I}_T} \left| \mu_t(\bar{\boldsymbol{\theta}}_t) - \mu_t(\widehat{\boldsymbol{\theta}}_{b(t)}) \right| \left\| \boldsymbol{x}_t \right\|_{\widetilde{\boldsymbol{G}}_{b(t)}^{-1}(\widehat{\boldsymbol{\theta}}_{b(t)}, \boldsymbol{\nu}_t)}. \qquad \text{(Lemma E.3)}$$

We then inevitably introduce a $\kappa(T)$ dependency to use the elliptical potential arguments w.r.t. $\boldsymbol{V}_t$:

$$\sum_{t \notin \mathcal{I}_T} B_t \leq R_s \sqrt{4\lambda S^2 + \textcolor{blue}{\beta_T(\delta)^2}} \sqrt{2 g(\tau) \kappa(T)} \sum_{t \notin \mathcal{I}_T} \left| \mu_t(\bar{\boldsymbol{\theta}}_t) - \mu_t(\widehat{\boldsymbol{\theta}}_{b(t)}) \right| \left\| \boldsymbol{x}_t \right\|_{\boldsymbol{V}_t^{-1}}$$
$$\text{(Lemma E.4, } b(t) \geq t\text{)}$$

$$\leq R_s \sqrt{4\lambda S^2 + \textcolor{blue}{\beta_T(\delta)^2}} \sqrt{2 g(\tau) \kappa(T)} \sum_{t \notin \mathcal{I}_T} \left| \mu_t(\boldsymbol{\nu}_t) - \mu_t(\widehat{\boldsymbol{\theta}}_{b(t)}) \right| \left\| \boldsymbol{x}_t \right\|_{\boldsymbol{V}_t^{-1}}$$
$$\text{(Definition of } \boldsymbol{\nu}_t \text{ (Eqn. (19)))}$$

$$\leq 4 R_s R_{\dot{\mu}} \kappa(T) (4\lambda S^2 + \textcolor{blue}{\beta_T(\delta)^2}) \sqrt{g(\tau)} \sum_{t \notin \mathcal{I}_T} \left\| \boldsymbol{x}_t \right\|_{\boldsymbol{V}_t^{-1}}^2$$
$$(\boldsymbol{\nu}_t, \widehat{\boldsymbol{\theta}}_{b(t)} \in \mathcal{C}_{b(t)}, \text{ Lemma E.6 } (ii))$$

$$\leq 4 R_s R_{\dot{\mu}} \kappa(T) (4\lambda S^2 + \textcolor{blue}{\beta_T(\delta)^2}) \sqrt{g(\tau)} \sum_{t \in [T]} \min \left\{ 1, \left\| \boldsymbol{x}_t \right\|_{\boldsymbol{V}_t^{-1}}^2 \right\} \qquad \text{(Definition of } \mathcal{I}_T\text{)}$$

$$\leq 8 d R_s R_{\dot{\mu}} \kappa(T) (4\lambda S^2 + \textcolor{blue}{\beta_T(\delta)^2}) \sqrt{g(\tau)} \log \left( 1 + \frac{T}{2 d g(\tau) \kappa(T) \lambda} \right).$$
$$\text{(EPL (Lemma E.2))}$$

**Bounding $\sum_t C_t$.** We proceed similarly as bounding $\sum_t B_t$:

$$\sum_{t \notin \mathcal{I}_T} C_t \leq R_s \sqrt{4\lambda S^2 + \textcolor{blue}{\beta_T(\delta)^2}} \sum_{t \notin \mathcal{I}_T} \tilde{\alpha}_{b(t)}(\widehat{\boldsymbol{\theta}}_{b(t)}, \boldsymbol{\nu}_t) \left\| \boldsymbol{x}_t \right\|_{\widetilde{\boldsymbol{G}}_{b(t)}(\widehat{\boldsymbol{\theta}}_{b(t)}, \boldsymbol{\nu}_t)^{-1}}^2$$
$$(\boldsymbol{\nu}_t \in \mathcal{C}_{b(t)}, \text{ Lemma E.6 } (i))$$

$$\leq R_s R_{\dot{\mu}} g(\tau) \kappa(T) \sqrt{4\lambda S^2 + \textcolor{blue}{\beta_T(\delta)^2}} \sum_{t \notin \mathcal{I}_T} \left\| \boldsymbol{x}_t \right\|_{\boldsymbol{V}_t^{-1}}^2 \qquad \text{(Lemma E.4, } b(t) \geq t\text{)}$$

$$\leq R_s R_{\dot{\mu}} g(\tau) \kappa(T) \sqrt{4\lambda S^2 + \textcolor{blue}{\beta_T(\delta)^2}} \sum_{t \in [T]} \min \left\{ 1, \left\| \boldsymbol{x}_t \right\|_{\boldsymbol{V}_t^{-1}}^2 \right\} \qquad \text{(Definition of } \mathcal{I}_T\text{)}$$

$$\leq 2 d R_s R_{\dot{\mu}} g(\tau) \kappa(T) \sqrt{4\lambda S^2 + \textcolor{blue}{\beta_T(\delta)^2}} \log \left( 1 + \frac{T}{2 d g(\tau) \kappa(T) \lambda} \right) \qquad \text{(EPL (Lemma E.2))}$$

**Wrapping Up.** Let us choose $\lambda = \frac{1}{4S^2}$, and let us denote $A \lesssim B$ if $A \le cB$ for some absolute constant $c > 0$. Then, combining everything, we have:

$$\sum_{t \notin \mathcal{I}_T} \left| \mu_t(\boldsymbol{\nu}_t) - \mu_t(\widehat{\boldsymbol{\theta}}_{b(t)}) \right|$$

$$\le \sum_{t \notin \mathcal{I}_T} A_t + \sum_{t \notin \mathcal{I}_T} B_t + \sum_{t \notin \mathcal{I}_T} C_t$$

$$\lesssim \beta_T(\delta) \sqrt{dg(\tau) \log \left(1 + \frac{R_{\dot\mu} S T}{d}\right)} \sqrt{\frac{T}{\kappa_\star(T)} + R_s \sum_{t \notin \mathcal{I}_T} \left| \mu_t(\boldsymbol{\nu}_t) - \mu_t(\widehat{\boldsymbol{\theta}}_{b(t)}) \right|}$$

$$+ dR_s R_{\dot\mu} \kappa(T) \beta_T(\delta) \sqrt{g(\tau)} \log \left(1 + \frac{ST}{dg(\tau)\kappa(T)}\right),$$

as the upper bound for $\sum_t C_t$ is asymptotically negligible compared to that of $\sum_t B_t$.

This is of the form $X \lesssim A\sqrt{B + R_s X} + C$ with $X := \sum_{t \notin \mathcal{I}_T} \left| \mu_t(\boldsymbol{\nu}_t) - \mu_t(\widehat{\boldsymbol{\theta}}_{b(t)}) \right|$, which then implies $X \lesssim A\sqrt{B} + A\sqrt{R_s} + C$ thanks to an elementary polynomial inequality (Abeille et al., 2021, Proposition 7). We then conclude by combining the above inequality with the regret decomposition done at the beginning. $\qquad\square$

### E.3 Supporting Lemmas

The following are the elliptical potential arguments, which we state without proof:

**Lemma E.1** (Elliptical Potential Count Lemma; EPCL[12]). *For $X, L > 0$, let $\boldsymbol{x}_1, \cdots, \boldsymbol{x}_T \in \mathcal{B}^d(X)$ be a sequence of vectors, $\boldsymbol{V}_t := \lambda \boldsymbol{I} + \sum_{s=1}^{t-1} \boldsymbol{x}_s \boldsymbol{x}_s^\intercal$, and let us define the following: $\mathcal{H}_T := \left\{ t \in [T] : \|\boldsymbol{x}_t\|_{\boldsymbol{V}_t^{-1}}^2 > L \right\}$. Then, we have that*

$$|\mathcal{H}_T| \le \frac{2d}{\log(1 + L^2)} \log \left(1 + \frac{X^2}{\lambda \log(1 + L^2)}\right). \tag{23}$$

**Lemma E.2** (Elliptical Potential Lemma; EPL[13]). *Let $\boldsymbol{x}_1, \cdots, \boldsymbol{x}_T \in \mathcal{B}^d(X)$ be a sequence of vectors and $\boldsymbol{V}_t := \lambda \boldsymbol{I} + \sum_{s=1}^{t-1} \boldsymbol{x}_s \boldsymbol{x}_s^\intercal$. Then, we have that*

$$\sum_{t=1}^{T} \min \left\{ 1, \|\boldsymbol{x}_t\|_{\boldsymbol{V}_t^{-1}}^2 \right\} \le 2d \log \left(1 + \frac{X^2 T}{d\lambda}\right). \tag{24}$$

The following is a "self-bounding" property of self-concordant function:

**Lemma E.3.** *For $\boldsymbol{\theta}, \boldsymbol{\nu} \in \mathbb{R}^d$, $|\dot\mu_t(\boldsymbol{\theta}) - \dot\mu_t(\boldsymbol{\nu})| \le R_s |\mu_t(\boldsymbol{\theta}) - \mu_t(\boldsymbol{\nu})|$*

*Proof.* This follows from direct computation:

$$|\dot\mu_t(\boldsymbol{\theta}) - \dot\mu_t(\boldsymbol{\nu})| = \left| \langle \boldsymbol{x}_t, \boldsymbol{\theta} - \boldsymbol{\nu} \rangle \int_0^1 \ddot\mu_t(\boldsymbol{\nu} + v(\boldsymbol{\theta} - \boldsymbol{\nu}) dv \right|$$

$$\le |\langle \boldsymbol{x}_t, \boldsymbol{\theta} - \boldsymbol{\nu} \rangle| \int_0^1 \left| \ddot\mu_t(\boldsymbol{\nu} + v(\boldsymbol{\theta} - \boldsymbol{\nu})) \right| dv$$

$$\le R_s \left| \langle \boldsymbol{x}_t, \boldsymbol{\theta} - \boldsymbol{\nu} \rangle \right| \int_0^1 \left| \dot\mu_t(\boldsymbol{\nu} + v(\boldsymbol{\theta} - \boldsymbol{\nu})) \right| dv \qquad \text{(Assumption 4)}$$

---

[12]This is a generalization of Exercise 19.3 of Lattimore and Szepesvári (2020), presented (in parallel) at Lemma 7 of Gales et al. (2022) and Lemma 4 of Kim et al. (2022).

[13]Lemma 11 of Abbasi-Yadkori et al. (2011).

$$= R_s \left| \langle \boldsymbol{x}_t, \boldsymbol{\theta} - \boldsymbol{\nu} \rangle \int_0^1 \dot{\mu}_t(\boldsymbol{\nu} + v(\boldsymbol{\theta} - \boldsymbol{\nu})) dv \right|$$

$$(m \text{ is convex, and thus } \dot{\mu} = m'' \geq 0)$$

$$= R_s \left| \mu_t(\boldsymbol{\theta}) - \mu_t(\boldsymbol{\nu}) \right|.$$

$\square$

This self-concordant result is distinct from the original self-concordance control lemma (Faury et al., 2020, Lemma 9) and does not incur any dependency on $S$. We also remark that the above self-bounding lemma has been independently proven and used for regret analyses of **GLB**s in two concurrent works (Janz et al., 2024, Claim 14) (Liu et al., 2024, Lemma 31).

The following properties are crucial in allowing for the application of EP(C)L:

**Lemma E.4.** *For any $\boldsymbol{\theta}, \boldsymbol{\nu} \in \mathbb{R}^d$, $\frac{1}{2\kappa(T)} \leq \tilde{\alpha}_t(\boldsymbol{\theta}, \boldsymbol{\nu}) \leq \frac{R_{\dot{\mu}}}{2}$, and thus, $\frac{1}{2g(\tau)\kappa(T)} \boldsymbol{V}_t \preceq \widetilde{\boldsymbol{G}}_t(\boldsymbol{\theta}, \boldsymbol{\nu})$.*

*Proof.* Follows from straightforward computation. $\square$

In the following two lemmas, $b(t)$ is as defined in Eqn. (19).

**Lemma E.5.** $\widetilde{\boldsymbol{G}}_{b(t)}(\widehat{\boldsymbol{\theta}}_{b(t)}, \boldsymbol{\nu}_t) \succeq \frac{1}{2g(\tau)} \bar{\boldsymbol{H}}_t$.

*Proof.* For each $s \leq b(t)$,

$$\tilde{\alpha}_s(\widehat{\boldsymbol{\theta}}_{b(t)}, \boldsymbol{\nu}_t) = \int_0^1 (1-v) \dot{\mu}_s \left( \widehat{\boldsymbol{\theta}}_{b(t)} + v(\boldsymbol{\nu}_t - \widehat{\boldsymbol{\theta}}_{b(t)}) \right) dv \overset{(*)}{\geq} \dot{\mu}_s(\bar{\boldsymbol{\theta}}_s) \int_0^1 (1-v) dv = \frac{1}{2} \dot{\mu}_s(\bar{\boldsymbol{\theta}}_s),$$

where $(*)$ follows from the observations that $\boldsymbol{\nu}_t, \widehat{\boldsymbol{\theta}}_{b(t)} \in \mathcal{C}_{b(t)}$ and $\mathcal{C}_{b(t)}$ is convex. We then conclude by noting that $b(t) \geq t$, and thus $\bar{\boldsymbol{H}}_{b(t)} \succeq \bar{\boldsymbol{H}}_t$. $\square$

**Lemma E.6.** *For any $t \geq 1$ and $\boldsymbol{\theta}, \boldsymbol{\nu} \in \mathcal{C}_{b(t)}$, we have the following:*

*(i)* $\left\| \boldsymbol{\nu} - \widehat{\boldsymbol{\theta}}_{b(t)} \right\|_{\widetilde{\boldsymbol{G}}_{b(t)}(\widehat{\boldsymbol{\theta}}_{b(t)}, \boldsymbol{\nu})} \leq \sqrt{4\lambda S^2 + \beta_T(\delta)^2}$,

*(ii)* $\left| \mu_t(\boldsymbol{\nu}) - \mu_t(\boldsymbol{\theta}) \right| \leq 2R_{\dot{\mu}} \sqrt{2 \left( 4\lambda S^2 + \beta_T(\delta)^2 \right) \kappa(T)} \left\| \boldsymbol{x}_t \right\|_{\boldsymbol{V}_t^{-1}}$.

*Proof.* *(i)* follows from Taylor's theorem with integral remainder, first-order condition for convex constrained optimization (see footnote 10 in Appendix D), and the fact that $\mathcal{C}_{b(t)} \subseteq \mathcal{B}^d(S)$:

$$\beta_T(\delta)^2 \geq \mathcal{L}_{b(t)}(\boldsymbol{\nu}) - \mathcal{L}_{b(t)}(\widehat{\boldsymbol{\theta}}_t) = \underbrace{\langle \nabla \mathcal{L}_{b(t)}(\widehat{\boldsymbol{\theta}}_{b(t)}), \boldsymbol{\nu} - \widehat{\boldsymbol{\theta}}_{b(t)} \rangle}_{\geq 0} + \left\| \boldsymbol{\nu} - \widehat{\boldsymbol{\theta}}_{b(t)} \right\|_{\widetilde{\boldsymbol{G}}_{b(t)}(\widehat{\boldsymbol{\theta}}_{b(t)}, \boldsymbol{\nu}) - \lambda \boldsymbol{I}}^2$$

$$\geq \left\| \boldsymbol{\nu} - \widehat{\boldsymbol{\theta}}_{b(t)} \right\|_{\widetilde{\boldsymbol{G}}_{b(t)}(\widehat{\boldsymbol{\theta}}_{b(t)}, \boldsymbol{\nu})}^2 - 4\lambda S^2.$$

*(ii)* follows from *(i)* and similar arguments:

$$\left| \mu_t(\boldsymbol{\nu}) - \mu_t(\boldsymbol{\theta}) \right| = \left| \langle \boldsymbol{x}_t, \boldsymbol{\nu} - \boldsymbol{\theta} \rangle \int_0^1 \dot{\mu}_t(\boldsymbol{\theta} + v(\boldsymbol{\nu} - \boldsymbol{\theta})) dv \right|$$

$$\leq R_{\dot{\mu}} \left\{ \left\| \boldsymbol{\nu} - \widehat{\boldsymbol{\theta}}_{b(t)} \right\|_{\widetilde{\boldsymbol{G}}_{b(t)}(\widehat{\boldsymbol{\theta}}_{b(t)}, \boldsymbol{\theta})} + \left\| \boldsymbol{\theta} - \widehat{\boldsymbol{\theta}}_{b(t)} \right\|_{\widetilde{\boldsymbol{G}}_{b(t)}(\widehat{\boldsymbol{\theta}}_{b(t)}, \boldsymbol{\theta})} \right\} \left\| \boldsymbol{x}_t \right\|_{\widetilde{\boldsymbol{G}}_{b(t)}(\widehat{\boldsymbol{\theta}}_{b(t)}, \boldsymbol{\theta})^{-1}}$$

$$(\text{Cauchy-Schwartz \& triangle inequalities})$$

$$\leq 2R_{\dot{\mu}} \sqrt{2 \left( 4\lambda S^2 + \beta_T(\delta)^2 \right) \kappa(T)} \left\| \boldsymbol{x}_t \right\|_{\boldsymbol{V}_{b(t)}^{-1}} \qquad ((i), \text{Lemma E.4})$$

$$\leq 2R_{\dot{\mu}} \sqrt{2 \left( 4\lambda S^2 + \beta_T(\delta)^2 \right) \kappa(T)} \left\| \boldsymbol{x}_t \right\|_{\boldsymbol{V}_t^{-1}}. \qquad (b(t) \geq t)$$

$\square$

# F  Alternate CS via Discrete Uniform Prior and Covering Argument

In this Appendix, instead of the PAC-Bayes with a continuous uniform prior/posterior as in the main text, we explore an alternate derivation of CS using a discrete uniform prior. This is a supplementary discussion for the "**Fast Rates in Statistical Learning**" paragraph in Appendix A.

We present the alternate CS, which is strictly looser than our Theorem 3.1 but more "elementary":

> **Theorem F.1** (Slightly Looser, Unified CS for GLMs). *Let $L_t := \max_{\boldsymbol{\theta} \in \Theta} \left\| \nabla \mathcal{L}_t(\boldsymbol{\theta}) \right\|_2$ be the Lipschitz constant of $\mathcal{L}_t(\cdot)$ that may depend on $\{(\boldsymbol{x}_s, r_s)\}_{s=1}^{t-1}$. Then, we have $\mathbb{P}[\exists t \geq 1 : \boldsymbol{\theta}_\star \notin \mathcal{C}_t(\delta)] \leq \delta$, where*
>
> $$\beta_t(\delta)^2 = \log \frac{\pi^2 t^2}{6\delta} + \inf_{c \in (0, 5S]} \left\{ d \log \frac{5S}{c} + c L_t \right\} \leq \log \frac{\pi^2 t^2}{6\delta} + d \log(1 \vee 5SL_t) + 1,$$
>
> *where the last inequality follows from the choice $c = 5S \wedge \frac{1}{L_t}$.*

*Proof.* Consider $p = \mathcal{U}(\{\boldsymbol{\theta}_i\}_{i \in [N]})$, where the $\boldsymbol{\theta}_i$'s will be determined later. In that case, we have:

$$\log \mathbb{E}_{\boldsymbol{\theta} \sim p}[M_t(\boldsymbol{\theta})] = \mathcal{L}_t(\boldsymbol{\theta}_\star) + \log \mathbb{E}_{\boldsymbol{\theta} \sim p}[\exp\left(-\mathcal{L}_t(\boldsymbol{\theta})\right)]$$

$$= \mathcal{L}_t(\boldsymbol{\theta}) + \log \left\{ \frac{1}{N} \sum_{i=1}^N \exp\left(-\mathcal{L}_t(\boldsymbol{\theta}_i)\right) \right\}$$

$$\geq \mathcal{L}_t(\boldsymbol{\theta}_\star) + \log \left\{ \frac{1}{N} \max_{i \in [N]} \exp\left(-\mathcal{L}_t(\boldsymbol{\theta}_i)\right) \right\}$$

$$= \mathcal{L}_t(\boldsymbol{\theta}_\star) - \min_{i \in [N]} \mathcal{L}_t(\boldsymbol{\theta}_i) + \log \frac{1}{N}. \tag{25}$$

From the proof of Lemma 3.1, one can see that $\mathbb{E}[M_t(\boldsymbol{\theta})|\boldsymbol{\theta}] = 1$ where $\mathbb{E}$ is w.r.t. the randomness of the sequential data (i.e., of $\mathcal{L}_t(\cdot)$). Then, by Markov's inequality, for any $\delta \in (0, 1)$,

$$\mathbb{P}\left( M_t(\boldsymbol{\theta}_i) \geq \frac{N}{\delta} \right) \leq \frac{\delta}{N}, \quad \forall i \in [N], \tag{26}$$

where again, $\mathbb{P}$ is w.r.t. the randomness of $\mathcal{L}_t(\cdot)$. Then, we have that

$$\mathbb{P}\left( \mathbb{E}_{\boldsymbol{\theta} \sim p}[M_t(\boldsymbol{\theta})] = \frac{1}{N} \sum_{i \in [N]} M_t(\boldsymbol{\theta}_i) \geq \frac{1}{\delta} \right) \leq \mathbb{P}\left( \max_{i \in [N]} M_t(\boldsymbol{\theta}_i) \geq \frac{N}{\delta} \right)$$

$$\leq \sum_{i \in [N]} \mathbb{P}\left( M_t(\boldsymbol{\theta}_i) \geq \frac{N}{\delta} \right) \qquad \text{(union bound)}$$

$$\leq \sum_{i \in [N]} \frac{\delta}{N} = \delta. \qquad \text{(Eqn. (26))}$$

Combining this with Eqn. (25), we have that

$$\mathbb{P}\left( \mathcal{L}_t(\boldsymbol{\theta}_\star) - \min_{i \in [N]} \mathcal{L}_t(\boldsymbol{\theta}_i) \leq \log \frac{N}{\delta} \right) \geq 1 - \delta, \quad \forall t \geq 1.$$

By reparametrizing $\delta$ as $\frac{\delta}{t^2}$ and taking the union bound over $t \geq 1$, we have that by the Basel sum,

$$\mathbb{P}\left( \exists t \geq 1 : \mathcal{L}_t(\boldsymbol{\theta}_\star) - \min_{i \in [N]} \mathcal{L}_t(\boldsymbol{\theta}_i) \geq \log N + \log \frac{\pi^2 t^2}{6\delta} \right) \leq \delta. \tag{27}$$

Thus, the following holds with probability at least $1 - \delta$: for all $t \geq 1$,

$$\mathcal{L}_t(\boldsymbol{\theta}_\star) - \min_{\boldsymbol{\theta} \in \Theta} \mathcal{L}_t(\boldsymbol{\theta}) \leq \log \frac{\pi^2 t^2}{6\delta} + \log N + \min_{i \in [N]} \mathcal{L}_t(\boldsymbol{\theta}_i) - \min_{\boldsymbol{\theta} \in \Theta} \mathcal{L}_t(\boldsymbol{\theta})$$

$$\leq \log \frac{\pi^2 t^2}{6\delta} + \log N + L_t \min_{i \in [N]} \left\| \boldsymbol{\theta}_i - \widehat{\boldsymbol{\theta}}_t \right\|_2, \quad (\widehat{\boldsymbol{\theta}}_t = \arg\min_{\boldsymbol{\theta} \in \Theta} \mathcal{L}_t(\boldsymbol{\theta}))$$

where we recall that $L_t$ is the Lipschitz constant of $\mathcal{L}_t(\cdot)$.

We choose $\{\boldsymbol{\theta}_i\}$ to be a $c$-net of $\Theta$ for $c \in (0, 5S]$. Then, $\min_{i \in [N]} \left\| \boldsymbol{\theta}_i - \widehat{\boldsymbol{\theta}}_t \right\|_2 \leq c$ by definition, and as $\Theta \subseteq \mathcal{B}^d(S)$, $N \leq \left( \frac{5S}{c} \right)^d$ (Vershynin, 2018, Corollary 4.2.13). Combining everything and taking $\min_{c \in (0, 5S]}$ gives the desired statement. $\qquad\square$

# G  Deferred Experimental Details and Results from Section 5

## G.1  Implementation Details

For time-varying arm-sets, the randomness of the arm-sets is shared across all the algorithms, i.e., at each time-step $t$, all the algorithms see the same arm-set. Thus, the only randomness is from the reward distributions. Whenever applicable, we utilize the Sequential Least SQuares Programming (SLSQP; Kraft (1988)) implemented in SciPy (Virtanen et al., 2020) for computing the norm-constrained MLE. This minimizes the effect of optimization errors whenever possible, allowing us to compare the algorithms clearly from a statistical perspective. For OFUGLB, EMK, RS-GLinCB, and OFULog+, SLSQP is utilized to compute the UCB index as well. We use the same implementation of ada-OFU-ECOLog and RS-GLinCB as in the publicly available GitHub repository of Faury et al. (2022)[14] and Sawarni et al. (2024)[15], respectively. As mentioned in the main text, we utilize the exact theoretical hyperparameters for RS-GLinCB as provided in Algorithm 2 of Sawarni et al. (2024).

## G.2  Additional Results for Logistic Bandits with Fixed Arm-Set

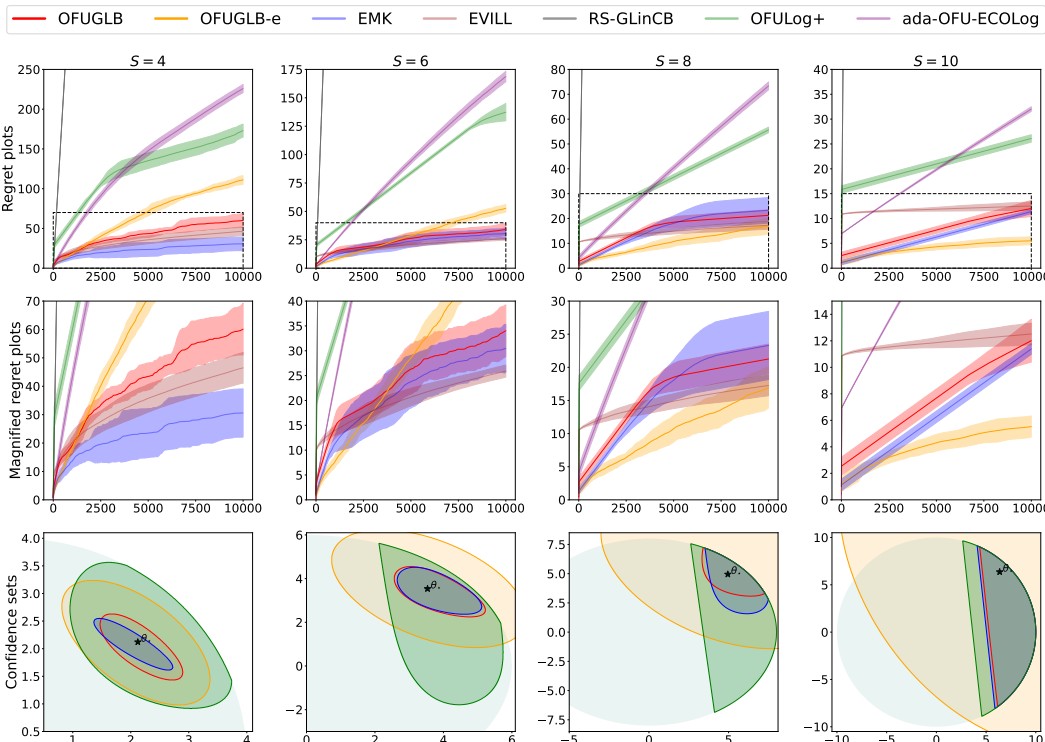

Figure 2: Fixed arm-set. (First row) Regret plots of all considered algorithms. (Second row) Magnified regret plots. (Third row) Confidence set plots at the final time $t = 10000$ when applicable. Each column represents a different logistic bandit instance for $S \in \{4, 6, 8, 10\}$.

**Results and Discussions.**     The results are shown in Figure 2. There are some common characteristics compared to the plots for time-varying arm-sets (Figure 1 in the main text). There is still a discrepancy between the tightness of the CSs and the actual regret for OFULog+ vs. OFUGLB-e, and RS-GLinCB still performs the worst. Also, OFUGLB, EMK, and EVILL are still the best-performing algorithms, at least eventually. Let us now highlight some key qualitative differences from time-varying arm-set plots as well as relevant discussions.

First, the regret curves seem linearly increasing overall, especially at $S \in \{8, 10\}$. In our settings, $T = 10000$ is still in a transient phase of all the algorithms and, thus, yet to reach the asymptotic

[14]https://github.com/louisfaury/logistic_bandit
[15]https://github.com/nirjhar-das/GLBandit_Limited_Adaptivity

regime. To see the logarithmic-looking regret curve and to numerically compare the "numerical asymptotic regret" of the algorithms, we plan to run the experiments for much longer timesteps, e.g., $T = 50000$.

Second, for $S = 10$, it *seems* that `OFUGLB-e` is the best performing algorithm. We suspect that this is because the `OFUGLB-e` happens to exploit a "good" direction in the beginning, and In other words, we believe that if the experiments are run with much more iterations, then at the end, due to its design, `OFUGLB-e` will have to explore other unexplored directions, causing an increase in the regret. Indeed, if one takes a close look at $S \in \{6, 8\}$, note that there is a phase at which `OFUGLB-e` seems to perform the best in the beginning, but in the end its regret increases well beyond other well-performing baselines: `OFUGLB`, `EMK`, and `EVILL`.

**Remark 5.** *Although our current implementation always uses SLSQP for all the optimization proce-dures (for MLE and UCB index computations), when the arm-set is fixed, the overall implementations of all the algorithms can be made more computationally efficient. One approach is to utilize the iterative reweighted least squares (IRLS; Wolke and Schwetlick (1988)) and keep track of the number of pulls of each arm vector, which is possible as the arm-set is fixed); see Section 3.3 of Kveton et al. (2020) and the original implementation[16] of* `EVILL` *using IRLS.*

### G.3 Additional Results for Logistic Bandits with $|\mathcal{A}_t| = 10$, with Some Updates

Here, we provide additional results for $|\mathcal{A}_t| = 10$, for both time-varying and fixed arm-sets. These results were obtained as a test run of the significantly refactored codes in our GitHub repository (see the commits in Jan 2025). Compared to the experimental details presented so far, there are two main updates.

**Notable Updates.**  One is that we utilize the exact theoretical hyperparameters for `EVILL` as provided in Appendix E of Janz et al. (2024). To be as faithful to the theoretical results presented in Janz et al. (2024), for the fixed arm-set setting, we implement the warm-up phase of `EVILL` via the G-optimal design (Pukelsheim, 2006), which is in turn implemented using Frank-Wolfe iterations (Frank and Wolfe, 1956); see Appendix B of Janz et al. (2024) and references therein for more discussions. Another is that for the implementation of `EMK` (Emmenegger et al., 2023), we utilize the Vovk-Azoury-Warmuth forecaster type regularizer due to `AIOLI` of Jézéquel et al. (2020) instead of the log-partition-based regularizer as suggested by Emmenegger et al. (2023).

**Results and Discussions.**  The results are shown in Figure 3 and 4. There are several notable observations to be made. One is that with the theoretical hyperparameter, `EVILL` performs worst or second-worst, suggesting that despite its practical efficacy (Chapelle and Li, 2011; Russo et al., 2018), there is still a big theory-practice gap, at least for logistic bandits. Second observation is that our `OFUGLB` performs worse than `EMK`. We believe this is due to the change in the arm set size, $|\mathcal{A}_t|$. Still, the trend suggests that as $S$ gets larger, our `OFUGLB` may perform better than `EMK`. This is expected, considering how our theories focus on removing the $S$-dependency, which is significant only when $S$ is large. We should, however, emphasize that Emmenegger et al. (2023) does not provide a *tight theoretical (regret) guarantee* for `EMK`, while ours do. Also, in the time-varying arm-set setting, our `OFUGLB`'s CS is tighter than `EMK`'s. Lastly, in the fixed arm-set setting, `OFUGLB-e` behaves quite unstably as $S$ increases. We believe this is due to our theoretical choice of $\lambda = \frac{1}{8S^2(1+SR_s)}$ decaying with $S$, possibly making $\nabla^2 \mathcal{L}_t(\widehat{\boldsymbol{\theta}}_t) + \lambda \boldsymbol{I}_d$ ill-conditioned. In practice, one could tune $\lambda$ for a good and stable performance.

---

[16]https://github.com/DavidJanz/EVILL-code

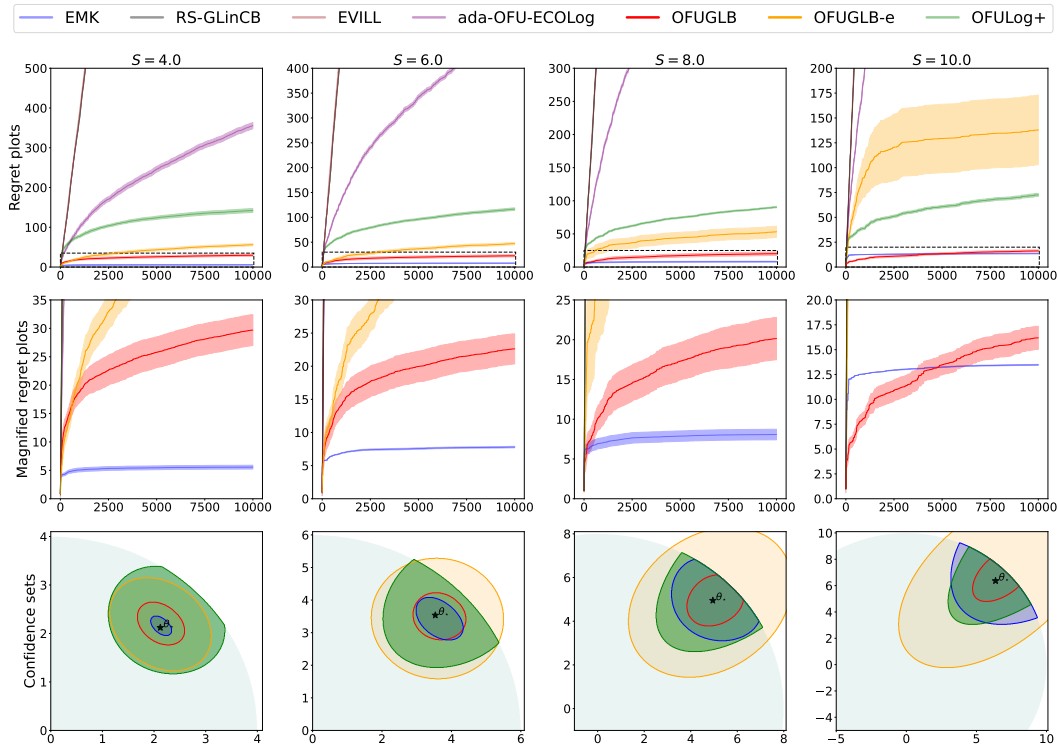

Figure 3: Time-varying arm-sets with $|\mathcal{A}_t| = 10$.

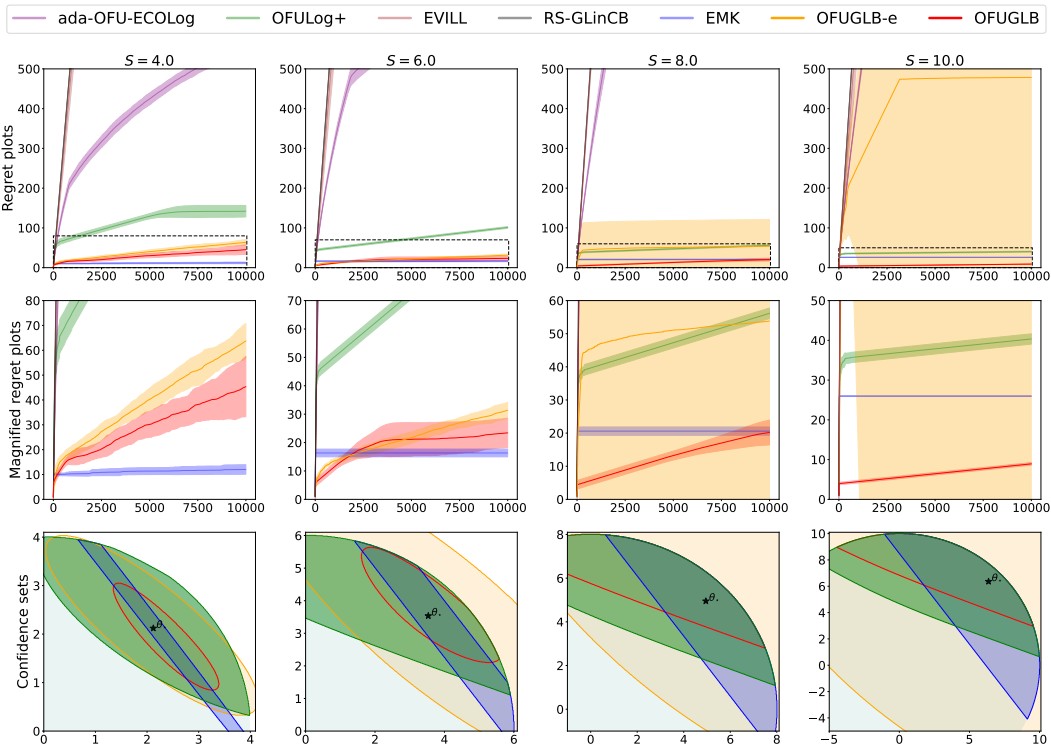

Figure 4: Fixed arm-set with $|\mathcal{A}| = 10$.

