# OpenReview forum: "A Unified Confidence Sequence for Generalized Linear Models, with Applications to Bandits"
_NeurIPS.cc/2024/Conference — NeurIPS 2024 poster_

### Official Review · Reviewer_aQNL · 2024-06-19

**Soundness:** 4
**Presentation:** 3
**Contribution:** 3
**Rating:** 6
**Confidence:** 4

**Summary:**

The paper derives a new   time-independent inequality for likelihood ratios in the Generalized Linear Model. The proof is based on a PAC-Bayesian approach with a well-chosen prior. The result is applied to GLM bandit models, improving a variant of GLM-UCB. The resulting regret bound removes a exponential dependency in the norm of the unknown parameter.
Several examples of GLM are discussed, and minimal numerical experiments are reported. The comparison with OFULLog+ is convincingly exposed, both in theory and in practice.

**Strengths:**

Overall, I consider that the submission is technically sound, somewhat incremental, but will be of interest to the community.

**Weaknesses:**

The submission is correctly written, but with a number of clumsinesses (some are listed below).
The main text contains not much more than the statement of the results and the (unsurprising) description of GLM-UCB+, while the supplementary material is written a collection of appendices that are not very well presented. For example, the first appendix is called "Missing Results", which is pretty unspecific. I did not check all the details, but the main lines look correct.

l.69 undefined notation \mathcal{B}^d(1)
l.77 give a reference
l.80 could you precise what is R_\mu in those examples?
l.95 the sentence is grammatically wrong, something is missing
Thm. 3.1: replace "where" by "for the choice"
l.129 it is not the log-likelihood but the likelihood
l.583 is it not possible to suggest a proof instead of a reference to "WoframAlpha" (to a plot?)

**Questions:**

How smaller can \beta_t(\delta) be for a confidence region valid only for a given t>0 (instead of all t>0)? This might be worth writing in the paper

**Limitations:**

This section does not really seem relevant to this mostly theoretical submission

---

> ### Author Rebuttal · Authors · 2024-08-05
>
> We thank the reviewer for valuable feedback and suggestions, and for recognizing our paper's technical soundness and contributions.
>
> **Number of clumsinesses, Typos, Organization issues**
>
> We apologize for the typographical errors and the organizational issues identified in the manuscript. We are committed to correcting these in the revised version, including re-organizing the appendices as suggested by the reviewer. We sincerely appreciate the reviewer's effort in pointing out these issues.
>
> To ensure clarity, we address each typo as follows:
> - Line 69. $\mathcal{B}^d(1)$ is the ball of radius 1 centered at the origin in R^d.
> - Line 77. The mentioned properties of GLM are provided in [14], which we will add as reference
> - Line 80. Of course. For Gaussian, $R_{\dot{\mu}} = 1$; for Poisson, $R_{\dot{\mu}} = e^S$; for Bernoulli, $R_{\dot{\mu}} = \frac{1}{4}$. Thank you for this suggestion, which would make our presentation clearer.
> - Line 95. Yes. We will appropriately change the sentence.
> - Line 129. Yes, it should be the likelihood ratio.
> - Line. 583. We sincerely apologize for this mishap. Due to the pressing time, we forgot to replace this with rigorous proof. Below, we provide rigorous proof of this result, which will, of course, be reflected in the revision:
>
> Let us denote $f(p) := p^{-1/2} \left( \mathbb{E}[{X}^p] \right)^{1/p}$ for $p \in \mathbb{N}$.
>
> Then, using well-known properties of the Gamma function, we have that
> \begin{equation*}
>     f(2p) = \sigma 2^{\frac{2p-1}{4p}} (2p)^{\frac{1}{2p} - \frac{1}{2}} \left( (p-1)! \right)^{\frac{1}{2p}}
>     = \sigma \sqrt{ p^{-1} \left( \sqrt{2} p! \right)^{\frac{1}{p}} }
> \end{equation*}
> and
> \begin{equation*}
>     f(2p - 1) = \sigma 2^{\frac{2p - 2}{2(2p - 1)}} (2p - 1)^{\frac{1}{2p - 1} - \frac{1}{2}} \left( \sqrt{\pi} \frac{(2p - 3)!!}{2^{p - 1}} \right)^{\frac{1}{2p - 1}}
>     = \sigma (2p - 1)^{\frac{1}{2p - 1} - \frac{1}{2}} \left( \sqrt{\pi} (2p - 3)!! \right)^{\frac{1}{2p-1}},
> \end{equation*}
> where we define $(-1)!! := 1$.
>
> Then, we have that
> \begin{equation*}
>     f(2p) \overset{(i)}{<} \sigma \sqrt{p^{-1} (\sqrt{2} p^p)^{\frac{1}{p}}}
>     = \sigma 2^{\frac{1}{4p}}
>     \leq \sigma 2^{\frac{1}{4}},
> \end{equation*}
> where $(i)$ follows from $p! < p^p$.
> We also have that
> \begin{equation*}
>     f(2p-1) \overset{(i)}{<} \sigma (2p - 1)^{\frac{1}{2p - 1} - \frac{1}{2}} \left( \sqrt{\pi} (2p - 1)^p \right)^{\frac{1}{2p-1}}
>     \overset{(ii)}{<} \sigma \left( \sqrt{\pi} (2p - 1)  \right)^{\frac{1}{2p-1}}
>     \overset{(iii)}{<} \sigma \sqrt{\pi},
> \end{equation*}
> where $(i)$ follows from $(2p - 3)!! < (2p - 1)^p$, $(ii)$ follows from $\frac{p}{2p - 1} > \frac{1}{2}$, and $(iii)$ follows from the observations that for $z \geq e$, $f(z) = (\sqrt{\pi} z)^{1/z}$ is decreasing, and $f(1) = \sqrt{\pi} > f(3) = (3 \sqrt{\pi})^{1/3}$. The first observation can be easily verified as follows: $\frac{d}{dz} \log f(z) = \sqrt{\pi} \frac{1 - \log z}{z^2} \leq 0, \quad \forall z \geq e.$
>
> Finally, as $2^{1/4} < \sqrt{\pi}$, we have that $\sup_{p \in \mathbb{N}} f(p) \leq \sqrt{\pi} \sigma.$
>
> ----
>
>
> **How small can $\beta_t(\delta)$ be when we don’t require the confidence region to be time-uniform?**
>
> This is an interesting question, and thank you for bringing this up. First, we note that the mixture of martingale-type arguments (e.g., our PAC-Bayesian time-uniform guarantees) usually gives an anytime-valid guarantee *for free*.
>
> If the reviewer is asking about the optimal $\beta_t(\delta)$ when the data $\mathcal{D}_t =  \\{ (x_s, r_s) \\}\_{s \in [t]}$ can be adaptively collected for a given $t$, we don't have a definitive answer. This resembles the fixed-budget best-arm identification in bandits [15], which may be a good starting point for this direction.
>
> When the data $\mathcal{D}_t$ is collected independently, the problem reduces to the fixed-design setup. For the linear case, one has the confidence region of the following form [Eqn. (20.2) of 16]: for any fixed $x \in \mathcal{B}^d(1)$,
>
> \begin{equation}
> \mathbb{P}\left( \langle x, \hat{\theta}_t - \theta\_{\star} \rangle \leq \mathcal{O}\left( \lVert x \rVert\_{V_t^{-1}} \sqrt{\log\frac{1}{\delta}} \right) \right) \geq 1 - \delta,
> \end{equation}
>
> where $V_t = \sum_{s=1}^t x_s x_s^\top$ is the design matrix that we assume to be invertible for now.
>
> For the logistic model, we have the following [Theorem 1, 17]:
>
> \begin{equation*}
> \mathbb{P}\left( \langle x, \hat{\theta}_t - \theta\_{\star} \rangle \leq \mathcal{O}\left( \lVert x \rVert\_{H_t^{-1}} \sqrt{\log\frac{t\_{eff}}{\delta}} \right) \right) \geq 1 - \delta,
> \end{equation*}
>
> where $t\_{eff}$ is the number of distinct vectors in $\\{x_s\\}\_{s \in [t]}$ and $H_t = \sum_{s=1}^t \dot{\mu}(\langle \widehat{\theta}_t, x \rangle) x_s x_s^\top$ is the Fisher information matrix that we assume to be invertible for now.
> In both cases, the confidence width is dimension-independent and asymptotically optimal with respect to the Cramér-Rao lower bound. Thus, for each fixed $t$, we indeed have a tighter $\beta_t(\delta)$.

---

### Official Review · Reviewer_HLad · 2024-06-24

**Soundness:** 3
**Presentation:** 3
**Contribution:** 3
**Rating:** 3
**Confidence:** 4

**Summary:**

Confidence intervals for GLMs using a PAC-Bayes approach.

**Strengths:**

Confidence sequences and bandit algorithms for GLMs are a hot topic, and the authors contribution to that topic is solid, both because of the strength of the result itself, and because the proof itself is very simple and easy to verify---something that cannot be said about some of the previous work on the topic.

**Weaknesses:**

The paper is written in a style that might be described as "being written for the reviewers". It contains misleading and inaccurate claims which are meant to, presumably, impress the reader with the quality of the results. I give some examples shortly. Its unfortunate, because the ideas in the paper are good, and if only the authors had simply written about their work in a neutral, descriptive manner, I would absolutely recommend that the paper is accepted. However, I do not believe the overclaiming to be some sort of an accident---which if it were, the authors might fix upon my pointing it out---and so without any system for "revise and resubmit", must recommend rejection.

Some examples of problematic claims (emphasis mine in all quotes):
1) Lines 1-2, the authors claim that "[...] for any generalised linear models (GLMs) that is guaranteed to be convex and numerically tight". To my understanding, the authors have no guarantee that their result is numerically tight for every GLM, and thus the first sentence of the abstract is false (is there even a guarantee in the paper that there exists a single GLM for which it is numerically tight?).
2) Line 53, "Our main novely lies in __cleverly__ using [...]"; whether the approach is clever or not is perhaps something for readers to judge, not for the authors to proclaim.
3) Line 107, "we completely remove the poly(S)-dependency from the radius, resolving one of the __open problems__ posited by Lee et al. (__2024__)." An open problem is an well-known unsolved problem in a field (that is, a problem that numerous other researchers have attempted and failed to solve). Your "open problem" is not that. Rather, it is a single sentence in the middle of a paper published at a conference that took place less than 3 weeks before this paper was submitted that simply states that the authors leave it for future work.
4) Line 111-112 authors claim that "perhaps more importantly, [ellipsoidal confidence sequences allow] one to equivalently rewrite the optimistic optimization in the UCB algorithm as a closed-form bonus-based UCB algorithm". Is this true? The claim appears to be that, if $\mathcal{X}$ is a subset of the unit ball and $\Theta$ is an ellipsoid, then the quantity $b(\Theta) = \\max_{x \in \mathcal{X}} \max_{\theta \in \Theta} \langle x, \theta \rangle$ has a closed form expression. Could the authors state the closed form expression they have in mind for, say, $\mathcal{X}$ being an irregular polytope with $2^d$-many vertices? And if the resulting expression requires $O(2^d)$ time to evaluate... then the statement is trivial, and ellipsoidal form of the confidence sequences is unimportant. (and of course, fit a spline to those 2^d vertices, and now you cannot even write the solution as a sum over the vertices!).
5) Lines 307-308: in the conclusion, the authors state that "[their algorithm] is numerically verified to be the best performing algorithm." The authors compared against __one baseline__ on a __single experiment__ varying only __one experimental parameter__ between __two values__, and the experiment was __two dimensional__. *Really?*

Other than that, section 3.3 seems like it could contain interesting insight, but I'm worried that the way it is written, the only persons that might be able to understand it are either the authors themselves, or someone that has spent just as much time poring over the paper as the authors have done. I would suggest that the authors either expand on it and explain it better, or cut it.

Some minor typos:
- Line 312-313 you talk of "Bayesian randomized (exploration) algorithms for GLM bandits", but then cite 5 papers for these algorithms, out of which the four I am familiar with are frequentist algorithms, not Bayesian.
- Line 95-97, second half of sentence seems to be missing
- Line 129, the definition looks like the likelihood ratio, not the log-likelihood ratio as claimed
- Theorem 3.2, second line
- Line 82, the set in probability is wrong
- Line 75 rewards should be in filtration

**Questions:**

See weaknesses section.

**Limitations:**

.

---

> ### Author Rebuttal · Authors · 2024-08-05
>
> We thank the reviewer’s valuable reviews and comments.
>
> **W1. “Numerically tightness” of CS**
>
> We intended "numerically tight" to mean numerically tighter than previously known and non-vacuous, a phrase often used in prior works on PAC-Bayes. We will clarify this sentence in the revised manuscript.
>
> **W2. “Cleverly”**
>
> We concur with the reviewer that the novelty of the approach should be left to the reader's judgment. Accordingly, we will remove this word.
>
> **W3. Open problem should be well-known that numerous other people should’ve tackled and fail**
>
> We start by pointing out that obtaining a tight (hopefully optimal) regret bound for generalized linear bandits, including logistic bandits, has been a long unresolved open problem in bandits, starting from the seminal work of Filippi et al. (2010) and tackled by numerous researchers. Although we cited Lee et al. (2024) [1] as the source of the open problem, this is to emphasize the dependency of $S$ that has been ignored for quite sometime before [1]. We will fix the references to the original reference and provide more context into the importance of our open problem.
>
> We also highlight that Lee et al. (2024) [1] has been available on arXiv since October 2023. In our submission, we cited it using the BibTeX of its most recent conference version in 2024.
>
> **W4. Importance of ellipsoidal confidence set**
>
> Thank you for pointing this out. We would like to clarify that our original statement did not imply that the solution to the UCB $\max_{x \in X} \max_{\theta \in \mathcal{C}_t(\delta)} \mu(\langle x, \theta \rangle)$ has a closed solution. Upon further reflection, we acknowledge that the current wording may lead to misunderstanding, and we appreciate the reviewer for highlighting this. In Global Response #2, we elaborate on our intended meaning and how it can be corrected. We assure the reviewer that these corrections will be reflected in the revised manuscript. Additionally, as the reviewer mentioned, if the arm set is of combinatorial size, e.g., $2^d$, then we are indeed bound to incur at least $\Omega(2^d)$ computational cost. However, we demonstrate in Global Response #2 that if the CS is precisely an ellipsoid, the computational cost is $O(2^d)$, which is significantly more efficient than solving a convex optimization for each arm.
>
> **W5. Weak experiments**
>
> Please refer to Global Response #1.
>
> **Regarding Section 3.3**
>
> Thank you for your suggestion. We will make sure to explain the relationships more clearly or consider removing this section from the revision.
>
> **Typos**
>
> We apologize for the typographical errors and will commit to correcting them in the revision. We sincerely thank the reviewer for taking the time to point them out.
>
> To ensure clarity, we address each typo as follows:
> - Lines 312-313: The reviewer is correct. We will correct it to “randomized (exploration) algorithms.”
> - Lines 95-97: We will revise this sentence appropriately.
> - Line 129: It should indeed be the likelihood ratio.
> Theorem 3.2, second line: We will correct it to $\mathbb{P}(\exists t \geq 1 : \theta_\star \not\in \mathcal{E}_t(\delta)) \leq \delta$.
> - Line 82: We will correct it to $\mathbb{P}(\exists t \geq 1 : \theta_\star \not\in \mathcal{C}_t(\delta)) \leq \delta$.
> - Line 75: The rewards should indeed be in the filtration.
>
>
> We hope that our response has properly addressed the reviewer’s concerns and that the reviewer would reconsider the score. Thank you.

---

> ### Comment · Reviewer_q1y1 · 2024-08-10
> **Review of the review**
>
> I wanted to voice that I find this review to be an unfair, if not adversarial, assessment of the work. The review makes some comments about writing style until line ~110 and then something about the conclusion. This makes me wonder, if the reviewer read the full paper with sufficient care and give it enough thought.
>
> It has become common fashion to emphasize the paper's contributions in the introduction. Compared to the standard, I would **not** say that this paper is heavily exaggerating the results that follows. However, I realize that qualitative descriptions of the results can through-off a reader, e.g. them getting mad over the usage of a simple adjective ("cleverly").
>
> I think the paper indeed makes non-trivial contributions, and has a fresh approach (combining Ville's with the change of measure technique in the proof) that I have not seen in this immediate area of work.
> In their rebuttal, the authors also provide a benchmark including the most related work, even reporting the runtimes. The benchmark is still for a synthetic dataset, with small values of $S$, but already gives some insight into the average case performance of the algorithms (as the bounds are worst-case wrt the reward class).
>
> I think the review focuses on form/presentation, and misses out on the content. I am not sure how useful or reliable is this type of review, to make a fair assessment of soundness and relevance of the contributions.

---

### Official Review · Reviewer_q1y1 · 2024-07-08

**Soundness:** 4
**Presentation:** 4
**Contribution:** 2
**Rating:** 7
**Confidence:** 4

**Summary:**

This paper proposes Likelihood ratio based confidence sequences for generalized linear inference, who's width only depends logarithmically on $S$, the bound on norm of the parameter vector. This is achieved by utilizing a pac-bayesian change of measure inequality, with the prior and posterior distributions chosen very carefully. Tightness of the bounds are compared with some prior work and a small numerical experiment is provided to demonstrate potential benefits of the confidence bounds, for the application of logistic bandits.

**Strengths:**

- Key contribution is reducing the $\mathrm{poly}(S)$ dependency of the width of the confidence sets to $\log S$. Practitioners typically do not use state-of-the-art CS prescribed by theory because 1) they are tough to compute 2) depend on unknown parameter. This paper takes a step towards mending this gap by addressing the second issue.
Since dependence of the width on $S$ is logarithmically, then the practitioner can choose a conservatively large upper bound on $||\theta_\star||$ and not suffer from loose/uninformative CS.
- The work benefits from versatility of LR-based CS and automatically gives results that are applicable to multiple data/noise models.
- Gives convex CS for logistic bandits that have rate optimal dependence on $\kappa$ parameter. I might be wrong, but as far as I know, prior work with such dependency on $\kappa$ resort on non-convex sets [Faury 2022]. *I am not aware of the latest results, particularly [Lee 2024].*
- The proof is light and combines simple ideas from not-so-connected areas. Compared to prior work on parallel topics (on GLM, logistic, or poisson models), this seems like a more elegant approach to prove anytime validity of confidence sequences.

**Weaknesses:**

I think improving dependency on $S$ gains significant relevance, once it yields a practical algorithm.
If there is no computationally efficient/stable way of calculating the confidence sets, then the relevance of the results is limited to a subset of the bandit theory community. Maximising the UCB on the proposed sets (Thm 3.1 and Thm 3.2) does not seem to be a computationally feasible task, particularly for higher dimensions. Further, I'm afraid that practical relaxations/approximations, would blow up the width, and loosing the current theoretical edge.

- The ellipsoid sets are proposed as a easy-to-implement alternative (which actually might not be the case, due to the hessian in the norm). But it is not clear to me if they are similarly tight. Depending on the parameter $R_s$, these sets may again scale with $\mathrm{poly}(S)$.

- While the paper makes a valuable theoretical contribution, I think it would need more experiments to appeal to a broader community. For instance, a proper benchmarking of the CS against practically common choices (vanilla GP-based CS based on a very loose Linear regression model that uses a sub-Gaussian noise with a large variance) and showing that the statistical gains are worth the computational effort by reporting the regret, and the computational efficiency (e.g. number of flops).

- Experiments do lack comparison with relevant baselines and are only in the logistic case. In particular, Emmenegger 2023 and Faury 2022 are two algorithms that should be compared to. IIRC, Emmenegger 2023 does not work well for logistic bandits, so this would help strengthen the message of the paper. However, Faury 2022 seems to make a strong case on the joint computational and statistical efficiency of their algorithm (for logistic bandits).


This is not a weakness, but I should mention that I am not personally aware of the common rates and parameter dependencies in LR-based confidence sequences. Therefore, I *could not* verify the dependence of the width on certain parameters, so I don't know if everything is optimal, or we are sacrificing something else on the path to $\log S$.

**Questions:**

### Questions:
1) For equation (3), I did not understand why this is a batched estimator and not a sequential one. Can you clarify your terminology of batched vs sequential estimator?
2) Is there theoretical benefit/necessity to solving the constrained problem rather than regularizing? I realize that for validity of the CS, we require $\hat \theta_t$ to also lie within $\Theta$, but is there any other reason why we consider the MLE rather than generalized ridge loss? Asking because from a practical perspective, the latter is better/more stable.
3) What is the dependency of Theorem 3.2 on $S$? Mainly, do the ellipsoid sets blow up for logistic bandits?
4) Can you comment on the computational complexity of calculating the CS of Theorem 3.1 and Theorem 3.2? My feeling is that both are computationally tough to calculated (e.g. due to the lipschitz constant, or the matrix norm wrt the hessian matrix).
5) Do you see a clear kernelized extension? In particular, is there a choice of prior-posterior for which the KL divergence is bounded and allows for a similar rate?
6) Is Theorem 4.1 minimax optimal, e.g. wrt the lower-bound of Abeille 2021?
7) In Fig 1-(c), is the logistic CS is automatically an ellipsoid, or is it just resembling one?
8) Are you using the theoretical $\beta_t(\delta)$ for Fig (1-a) and (1-b)?
_____
### Small typos & Suggestions

- [Lines 57, 67, 69] space before parenthesis missing
- [Line 72] notation for derivates isn't coherent. Both $\dot{}$ and ${}^\prime$ are used.
- [Line 73] would make more sense to write for $t \geq 1$, since all statements are anytime.
- [Line 95] The sentence does not have a verb. Perhaps instead of "Despite" authors meant "Exists".
- [Line 100] First time I read "unbounded GML" I was confused. Would be good to mention that this refers to values o $\mu$.
- [Theorem 3.1] Second line should be $\mathcal L_t$, the subscript is missing.
- [Line 156] It is not clear what "it" refers to. Perhaps swap it out for "the analysis".
- [Line 161] Would be good to include a reference for the first equality in the equation. Does this hold in equality or is it an upper bound? Is this common knowledge? I did not know it.
- [Line 173] This line reads vaguely and informally, "even" is used twice. It's best if the justification is made rigorous or removed entirely.
- [Fig 1] To demonstrate the robustness of the algorithm to $S$ you could've chosen a much larger value and still significantly outperform the baseline. Would be nice to see the effect of S = 10**{1, 2, 3, 4}.

**Limitations:**

The proof technique might be limited to linear setting, with in my opinion can be viewed as a limitation, if one of the contribution points of the paper is the proof technique.

Practical limitations and applicability are not adequately addressed.

---

> ### Author Rebuttal · Authors · 2024-08-05
>
> We thank the reviewer for several enlightening questions and suggestions that will certainly improve our paper.
>
> **W1. & Q3. Statistical issues with ellipsoidal relaxation**
>
> Indeed, the ellipsoidal CS introduces an additional factor of $S$, as it uses the self-concordance control (Lemma A.4). For logistic bandits, the blowup is only by a factor of $S$, as $R_s = 1$ for Bernoulli distribution. Among the *ellipsoidal* CSs available for linear and logistic bandits, our ellipsoidal CS’s width is theoretically the tightest, ignoring absolute constants. Indeed, we believe that our ellipsoidal CS-based UCB algorithm attains $\mathcal{O}(d \sqrt{ST / \kappa_\star(T)})$ regret, which is strictly better than many baselines' regret guarantees, e.g., **ada-OFU-ECOLog**. Also, the complexity of computing the Hessian $H_t = \sum_{s=1}^t \dot{\mu}(\langle x_s, \hat{\theta}_t \rangle) x_s x_s^\top$ is similar to that of the design matrix in linear bandits, making our ellipsoidal CS practical.
>
> **W2. & W3. Lacking experiments**
>
> Please refer to our Global Response #1, where we address many of the reviewer’s suggestions and concerns about the experiments. As measuring the number of flops for constrained nonlinear convex optimization was unclear, in SPDF, we report the algorithms' runtimes (sec), measured by Python’s time module.
>
> **Q0. Are we sacrificing something for $\log S$?**
>
> Yes, obtaining $\log S$ requires more computational power (e.g., norm-constrained MLE). Please refer to our above response to W1 & W3.
>
> **Q1. Batched vs. sequential estimator?**
>
> We apologize for the confusion. These terms distinguish between $\mathcal{L}_t(\hat{\theta}_t)$, which uses a single MLE to compute the loss at $t$, and $\mathcal{L}_t(\\{\hat{\theta}_s\\}\_{s=1}^t)$, which uses all the MLEs computed to far to compute the loss at $t$ as in weighted likelihood ratio testing [4]. We will clarify this in the revision.
>
> **Q2. Why constrained MLE?**
>
> If we stick to the uniform prior, then we are not aware of any way to allow the regularized MLE since then we cannot ensure that there is a sufficient prior probability mass around the regularized MLE (the regularized MLE can be far outside of the support of the uniform prior). Constrained MLE ensures that the estimator falls in the support of the uniform prior, so one can find a sufficient probability mass around it.
>
> But then one may wonder why uniform prior? We made this choice since we found that the KL divergence between uniform prior and posterior has a closed form and that it provides us with much easier control of under-approximating the integral w.r.t. the posterior. Alternatively, one can consider, e.g., Gaussian prior, and attempt to make the regularized MLE work, but we found that terms are much harder to control. We have not done an exhaustive attempt on this, so we believe it is an interesting future direction!
>
> **Q4. & W1. Computational complexity of calculating the CS?**
>
> First, the radii in Theorem 3.1 and 3.2 can be explicitly computed using the Lipschitz constants for various GLMs provided in Table 1. Also, the complexity for computing the matrix norm w.r.t. the Hessian matrix $H_t = \sum_{s=1}^t \dot{\mu}(\langle x_s, \hat{\theta}_t \rangle) x_s x_s^\top$ isn’t too much, considering how its computation is composed of matrix-vector products. Please refer to our Global Response #2 for the computational issues in the UCB.
>
> **Q5. A clear kernelized extension?**
>
> No, extending to the RKHS setup seems non-trivial. The primary issue is that no translation-invariant Lebesgue measure exists in infinite-dimensional function space (e.g., RKHS), i.e., no uniform distribution and no concept of likelihood, among many counterintuitive issues. Indeed, the usual properties that hold in finite dimensions may fail in infinite dimensions (see, e.g., [13]), and one must be very careful when transferring the current PAC-Bayes proof to infinite dimensions. For instance, for the KL between two Gaussian measures to be well-defined, one measure must be absolutely continuous against another, which holds under certain nontrivial conditions on mean and covariance operators (Feldman–Hájek theorem). One could consider using the well-studied Gaussian Process prior/posteriors (as hinted at in our answer to **Q2**), but choosing the mean and kernel to obtain similar guarantees is non-trivial.
>
> **Q6. Is Thm 4.1 minimax optimal?**
>
> Somewhat yes. As elaborated in lines 250-272, the leading term of our regret bound for *logistic bandits* is indeed (locally) minimax optimal in $d, T, \kappa_\star(T)$ relative to the lower bound of [2]. Moreover, a closer look into their proof shows that their lower bound scales as $1 / S$, indicating a gap of $S$ between the lower and upper bounds. To the best of our knowledge, there isn’t any generic regret lower bound that holds simultaneously for all **GLB**s. We suspect a similar $d\sqrt{T / \kappa_\star(T)}$ lower bound would hold for **GLB**s. One could either carefully modify the proof of [2] for GLMs (e.g., their relative entropy decomposition lemma (Lemma 6) relies on the fact that the reward distribution is Bernoulli) or come up with something new. We leave this as future work.
>
> **Q7. Is Fig 1(c) ellipsoid?**
>
> No, our CS at each time $t$ is a level set of the negative log-likelihood loss $\mathcal{L}_t(\cdot)$, which isn’t precisely an ellipsoid. But, as the CS tightens, it more closely resembles an ellipsoid, as the second-order Taylor approximation (an ellipsoid as shown in our Theorem 3.2) gets tighter. We will elaborate on this in the revision.
>
> **Q8. Are you using theoretical $\beta_t(\delta)$ for experiments?**
>
> Yes, we use the precise theoretical $\beta_t(\delta)$ in all of our experiments, including additional ones in the supplementary pdf.

---

> > ### Comment · Reviewer_q1y1 · 2024-08-10
> > **Reviewer discussion**
> >
> > Thank you for your response and the new benchmarks.
> >
> > Interesting to see how close EMK and OfUGLB perform (statistically and computationally). I still think that you might be able to demonstrate a bigger advantage, if you consider significantly larger values for $S$ in the experiments, or consider benchmarks in which the true $S$ is not known.

---

### Official Review · Reviewer_TTTd · 2024-07-12

**Soundness:** 3
**Presentation:** 4
**Contribution:** 3
**Rating:** 7
**Confidence:** 2

**Summary:**

This work considers generalized linear models where the distribution of observations, conditionally on a context vector $x$ and an unknown parameter $\theta^\star$, are generated from a (known) exponential family of distribution. Their main contribution is a new confidence sequence for online estimates of $\theta^\star$, leading to improved regret in the context of Generalized Linear Bandits when used to calibrate a UCB algorithm corresponding to the model for the observations.

**Strengths:**

The paper is well written and easy to follow, and the presentation of the technical arguments is very pedagogical. The literature review seems extensive, and the authors carefully separated the literature that inspired the design of the confidence sequence and the literature related to GLM and bandits. The results are also well presented, with explicit constants so that it is not difficult to reproduce the results from the paper. In that regard, I appreciated how the authors cared about facilitating future implementation of their approach. I also appreciated the precise derivations proposed for some specific families of distributions, that are good illustrations of the results.

**Weaknesses:**

I am not very familiar with a large part of the literature invoked by the authors, basically the literature presented in Section 3.3. Hence, it is quite difficult for me to assess the technical contribution of the paper (not a weakness, but I'm using this space to say it). For a non-expert reader, Section 3.2 is a bit hard to follow. In particular, I did not see the connection between the result and Theorem 3 of Foster et al. (2018).

Regarding the bandit part, it might be beneficial to extend the "proof sketch" part to establish the main arguments that are different from previous works. In particular, l. 215 the authors say "one needs extra care in the analysis to ensure that the regret bound is also tight", but after that it seems that all arguments are standard. It might be interesting to provide more details or to remove that sentence.

**Questions:**

* The main novel arguments seem to be located in l.153-167. My understanding of the results (please correct me) is that the goal is to use Lemma 3.3 with a nice choice of prior/posterior distributions such as to make the KL divergence as small as possible. However, in Eq. (9) l.159 I fail to see why $P_t$ is a valid posterior given the prior and empirical data. Again, I am not a specialist of the PAC-Bayes literature so I'm sorry if the answer is obvious.

* In the bandit part, bounds are established for Self-Concordant GLM. Does the constant $R$ need to be known by the algorithm or is it just involved in the analysis?

* Can you elaborate on the connection with Theorem 3 of Foster et al. on Section 3.2?

---

> ### Author Rebuttal · Authors · 2024-08-05
>
> Thank you for your detailed review, for recognizing the significance of our technical contributions, and for providing valuable feedback.
>
> **W1. Writing and paper organization + Regret Analysis**
>
> Thank you for your suggestions. Indeed, Section 3.2 implicitly assumes that the reader is familiar with PAC-Bayesian analysis. Due to space constraints, we could not provide more details and instead referenced two standard sources [9,10]. In the revision, we will ensure Section 3.2 is easier to follow.
>
> Additionally, as the reviewer noted, the proof sketch of the regret bound can be improved. We will address this in the revision. Allow us to elaborate on the main technical novelty of our regret analysis, which will be included in the revised version.
>
> The existing proof relies on the original self-concordance lemma [11, Lemma 9], which allows the appearance of $\nabla^2 \mathcal{L}_t(\theta\_{*})$ (related to the matrix $H\_t(\\theta\_\*)$ in [11]). Consequently, it allows for one to use the elliptical potential lemma (Lemma B.1), with $x_s$ replaced by $\sqrt{\dot{\mu}(\langle \theta\_\star, x_s \rangle)} x_s$. However, as described in our proof sketch, the self-concordance lemma introduces an additional factor of $S$.
> To avoid this and at the same time to use our confidence sequence (Theorem 3.1), we utilize the Cauchy-Schwartz w.r.t. $\widetilde{G}_t(\widehat{\theta}_t, \nu_t) = \lambda \mathbf{I} + \frac{1}{g(\tau)} \sum\_{s=1}^{t-1} \tilde{\alpha}_s(\widehat{\theta}_t, \nu_t) x_s x_s^\top$ (see Eqn. (26) in Appendix B.1). $\widetilde{G}_t$ arises naturally when one performs second-order Taylor expansion of $\mathcal{L}_t(\cdot)$ around its minimum $\widehat{\theta}_t$; in particular the confidence set can be rewritten as a quadratic form involving $\widetilde{G}_t$ (Lemma B.6). However, now note that the elliptical arguments are no longer applicable to $\sum_t \lVert x_t \rVert\_{\tilde{G}_t(\hat{\theta}_t, \nu_t)^{-1}}^2$. Our main technical novelty is designating the "worst-case $\theta$" over all future confidence sets from time $s$ to $T$ (see $\bar{\theta}_s$ in Eqn. (24) of Appendix B.1). With this, we can follow an alternate chain of inequalities to obtain $\sum_t \lVert x_t \rVert\_{\bar{H}_t^{-1}}^2$, where $\bar{H}_t = 2 g(\tau) \lambda \mathbf{I} + \sum\_{s=1}^{t-1} \dot{\mu}(\langle \bar{\theta}_s, x_s \rangle) x_s x_s^\top$ (see Eqn. (26) in Appendix B.1).
> Then the elliptical potential lemma is applicable with $x_s$ replaced by $\sqrt{\dot{\mu}(\langle \bar{\theta}_s, x_s \rangle)} x_s$, concluding our proof.
> Note that there are other details we are omitting here, but they are relatively minor manipulation of adding and subtracting the desired quantities to get to the place where we can apply the elliptical potential lemma as we stated above.
>
> **Q1. Why $P_t$ is a valid posterior?**
>
> The terms "prior" and "posterior" are somewhat misleading in PAC-Bayes, as they need not strictly be the Bayesian prior and posterior. Instead, "prior" should be interpreted as any data-free distribution. and "posterior" as any data- and prior-dependent distribution, providing more flexibility in choosing them. The reviewer is correct that one should choose the "prior" and "posterior" such that their KL divergence is not too large. Simultaneously, in our context, the posterior should be chosen so that the Lipschitz inequality is not too loose (see line 164). We refer the reviewer to the excellent introduction to PAC-Bayes by P. Alquier [9] and a survey on PAC-Bayes time-uniform bounds by Chugg et al. [10]. There is no need to apologize; we also found it challenging to adapt to PAC-Bayes terminologies while writing the paper.
>
> **Q2. Does the learner require the knowledge of $R_\mu$?**
>
> Thank you for highlighting this point. Indeed, the learner needs to know $R_{\dot{\mu}}$ or its upper bound. This requirement arises because the bandit algorithm relies on our new confidence sequence (Theorem 3.1), which in turn depends on the Lipschitz constant $L_t$ of the GLM negative log-likelihood, which depends on $R_{\dot{\mu}}$; see Table 1 and Appendix A of our submission. We will clarify this in the revision.
>
> **Q3. Elaborating on the connection of Theorem 3 of Foster et al. (2018) with our Section 3.2**
>
> To recall the proof of Theorem 3 of [12], they first consider a distribution $P_t(\cdot)$ over the parameter $W$ (see their Algorithm 1) and use $\eta$-mixability of the logistic loss to obtain an inequality involving the negative-log-integral term $\int_{\mathcal{W}} \exp\left( -\eta \sum_t \ell(Wx_t, y_t) \right) dW$. They define $S = \theta W^\star + (1 - \theta) \mathcal{W} \subseteq \mathcal{W}$, where $W^\star$ is the ground-truth optimal parameter and $\theta \in [0, 1)$ is to be determined later. The proof concludes by chaining the inequality $\int_{\mathcal{W}} \geq \int_S$ with the Lipschitzness of the logistic loss and expanding the integral.
>
> Our Section 3.2 follows a similar approach with some key differences. While their negative-log-integral also arises in our scenario, we adopt a more compact, streamlined PAC-Bayes approach. In our case, a similar quantity $\mathbb{E}\_{\theta \sim \mathbb{Q}}[\exp(-\mathcal{L}_t(\theta))]$ arises from our Donsker-Varadhan representation (Lemma 3.2), where $\mathbb{Q}$ is our prior. We then apply Ville’s inequality to obtain the time-uniform PAC-Bayes bound (Lemma 3.3), and our choices of prior/posterior resemble their choice of $S$. Our Lipschitzness arguments also resemble their $\ell\_{\infty}$ Lipschitzness argument (see their pg. 17 in the PMLR proceeding version).
>
> We will provide a more detailed explanation of the proof in the revision. Thank you for bringing this to our attention.

---

> > ### Comment · Reviewer_TTTd · 2024-08-12
> >
> > Thank you very much for your detailed response, I have no more questions for now.

---

### Author Rebuttal · Authors · 2024-08-05

We sincerely thank all the reviewers for providing detailed and insightful reviews. We are especially encouraged to see that the reviewers recognize the simple yet effective proof ideas (q1y1, HLaD), technically solid contribution in reducing $poly(S)$ to $\log S$ in the CS width (q1y1, HLaD, aQNL), easy-to-follow and pedagogical exposition (TTTd), thorough literature review (TTTd), and the importance of providing explicit constants for practitioners (TTTd). Here, we address two issues raised by multiple reviewers. In each reviewer's reply, we answer the remaining reviewers' comments point-by-point. We also attach a supplementary PDF (SPDF) containing additional experimental results.
# 1. Lack of experimental verifications (q1y1, HLaD)
Our experiments are meant to show that the theoretical improvement in $S$ for the logistic bandits is also numerically meaningful, as done in [1]. Still, we agree that our paper can benefit from more experiments to appeal to a broader community. SPDF provides additional experimental results for logistic bandits (Figure 1). We will make the codes publicly available.

We expand upon the considered settings by varying $S\in\\{5, 7, 9, 11\\}$. We tried $S\in\\{10^2, 10^3\\}$, but we could not handle the numerical instability from the optimization problems in time. We will *continue to try* to get the higher $S$ to work as well. We fix $d = 2$ due to the lack of time and space in SPDF, and also because only then can the complete confidence sets be fully visualized without any projection. We will *continuously work* to include additional experimental results for other values of $d$ in the revision. Also, as suggested by both reviewers, we will work to extend our codebase to linear/Poisson bandits and include the results in the revision.

Notably, we increased the number of baselines to eight, two of which are ours: **OFUGLB**, **OFUGLB-e** (ellipsoid CS), **EMK** [4], **RS-GLinB** [5], **OFULog+** [1], and **ada-OFU-ECOLog** [3]. **EMK** has been included as a practically common choice, as it performs better than other CS-based UCB algorithms, including GP-based CS [6]. For consistent experiments, we use the same setting as in our submission.

The results show that even against the additional baselines, our **OFUGLB** attains the best numerical regret. Compared to **EMK**, note that for small $S$, ours performs slightly worse. Still, for large $S$, the trend suggests that **OFUGLB** eventually attains smaller regret than **EMK**, suggesting that the theoretical regret of **EMK** may be looser than ours in terms of $S$. Also, we emphasize that *we provide a rigorous, state-of-the-art regret guarantee* for **OFUGLB**, while their theoretical guarantee scales with $\kappa$. The ellipsoidal version **OFUGLB-e** is also shown to have reasonable performance and even outperforms **ada-OFU-ECOLog** for small values of $S$. Of all considered algorithms, **RS-GLinCB** seems to have the highest regret, even though it also has $poly(S)$-free regret. We believe this significant difference is from the fact that **RS-GLinCB** involves an explicit warmup stage, while ours doesn’t. Indeed, in [5], the authors considered $20000$ rounds for logistic bandits, where **RS-GLinCB** is shown to eventually outperform **ada-OFU-ECOLog**, while we only considered $4000$. We will run the additional experiments with increased rounds for the revision for a fair comparison.

In SPDF's Table 1, we additionally report the runtime (sec) of a single run of each algorithm (measured via Python’s time module) for the logistic bandit instances. Note that **OFUGLB-e** is generally faster than **OFUGLB** by roughly 6 minutes. Also, **EMK** is generally slower than **OFUGLB**, as it needs to keep track of the entire sequence of estimators.
# 2. Computational feasibility of UCB (q1y1, HLaD)
Here, computational feasibility is being implementable using (efficient) convex solvers as subroutines, which is relatively standard in bandit literature [1,4,7]. While writing the response, we realized that the Theorem 3.2 should be $\mathcal{E}_t(\delta)=\left\\{\theta\in\mathbb{R}^d : \lVert\theta-\hat{\theta}_t\rVert\_{H_t}\leq\gamma_t(\delta)\right\\}$. We apologize for the confusion.

With this, let us assume that the arm set $X$ is finite. Then, it is easy to see that the UCB using the ellipsoidal CS $\mathcal{E}_t(\delta)$ is computationally efficient, as it reduces the following closed-form *objective*:

$$
\mathrm{argmax}\_{x \in X}\langle x,\hat{\theta}_t\rangle + \sqrt{\gamma_t(\delta)}\lVert x\rVert\_{H_t^{-1}}.
$$

In other words, there is no need to solve a convex optimization at each time $t$ and each arm $x \in X$.
In this case, the computational complexity of performing UCB at time $t$ is $\mathcal{O}(T\_{MLE}(t) + d^3)$, where $T\_{MLE}(t)$ is the complexity of computing the MLE at time $t$ and $d^3$ is the complexity of involved matrix computations, including computing $H_t^{-1}$.

Using the likelihood ratio-based CS $\mathcal{C}_t(\delta)$ still results in a convex optimization problem, albeit a bit less efficiently solvable than using $\mathcal{E}_t(\delta)$ due to lack of structure. We also remark that computational issues in high dimensions are present in many prior approaches to linear and logistic bandits [1,4,7], and our approach doesn’t introduce additional complexity relative to them.

We emphasize that we primarily focus on achieving the tightest regret guarantee for **GLB**s while being computationally *tractable*. We agree with the reviewers that obtaining jointly statistically tight and computationally *efficient* algorithms (e.g., [3]) is an important future direction but outside this paper’s scope.

---

### Author Response · Authors · 2024-08-07
**References for the rebuttal**

Let us collect all the relevant references for our rebuttal below.


[1] Lee et al. “Improved Regret Bounds of (Multinomial) Logistic Bandits via Regret-to-Confidence-Set Conversion.” AISTATS 2024.


[2] Abeille et al. “Instance-Wise Minimax-Optimal Algorithms for Logistic Bandits.” AISTATS 2021.


[3] Faury et al. “Jointly Efficient and Optimal Algorithms for Logistic Bandits.” AISTATS 2022.


[4] Emmenegger et al., “Likelihood Ratio Confidence Sets for Sequential Decision Making.” NeurIPS 2023.


[5] Sawarni et al. “Generalized Linear Bandits with Limited Adaptivity.” arXiv preprint arXiv:2404.06831.


[6] Neiswanger & Ramdas. “Uncertainty quantification using martingales for misspecified Gaussian processes.” ALT 2021.


[7] Flynn et al. “Improved Algorithms for Stochastic Linear Bandits Using Tail Bounds for Martingale Mixtures.” NeurIPS 2023


[8] Boyd & Vandenberghe. “Convex Optimization.” Cambridge University Press, 2004.


[9] Alquier. “User-friendly Introduction to PAC-Bayes Bounds.” Foundations and Trends in Machine Learning: Vol. 17: No. 2, pp 174-303, 2024.


[10] Chugg et al. “A Unified Recipe for Deriving (Time-Uniform) PAC-Bayes Bounds.” Journal of Machine Learning Research, 24(372):1-61, 2023.


[11] Faury et al. “Improved Optimistic Algorithms for Logistic Bandits.” ICML 2020.


[12] Foster et al. “Logistic Regression: The Importance of Being Improper.” COLT 2018.


[13] Gawarecki & Mandreka. “Stochastic Differential Equations in Infinite Dimensions with Applications to Stochastic Partial Differential Equations.” Springer Berlin, Heidelberg, 2010.


[14] McCullagh & Nelder. “Generalized Linear Models.” Chapman & Hall/CRC, 2 edition, 1989.


[15] Carpentier & Locatelli. “Tight (Lower) Bounds for the Fixed Budget Best Arm Identification Bandit Problem.” COLT 2016.


[16] Lattimore & Szepesvari. “Bandit Algorithms.” Cambridge University Press, 2020.


[17] Jun et al. “Improved Confidence Bounds for the Linear Logistic Model and Applications to Bandits.” ICML 2021.

---

### Decision · Program_Chairs · 2024-09-25

**Decision:**

Accept (poster)

**Comment:**

The paper proposes a new way to obtain anytime sequential confidence regions for the parameters of generalized linear models (GLM) with application to UCB-style algorithms in GLM bandits. It's a topic that has received a renewed interest recently, with novel ideas, based notably on the self-concordant property (in particular for logistic models). This paper provides a new construction of independent interest -with an ellipsoidal region around the nom-constrained MLE, whose coverage is guaranteed through arguments based on PAC Bayesian ideas- as well as an improved regret bound for GLM bandits.

Most reviewers evaluated the contribution positively while making a number of important remarks, which I hope the authors will take into account in their revision: better organization of the paper, more careful wording of the contributions, improved experimental part, more clearly highlighting important issues such as Rebuttal.2, TTTd.Q2, q1y1.Q0 and Q2, aQNL typos and missing proof.

Both the technical content and the potential of the method in GLM bandit applications are interesting and justify acceptance for the conference.